# Learnable Koopman-Enhanced Transformer-Based Time Series Forecasting with Spectral Control

## Abstract

This paper proposes a unified family of learnable Koopman operator parameterizations that integrate linear dynamical systems theory with modern deep learning forecasting architectures. We introduce four learnable Koopman variants—`scalar-gated`, `per-mode gated`, `MLP-shaped` spectral mapping, and `low-rank` Koopman operators—which generalize and interpolate between strictly stable Koopman operators and unconstrained latent linear dynamics. The proposed formulation enables explicit control over the spectrum, stability, and rank of the latent transition operator while remaining compatible with expressive nonlinear forecasting backbones such as `PatchTST`, `Autoformer`, `Informer`, `iTransformer`, and `TimesNet`.

We evaluate the proposed operators in a large-scale benchmark including LSTM, `DLinear`, simplified dense State-Space Models (SSMs), and lightweight Transformer variants across multiple forecasting horizons and patch lengths. Empirical results show that Koopman-enhanced models achieve competitive forecasting performance while exhibiting more structured and well-conditioned latent dynamics. The experiments further demonstrate that the empirical impact of the Koopman layer depends strongly on the underlying forecasting backbone and dataset, with predictive improvements that are generally modest and not uniform across all configurations. In particular, the proposed framework is better interpreted as introducing structured and spectrally controlled latent dynamics rather than universally improving forecasting accuracy.

Beyond predictive evaluation, we provide a detailed analysis of the learned latent operators, including learned eigenvalue distributions, spectral conditioning behavior, and visualizations of the imposed stability constraints across different Koopman parameterizations. These analyses offer insight into how different spectral parameterizations influence latent stability, coupling structure, and representation geometry across heterogeneous forecasting architectures. The results suggest that learnable Koopman operators provide a principled and flexible framework for incorporating controlled linear latent dynamics into deep forecasting models, with potential benefits in stability, conditioning, and interpretability.

Importantly, the proposed Koopman module is encoder-agnostic and can be combined with a broad class of neural forecasting backbones, including both Transformer-based and non-attention architectures such as `DLinear` and SSMs.

## 1 Introduction

Time series forecasting underpins applications in retail demand Wen et al. (2017); Salinas et al. (2020), traffic management Lv et al. (2014); Li et al. (2018), Energy Systems Dimoulkas et al. (2019); Saxena et al. (2019); Forootani et al. (2025), and finance Callot et al. (2017); Yan et al. (2018). The growing scale, dimensionality, and non-stationarity of modern data challenge classical statistical methods Hyndman & Athanasopoulos (2018), motivating deep learning approaches with greater expressive power and scalability. Early neural models—including RNNs/LSTMs Hochreiter & Schmidhuber (1997); Salinas et al. (2020) and CNN/TCN architectures LeCun & Bengio (1995); Bai et al. (2018)—learned temporal dependencies directly

from data. Transformer-based models Wu et al. (2021); Zhou et al. (2021); Nie et al. (2022); Vaswani et al. (2017) later became dominant due to self-attention's ability to capture long-range dependencies, achieving strong results in benchmarks such as M4 and M5 Smyl (2020); Makridakis et al. (2021).

Numerous variants improve efficiency and structure: `Informer` Zhou et al. (2021) proposes ProbSparse attention; `Autoformer` Wu et al. (2021) incorporates trend–seasonal decomposition; `FEDformer` Zhou et al. (2022) leverages Fourier mixing; and `Pyraformer` Liu et al. (2022a) models multi-scale hierarchies. In contrast, `PatchTST` Nie et al. (2022) demonstrates that simple design choices can outperform deeper Transformers Zheng et al. (2014). The `iTransformer` Liu et al. (2023a) revisits the role of tokenization by treating variables as tokens rather than time steps, thereby emphasizing cross-variable dependencies while maintaining efficient temporal modeling. This formulation has been shown to improve scalability and performance in high-dimensional multivariate settings. In contrast, `TimesNet` Wu et al. (2022) departs from attention mechanisms and introduces a temporal convolution-based framework that captures multi-periodic patterns by reshaping one-dimensional time series into two-dimensional representations. This design enables effective modeling of complex seasonal and multi-scale temporal structures without relying on self-attention. These developments highlight that accurate forecasting can be achieved with diverse architectural choices, motivating the need for frameworks that remain compatible with a broad class of backbone models. Moreover, `DLinear` Zeng et al. (2023) reveals that even channel-wise linear models can rival complex architectures, raising a key question: *how much architectural complexity is truly necessary for accurate forecasting?* Despite their success, deep models often lack interpretability, struggle under distribution shifts Kuznetsov & Mohri (2020); Liu et al. (2022b); Kim et al. (2022), and fail to encode explicit dynamical structure, limiting robustness. Recent work incorporates domain priors through hybrid statistical–deep models Wang et al. (2019); Sen et al. (2019); Smyl (2020), neural ODE frameworks Chen et al. (2018); Vialard et al. (2020), and physics-inspired designs, though such approaches frequently rely on fixed dynamics assumptions and face scalability challenges in complex multivariate settings.

## 1.1 Koopman Operators and Deep Learning

The *Koopman operator* provides a linear perspective on nonlinear dynamical systems by evolving observables instead of states. As an infinite-dimensional linear operator on state functions, it enables spectral analysis of nonlinear, nonstationary, or chaotic systems Mezić (2017); Khosravi (2023). Data-driven methods such as Dynamic Mode Decomposition (DMD) approximate the Koopman spectrum for prediction and modal analysis Williams et al. (2015); Kutz et al. (2016); Drgona et al. (2022); Skomski et al. (2021); Drgona et al. (2021), but often degrade in high-dimensional, noisy, or multivariate settings. Deep learning has renewed interest in Koopman theory by learning nonlinear embeddings with linear latent evolution. Neural Koopman models typically use an encoder $\mathcal{E}\theta$ to map an input window $X_t$ to latent coordinates $z_t = \mathcal{E}\theta(X_t)$, followed by a linear propagator $z_{t+1} = \mathcal{K}z_t$, often combined with nonlinear decoders and spectral or stability constraints Lusch et al. (2018); Yeung et al. (2019); Azencot et al. (2020); Morton et al. (2018); Takeishi et al. (2017). These approaches improve interpretability and rollout stability, with extensions incorporating low-rank structure, symmetry priors, and multiresolution representations Brunton et al. (2021). However, most remain architecturally shallow and struggle with long-range, non-local dependencies effectively handled by Transformer models.

Recent work integrates Koopman operators into Transformer architectures to achieve stable and interpretable latent dynamics, notably `DeepKoopFormer` Forootani et al. (2026); Forootani & Khosravi (2026). Unlike prior Koopman-based forecasting models Lusch et al. (2018); Azencot et al. (2020), `DeepKoopFormer` provides provable spectral conditioning properties, Lyapunov-consistent latent regularization, and operator-level perturbation bounds within a flexible encoder–propagator–decoder framework aligned with physics-informed forecasting Karniadakis et al. (2021); Nghiem et al. (2023) and interpretable dynamical systems modeling Brunton et al. (2016). Koopa Liu et al. (2023b) proposes a Koopman-based forecasting architecture that disentangles time-invariant and time-variant dynamics using Fourier filtering and localized Koopman predictors. The framework combines globally learned and locally estimated Koopman operators to model non-stationary temporal dynamics while improving computational efficiency and scalability across long forecasting horizons.

Existing Koopman-based forecasting methods have demonstrated the value of linear latent evolution for improving interpretability and dynamical consistency, but most approaches rely on either fixed Koopman

operators or unconstrained latent linear layers with limited spectral control. In addition, prior methods are often developed for relatively shallow encoder architectures and are not systematically integrated with modern Transformer-based forecasting backbones capable of modeling long-range temporal dependencies. Consequently, the interaction between spectral structure, latent stability, and expressive neural forecasting architectures remains insufficiently understood. The proposed framework addresses this gap by introducing a unified family of learnable and explicitly parameterized Koopman operators that provide controllable spectral dynamics while remaining compatible with a broad range of Transformer and non-Transformer forecasting models.

### 1.2 Contributions: Learnable Koopman Framework with Neural Backbones

Although promising, most Koopman–neural models rely on fixed or heavily constrained operators, leaving much of the learnable operator space unexplored. Existing approaches typically pair nonlinear encoders with a latent linear operator Takeishi et al. (2017); Lusch et al. (2018), but the operator is usually learned *without explicit spectral parameterization*—implemented as an unconstrained linear layer or least-squares fit. This offers no direct control over contraction, oscillation, or multi-scale temporal dynamics. Furthermore, such designs are rarely integrated with modern Transformer architectures, despite their strength in modeling long-range dependencies. The resulting combination of *spectral non-control* and *architectural underuse* limits both interpretability and dynamical expressiveness.

To address these limitations, we introduce the `Learnable-DeepKoopFormer`, a unified framework that equips neural forecasting backbones with *parameterized Koopman operators* whose spectra are learned directly from data. While we instantiate the framework with Transformer-based encoders (`PatchTST`, `Informer`, `Autoformer`, `iTransformer`, `TimesNet`), the proposed Koopman module is not specific to attention architectures and can be integrated with alternative backbones such as linear models (`DLinear`) and state-space models (SSMs).

Our formulation includes four operator families: (i) `scalar-gated`: global trainable parameters $(\alpha, \beta)$ control spectral shifting and scaling, enabling tunable damping or persistent dynamics; (ii) `per-mode gated`: dimension-wise parameters $(\alpha_i, \beta_i)$ model anisotropic temporal responses and multi-frequency evolution; (iii) `MLP-shaped` spectral mapping: neural spectral mappings are projected onto stable linear propagators with controlled spectral radius; (iv) `low-rank` Koopman: relatively low rank factorizations of the Koopman operator to capture high dimensional latent dynamics efficiently through structured compression. These operator parameterizations are integrated into both lightweight and full Transformer forecasters—`PatchTST` Nie et al. (2022), `Informer` Zhou et al. (2021), and `Autoformer` Wu et al. (2021)—forming a unified family of `DeepKoopFormer` models. Spectral stability constraints and operator regularization enable joint optimization of latent dynamics and representations while retaining interpretability, as dominant modes, stability margins, and spectral envelopes remain directly analyzable.

We conduct an extensive simulation benchmark comparing the proposed learnable Koopman variants against two Koopman baselines (spectrally constrained and unconstrained), as well as standard forecasting models including LSTM, `DLinear` Zeng et al. (2023), and a simplified dense SSM Gu et al. (2021b). Beyond forecasting potential, we present the first systematic *spectral evaluation* of Koopman–Transformers, analyzing eigenvalue densities, training trajectories, spectral constraint visualization, and horizon sensitivity. In summary, this work provides: (i) a unified parameterization of *learnable Koopman operators* (scalar, per-mode, neural, and low-rank); (ii) the `Learnable-DeepKoopFormer` framework, embedding these operators into three Transformer backbones; (iii) a benchmark of *33 models* spanning constrained, learnable, and unconstrained variants, plus LSTM, `DLinear`, and simplified dense SSM baselines; (iv) the first comprehensive *spectral analysis of Koopman–Transformer models*, revealing stability, multi-scale expressiveness, and interpretable latent dynamics.

For real-world evaluation, we utilize high-dimensional datasets from multiple domains, including: the CMIP6 climate projections [1], focusing on Wind Speed and surface pressure forecasting over Germany; a financial

---

[1] https://cds.climate.copernicus.eu/datasets/projections-cordex-domains-single-levels?tab=overview

time series dataset [2] for Cryptocurrency market analysis; and an electricity generation dataset [3] for modeling energy supply dynamics. These datasets collectively span chaotic, periodic, and stochastic regimes, allowing for a comprehensive assessment of model accuracy, stability, and generalization across domains.

Compared to existing Koopman-based forecasting models Lusch et al. (2018); Takeishi et al. (2017); Azencot et al. (2020), `Learnable-DeepKoopFormer` generalizes the original `DeepKoopFormer` framework from a fixed spectrally constrained Koopman operator to a learnable and explicitly parameterized family of spectrally controlled latent propagators. While the encoder–Koopman–decoder structure, ODO parameterization, and Lyapunov-based regularization are inherited from `DeepKoopFormer`, the present work introduces adaptive spectral shaping mechanisms, multiple learnable Koopman parameterizations, and a systematic evaluation across a substantially broader range of forecasting backbones. The resulting framework remains aligned with broader directions in physics-informed forecasting Karniadakis et al. (2021); Nghiem et al. (2023) and interpretable dynamical systems modeling Brunton et al. (2016); Korda & Mezić (2016).

**Novelty relative to `DeepKoopFormer`** The proposed `Learnable-DeepKoopFormer` builds upon the architectural and theoretical foundations of `DeepKoopFormer`, including the encoder–Koopman–decoder structure, orthogonal–diagonal–orthogonal (ODO) parameterization, and Lyapunov-based stability regularization. These components are inherited and are not claimed as new contributions in this work. The primary novelty of this paper lies in extending the Koopman propagator from a fixed, spectrally constrained operator to a *learnable and explicitly parameterized family* of operators with controllable spectral structure. In particular, we introduce (i) `scalar-gated` and `per-mode gated` parameterizations enabling adaptive spectral scaling and shifting, (ii) neural (`MLP-shaped` spectral mapping) spectral mappings that allow nonlinear shaping of the Koopman spectrum, and (iii) `Learnable-DeepKoopFormer` operators that impose structured dimensionality constraints on latent dynamics. Moreover, `Learnable-DeepKoopFormer` considers systematic integration across a substantially broader set of forecasting backbones, including `PatchTST`, `Informer`, `Autoformer`, `iTransformer`, `TimesNet`, `DLinear`, and SSM variants, whereas the original `DeepKoopFormer` experiments were limited primarily to `PatchTST`, `Informer`, and `Autoformer`.

From a theoretical perspective, we generalize the stability and expressiveness properties of `DeepKoopFormer` to this broader class of learnable operators, showing that spectral control, contraction, and interpretability can be preserved under flexible parameterizations. In addition, we provide the first systematic empirical and spectral analysis of learnable Koopman operators within Transformer-based forecasting architectures, highlighting how different spectral parameterizations influence stability, conditioning, and forecasting performance. These contributions collectively define a new design space for Koopman-enhanced deep forecasting models beyond the fixed-operator setting of prior work.

This paper is organized as follows. Section 2 provides the preliminaries required for time series forecasting. In Section 3, we present `Learnable-DeepKoopFormer` architectures for multivariate time series forecasting. Numerical simulations are presented in Section 4 to evaluate the performance of `Learnable-DeepKoopFormer` versus other benchmarks. Finally, we conclude the article in Section 5.

## 2 `Learnable-DeepKoopFormer` Architecture

We begin by formalizing the multivariate time series forecasting task and introducing the key components of the proposed architecture. Given a multivariate time series $\{x_t\}_{t=1}^T$, with $x_t \in \mathbb{R}^{d_s}$, `Learnable-DeepKoopFormer` operates on context windows of length $P$ and predicts the subsequent $H$ samples. For each valid index $t$, an input segment $X_t = [x_t, \ldots, x_{t+P-1}] \in \mathbb{R}^{P \times d_s}$ is paired with its future sequence $Y_t = [x_{t+P}, \ldots, x_{t+P+H-1}] \in \mathbb{R}^{H \times d_s}$. The input window is passed through a Transformer encoder with positional encoding $\mathcal{E}_\theta$ (e.g., `PatchTST`, `Autoformer`, `Informer`, `iTransformer`, `TimesNet` or other neural backbones such as `DLinear` or SSM), producing a latent representation $z_t = \mathcal{E}_\theta(X_t) \in \mathbb{R}^{d_{\text{lat}}}$, where $d_{\text{lat}}$ is the latent dimension.

---

[2] https://github.com/Chisomnwa/Cryptocurrency-Data-Analysis
[3] https://github.com/afshinfaramarzi/Energy-Demand-electricity-price-Forecasting/tree/main

Typically $d_{\text{lat}} = d_{\text{model}}$, and not necessarily equal to the physical dimension $d_s$, where $d_{\text{model}}$ denotes the hidden (embedding) dimension of the Transformer backbone, i.e., the dimensionality of the token representations and of the latent state produced by the encoder $\mathcal{E}_\theta$. With the slight abuse of notation and for the sake of simplicity, hereinafter we assume $d_{\text{model}} = d_{\text{lat}} = d_s = d$.

Temporal evolution in the latent space is modeled through a learned Koopman operator $\mathcal{K}_\phi$, which advances the latent state linearly according to $z_{t+1} = \mathcal{K}_\phi z_t$. The propagated latent state is decoded into a direct $H$-step forecast using a linear mapping $\mathcal{D}_\varphi$, yielding a vector $\hat{y}_t = \mathcal{D}_\varphi(z_{t+1}) \in \mathbb{R}^{H \cdot d}$, which is then reshaped into the output sequence $\hat{Y}_t \in \mathbb{R}^{H \times d}$.

Here, the subscript $\theta$ denotes the parameters of the Transformer encoder $\mathcal{E}_\theta$, $\phi$ collects the parameters of the Koopman propagator $\mathcal{K}_\phi$ (including its spectral coefficients and orthogonal factors), and $\varphi$ denotes the parameters of the linear decoder $\mathcal{D}_\varphi$. During inference, a context window $X_t$ is encoded once, propagated through the Koopman operator $\mathcal{K}_\phi$, and decoded to produce the entire prediction horizon $H$ in a single forward pass, without autoregressive rollout.

All components are trained jointly in an end-to-end fashion using the Adam optimizer Kingma & Ba (2015), a first-order stochastic gradient method with adaptive moment estimation. In particular, training is performed end-to-end by minimizing a forecasting loss (e.g., mean squared error) augmented with a Lyapunov-inspired penalty as follows

$$\begin{aligned}
\mathcal{L} &= \mathbb{E}\left[\|\widehat{Y}_t - Y_t\|^2\right] + \lambda_{\text{Lyap}}\, \mathbb{E}\left[\left\|\mathcal{K}_\phi \boldsymbol{z}_t\right\|_P^2 - \left\|z_t\right\|_P^2\right]_+ \\
&= \mathcal{L}_{\text{MSE}} + \lambda_{\text{Lyap}}\mathcal{L}_{\text{Lyap}},
\end{aligned} \tag{1}$$

where $\|z\|_P^2 = z^\top P z$, $P \succ 0$ is a fixed positive-definite matrix, $[x]_+ = \max\{x, 0\}$, and $\lambda_{\text{Lyap}} > 0$ is a scalar regularization weight controlling the strength of the Lyapunov stability penalty.

The architecture presented above provides a unified encoder–Koopman–decoder framework in which modern neural forecasting backbones are coupled with a structured latent dynamical system. The encoder is responsible for extracting nonlinear temporal representations from the input sequence, while the Koopman layer imposes a controlled linear evolution in latent space before decoding the resulting representation into a multi-horizon forecast. Importantly, the overall architecture remains unchanged across all proposed models; the primary distinction lies in how the Koopman propagator $\mathcal{K}_\phi$ is parameterized and constrained. This separation between representation learning and latent dynamical evolution allows us to systematically investigate how different spectral structures influence stability, conditioning, interpretability, and forecasting performance. In the next section, we therefore introduce the family of learnable Koopman parameterizations that define the core contribution of the proposed `Learnable-DeepKoopFormer` framework.

## 3   Learnable Koopman Variants

We introduced the general `Learnable-DeepKoopFormer` pipeline and its encoder–Koopman–decoder structure. We now focus on the central component of the framework: the learnable Koopman propagator. While all model variants share the same neural backbone and forecasting pipeline, they differ in how the latent transition operator $K_\phi$ is constructed, parameterized, and spectrally constrained. The goal of this design is to balance dynamical stability and expressive capacity by controlling the spectrum of the latent linear evolution. Rather than using a fully unconstrained latent transition matrix, we introduce a family of structured Koopman operators that progressively interpolate between strictly stable operator-theoretic models and highly flexible data-driven latent dynamics. This formulation enables explicit control over contraction, mode coupling, spectral shaping, and low-rank structure, while remaining compatible with a broad class of Transformer and non-Transformer forecasting backbones.

In `Learnable-DeepKoopFormer` we assume the latent evolution to be modeled by a linear dynamics through a learned Koopman propagator $\mathcal{K}_\phi \in \mathbb{R}^{d \times d}$, $z_{t+1} = \mathcal{K}_\phi z_t$.

All proposed variants share the same orthogonal–diagonal–orthogonal (ODO) decomposition of the Koopman operator. The key difference between variants lies exclusively in how the diagonal spectral coefficients are

generated from learnable parameters. Consequently, the encoder architecture, decoder structure, and overall forecasting pipeline remain fixed across models, allowing the effect of spectral parameterization itself to be isolated and analyzed.

As shown later in the paper, to ensure identifiability and stable training, $\mathcal{K}_\phi$ is parameterized through an orthogonal–diagonal–orthogonal (ODO) factorization

$$\mathcal{K}_\phi \;=\; U_\phi \operatorname{diag}(\Sigma_\phi)\, V_\phi^\top, \tag{2}$$

where $U_\phi, V_\phi$ are learned orthonormal matrices obtained via QR retraction, and $\Sigma_\phi = (\Sigma_{\phi,1}, \ldots, \Sigma_{\phi,d})^\top \in \mathbb{R}^d$ (or $\mathbb{R}^r$ in the low-rank case, see later in the paper) the vector containing the learnable spectral coefficients. Equation (2) ensures that all nonlinearity and expressiveness are concentrated in the spectrum $\Sigma_\phi$, while the left/right singular directions evolve on the Stiefel manifold, i.e., the set of matrices with orthonormal columns Absil et al. (2008).

The key design freedom of our *learnable Koopman architecture* lies in how the diagonal spectral coefficients $\Sigma_\phi$ are generated. We consider Koopman variants that share the principle $\Sigma_{\phi,i} \;=\; \rho_{\max}\, \sigma_\phi\big(S_i\big), \qquad i = 1, \ldots, d,$ where $S_i \in \mathbb{R}$ are raw trainable parameters, $\rho_{\max} \in (0,1)$ is a prescribed spectral bound, and $\sigma_\phi(\cdot)$ is a smooth squashing map, i.e. a nonlinear "spectral shaping" function, which preserves differentiability and enables end-to-end learning while providing explicit control over contraction rates in the latent Koopman dynamics. This map is chosen to enforce

$$|\Sigma_{\phi,i}| < \rho_{\max} \quad \text{for all } i, \qquad \rho_{\max} \in (0,1), \tag{3}$$

whenever spectral stability is desired, while allowing different degrees of expressiveness. In particular, it is the composition of the logistic sigmoid $\sigma(x) = \frac{1}{1+e^{-x}}$ and a function chosen from a list of spectral shaping maps. More specifically, we introduce four learnable Koopman families that act on the vector of raw spectral parameters (i) `scalar-gated`. A shared affine gate controls the entire spectrum:

$$\sigma_\phi(S_i) \;=\; \sigma(\alpha S_i + \beta), \tag{4}$$

with $\alpha, \beta \in \mathbb{R}$ learned scalars.

(ii) `per-mode gated`. Each latent direction receives its own gate,

$$\sigma_\phi(S_i) \;=\; \sigma(\alpha_i S_i + \beta_i), \tag{5}$$

allowing anisotropic amplification/decay across modes.

(iii) `MLP-shaped` spectral mapping. A small neural operator $g_\phi : \mathbb{R} \to \mathbb{R}$ transforms raw spectral parameters before squashing:

$$\sigma_\phi(S_i) \;=\; \sigma(g_\phi(S_i)), \tag{6}$$

enabling flexible nonlinear reshaping of the spectrum.

(iv) `low-rank` Koopman. The operator is restricted to rank $r \ll d$ via

$$\mathcal{K}_\phi \;=\; U_{r,\phi} \operatorname{diag}(\Sigma_{r,\phi})\, V_{r,\phi}^\top, \quad U_{r,\phi}, V_{r,\phi} \in \mathbb{R}^{d \times r}, \tag{7}$$

with orthonormal columns and a spectral vector $\Sigma_{r,\phi} \in \mathbb{R}^r$ satisfying again $|\Sigma_{r,\phi,i}| < \rho_{\max}$. It encourages low-dimensional latent dynamics and reduces complexity.

Figure 1 illustrates the overall `Learnable-DeepKoopFormer` pipeline and clarifies the interaction between the neural forecasting backbone and the learnable Koopman latent dynamics module.

In addition, for comparison purpose, we also consider a fully free operator (unconstrained baseline) obtained when $\mathcal{K}_\phi$ is a free dense matrix, without the ODO structure (2) and without the spectral bound (3). This variant maximizes expressiveness but lacks the stability and dynamical bias of the structured families above. Overall, the learnable Koopman architecture provides a continuum between rigorously constrained operator-theoretic propagators and fully flexible data-driven linear maps, allowing us to isolate and study the role of spectral structure in Transformer-based forecasting models.

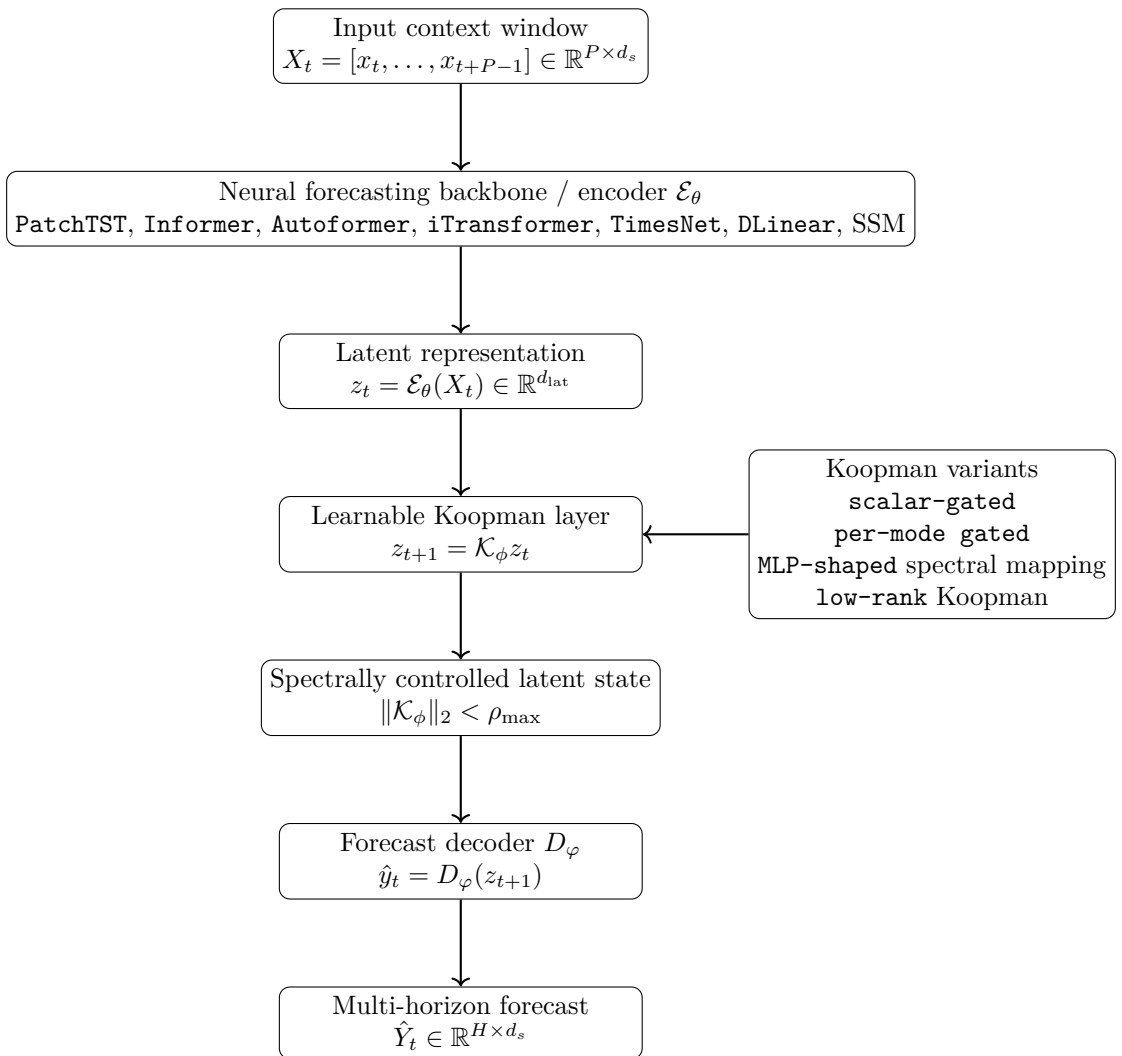

Figure 1: Overall `Learnable-DeepKoopFormer` architecture. The backbone encoder learns nonlinear temporal representations from the input window, while the learnable Koopman layer imposes a spectrally controlled linear transition in latent space. Different Koopman variants modify only the parameterization of $\mathcal{K}_\phi$; the encoder–Koopman–decoder pipeline remains unchanged.

## 3.1 Theoretical Properties of the Learnable Koopman Propagator

We consider latent states $\boldsymbol{z}_t \in \mathbb{R}^d$ evolving under a linear propagator

$$z_{t+1} = \mathcal{K}_\phi z_t, \qquad \mathcal{K}_\phi \in \mathbb{R}^{d \times d}, \tag{8}$$

where $\mathcal{K}_\phi$ is parameterized by the learnable weights $\phi$ of the Koopman module in `DeepKoopFormer`. This subsection collects several structural properties of the proposed parameterizations, focusing on stability, expressiveness, low-rank structure, and Lyapunov regularization. It is worth to highlight that since the current implementation employs direct multi-horizon decoding rather than recursive latent rollout, the theoretical stability analysis primarily characterizes the conditioning and structure of the latent transition operator rather than asymptotic forecast stability.

### 3.1.1 Spectral stability, contraction, and invertibility

The spectral bound (3) directly implies stability and contractivity of the associated linear dynamics.

**Proposition 1** (Spectral stability). *Let $\mathcal{K}_\phi$ be parameterized as in (2), with orthonormal $U_\phi, V_\phi$ and diagonal $\mathrm{diag}(\Sigma_\phi)$ satisfying (3). Then $\rho(\mathcal{K}_\phi) \leq \|\mathcal{K}_\phi\|_2 = \max_i|\Sigma_{\phi,i}| < \rho_{\max}$, where $\rho(\cdot)$ denotes the spectral radius and $\|\cdot\|_2$ the spectral norm.*

*Proof.* Because $U_\phi$ and $V_\phi$ are orthonormal, left and right multiplication do not change the spectral norm:

$$\|\mathcal{K}_\phi\|_2 = \left\|U_\phi \,\mathrm{diag}(\Sigma_\phi)\, V_\phi^\top\right\|_2 = \left\|\mathrm{diag}(\Sigma_\phi)\right\|_2. \tag{9}$$

The spectral norm of a diagonal matrix is the maximum absolute diagonal entry, hence $\|\mathcal{K}_\phi\|_2 = \max_i|\Sigma_{\phi,i}|$. For any matrix, the spectral radius is bounded above by the spectral norm, $\rho(\mathcal{K}_\phi) \leq \|\mathcal{K}_\phi\|_2$. Combining this with $|\Sigma_{\phi,i}| < \rho_{\max}$ for all $i$ yields the claim. $\square$

**Corollary 1** (Exponential contraction in latent space). *Under the assumptions of Proposition 1, for any $z_0 \in \mathbb{R}^d$ and any $n \in \mathbb{N}$, $\|\mathcal{K}_\phi^n z_0\|_2 \leq \|\mathcal{K}_\phi\|_2^n \|z_0\|_2 \leq \rho_{\max}^n \|z_0\|_2$, and hence $\lim_{n\to\infty} \mathcal{K}_\phi^n z_0 = 0$.*

*Proof.* Submultiplicativity of the spectral norm gives $\|\mathcal{K}_\phi^n\|_2 \leq \|\mathcal{K}_\phi\|_2^n$, so $\|\mathcal{K}_\phi^n z_0\|_2 \leq \|\mathcal{K}_\phi^n\|_2 \|z_0\|_2 \leq \|\mathcal{K}_\phi\|_2^n \|z_0\|_2$. Proposition 1 implies $\|\mathcal{K}_\phi\|_2 < \rho_{\max} < 1$, hence the right-hand side converges to zero as $n \to \infty$, i.e. the latent dynamics (8) is exponentially stable:

$$\|z_t\|_2 \leq \rho_{\max}^t \|z_0\|_2, \quad t \geq 0. \tag{10}$$

$\square$

In many forecasting settings, it is also natural to consider the inverse dynamics whenever the spectrum is bounded away from zero. This leads to the following simple consequence of the ODO structure.

**Proposition 2** (Invertibility and stability of the inverse). *Assume the ODO parameterization (2) satisfies the two-sided spectral bound*

$$0 < \rho_{\min} \leq |\Sigma_{\phi,i}| \leq \rho_{\max} < 1 \quad \text{for all } i. \tag{11}$$

*Then $\mathcal{K}_\phi$ is invertible with $\mathcal{K}_\phi^{-1} = V_\phi \,\mathrm{diag}\big(\Sigma_{\phi,1}^{-1}, \ldots, \Sigma_{\phi,d}^{-1}\big) U_\phi^\top$, and $\rho(\mathcal{K}_\phi^{-1}) \leq \|\mathcal{K}_\phi^{-1}\|_2 = \max_i|\Sigma_{\phi,i}^{-1}| \leq \rho_{\min}^{-1}$.*

*Proof.* Under (11), all diagonal entries of $\mathrm{diag}(\Sigma_\phi)$ are nonzero, so it is invertible. Since $U_\phi$ and $V_\phi$ are orthonormal, $\mathcal{K}_\phi$ is invertible and its inverse has the stated form. The spectral norm calculation proceeds exactly as in Proposition 1, yielding $\|\mathcal{K}_\phi^{-1}\|_2 = \max_i |\Sigma_{\phi,i}^{-1}|$, and $\rho(\mathcal{K}_\phi^{-1}) \leq \|\mathcal{K}_\phi^{-1}\|_2$. $\square$

This shows that the same spectral factors that enforce forward-time contraction can also guarantee that the inverse dynamics are well-conditioned, provided the spectrum is kept away from zero.

### 3.1.2 Expressiveness of spectrally bounded and low-rank operators

The ODO factorization parameterizes exactly the class of linearly stable operators with bounded spectral norm.

**Proposition 3** (Representation of all spectrally bounded operators). *Fix $\rho_{\max} > 0$ and define $\mathcal{K}(\rho_{\max}) := \left\{\mathcal{K} \in \mathbb{R}^{d\times d} : \|\mathcal{K}\|_2 \leq \rho_{\max}\right\}$. Then the set of matrices that admit an ODO factorization with $|\Sigma_{\phi,i}| \leq \rho_{\max}$ is precisely $\mathcal{K}(\rho_{\max})$. In particular, any $\mathcal{K}$ with $\|\mathcal{K}\|_2 < \rho_{\max}$ can be represented exactly by some choice of $(U_\phi, V_\phi, \Sigma_\phi)$.*

*Proof.* (*Surjectivity onto $\mathcal{K}(\rho_{\max})$.*) Let $\mathcal{K} \in \mathcal{K}(\rho_{\max})$. By the singular value decomposition, there exist orthonormal matrices $U, V \in \mathbb{R}^{d\times d}$ and nonnegative singular values $\Sigma_1, \ldots, \Sigma_d$ such that $\mathcal{K} = U \,\mathrm{diag}(\Sigma_1, \ldots, \Sigma_d)\, V^\top$. The spectral norm is $\|\mathcal{K}\|_2 = \max_i \Sigma_i$, so $\|\mathcal{K}\|_2 \leq \rho_{\max}$ implies $\Sigma_i \leq \rho_{\max}$ for all $i$. Thus $\mathcal{K}$ admits an ODO factorization with $U_\phi = U$, $V_\phi = V$ and $\Sigma_{\phi,i} = \Sigma_i$ satisfying $|\Sigma_{\phi,i}| \leq \rho_{\max}$.

(*Inclusion in* $\mathcal{K}(\rho_{\max})$.) Conversely, let $\mathcal{K}_\phi = U_\phi \mathrm{diag}(\Sigma_\phi) V_\phi^\top$ with $\max_i |\Sigma_{\phi,i}| \leq \rho_{\max}$. As in the proof of Proposition 1,

$$\|\mathcal{K}_\phi\|_2 = \left\|\mathrm{diag}(\Sigma_\phi)\right\|_2 = \max_i |\Sigma_{\phi,i}| \leq \rho_{\max}, \tag{12}$$

hence $\mathcal{K}_\phi \in \mathcal{K}(\rho_{\max})$. $\qquad\square$

Thus the strictly stable and learnable Koopman variants with $|\Sigma_{\phi,i}| < \rho_{\max}$ retain the full expressiveness of all linear operators in the spectrally bounded class $\mathcal{K}(\rho_{\max})$, while providing an explicit stability margin.

### 3.1.3 Low-rank structure and approximation

For the low-rank family (7), the Koopman operator is restricted to a rank–$r$ subspace of $\mathbb{R}^d$.

**Proposition 4** (Low-rank structure and norm bound)**.** *For the low-rank Koopman parameterization* (7)*: (i)* $\mathrm{rank}(\mathcal{K}_\phi) \leq r$*; (ii)* $\|\mathcal{K}_\phi\|_2 = \max_{i \leq r} |\Sigma_{r,\phi,i}|$*; (iii) if* $|\Sigma_{r,\phi,i}| < \rho_{\max}$ *for all* $i$*, then* $\rho(\mathcal{K}_\phi) \leq \|\mathcal{K}_\phi\|_2 < \rho_{\max}$*.*

*Proof.* (i) The product of a $d \times r$ matrix, an $r \times r$ diagonal matrix and an $r \times d$ matrix has rank at most $r$. (ii) Since $U_{r,\phi}$ and $V_{r,\phi}$ have orthonormal columns, they describe partial isometries from $\mathbb{R}^r$ into $\mathbb{R}^d$, and the nonzero singular values of $\mathcal{K}_\phi$ coincide with the absolute values of the entries of $\Sigma_{r,\phi}$. Hence $\|\mathcal{K}_\phi\|_2 = \max_{i \leq r} |\Sigma_{r,\phi,i}|$. (iii) The last claim follows from (ii) together with $\rho(\mathcal{K}_\phi) \leq \|\mathcal{K}_\phi\|_2$ and the bound $|\Sigma_{r,\phi,i}| < \rho_{\max}$. $\qquad\square$

Let $\mathcal{K}_\phi$ have singular values $\Sigma_{\phi,1} \geq \cdots \geq \Sigma_{\phi,d} \geq 0$, and let $\mathcal{K}_\phi^{(r)}$ denote the truncation of its singular value decomposition to rank $r$, i.e., $\mathcal{K}_\phi^{(r)}$ retains only the first $r$ singular values and vectors. The Eckart–Young theorem (see Lin (2011) for more details) states that $\mathcal{K}_\phi^{(r)}$ minimizes $\|\mathcal{K}_\phi - \tilde{\mathcal{K}}_\phi\|_F$ over all rank–$r$ matrices $\tilde{\mathcal{K}}_\phi$. Any such $\mathcal{K}_\phi^{(r)}$ admits a representation of the form (7), and thus belongs to the low-rank family. Consequently, whenever the latent Koopman dynamics are approximately low-dimensional, the low-rank variant can capture the principal modes while discarding high-rank noise.

### 3.1.4 Lyapunov regularization and energy decay

Besides spectral constraints, the training objective incorporates in (1) a Lyapunov-inspired penalty that encourages contractive behavior with respect to a fixed quadratic form. For latent pairs $(\boldsymbol{z}_t, \boldsymbol{z}_{t+1})$ the addition of the term $\mathcal{L}_{\mathrm{Lyap}} = \mathbb{E}\left[\left(\|\boldsymbol{z}_{t+1}\|_P^2 - \|\boldsymbol{z}_t\|_P^2\right)_+\right]$, $\|\boldsymbol{z}\|_P^2 = \boldsymbol{z}^\top P \boldsymbol{z}$, with $(x)_+ = \max\{x, 0\}$ and $P \succ 0$, provides a common regularizer across learnable, Koopman variants, which further biases optimisation toward strict contraction of latent energy by penalizing unstable latent growth and promoting Koopman-consistent dynamics. The following results formalize these considerations.

**Proposition 5** (Lyapunov consistency)**.** *Suppose* $P \succ 0$ *and the support of* $\boldsymbol{z}_t$ *spans* $\mathbb{R}^d$*. If* $\mathcal{L}_{\mathrm{Lyap}} = 0$*, then* $\mathcal{K}_\phi$ *satisfies the discrete Lyapunov inequality* $\mathcal{K}_\phi^\top P \mathcal{K}_\phi - P \preceq 0$*. In particular, the quadratic energy* $\|\boldsymbol{z}_t\|_P^2$ *is non-increasing under the dynamics* $\boldsymbol{z}_{t+1} = \mathcal{K}_\phi \boldsymbol{z}_t$*.*

*Proof.* The equality $\mathcal{L}_{\mathrm{Lyap}} = 0$ implies $\left(\|\boldsymbol{z}_{t+1}\|_P^2 - \|\boldsymbol{z}_t\|_P^2\right)_+ = 0$, almost surely, and thus $\|\boldsymbol{z}_{t+1}\|_P^2 \leq \|\boldsymbol{z}_t\|_P^2$, for all $\boldsymbol{z}_t$ in the support. Using $\boldsymbol{z}_{t+1} = \mathcal{K}_\phi \boldsymbol{z}_t$, $\boldsymbol{z}_t^\top \mathcal{K}_\phi^\top P \mathcal{K}_\phi \boldsymbol{z}_t \leq \boldsymbol{z}_t^\top P \boldsymbol{z}_t$, or equivalently $\boldsymbol{z}_t^\top (\mathcal{K}_\phi^\top P \mathcal{K}_\phi - P) \boldsymbol{z}_t \leq 0$, for all $\boldsymbol{z}_t$ in a set that spans $\mathbb{R}^d$. A symmetric matrix $M$ satisfies $M \preceq 0$ if and only if $\boldsymbol{z}^\top M \boldsymbol{z} \leq 0$ for all $\boldsymbol{z}$, hence $\mathcal{K}_\phi^\top P \mathcal{K}_\phi - P \preceq 0$. $\qquad\square$

Conversely, classical results in linear systems theory relate such a Lyapunov inequality to spectral stability.

**Proposition 6** (Spectral stability from a Lyapunov certificate)**.** *Let* $\mathcal{K} \in \mathbb{R}^{d \times d}$ *and suppose there exists* $P \succ 0$ *such that* $\mathcal{K}^\top P \mathcal{K} - P \prec 0$*. Then* $\rho(\mathcal{K}) < 1$*.*

*Proof.* Since $P \succ 0$, write $P = R^\top R$ for some invertible $R \in \mathbb{R}^{d \times d}$ and consider the similarity transform $\tilde{\mathcal{K}} = R\mathcal{K}R^{-1}$. The Lyapunov inequality becomes

$$\mathcal{K}^\top P\mathcal{K} - P = \mathcal{K}^\top R^\top R\mathcal{K} - R^\top R = R^\top(\tilde{\mathcal{K}}^\top\tilde{\mathcal{K}} - I)R \prec 0. \tag{13}$$

Premultiplying and postmultiplying by $(R^{-1})^\top$ and $R^{-1}$ yields $\tilde{\mathcal{K}}^\top\tilde{\mathcal{K}} - I \prec 0$, which implies $\|\tilde{\mathcal{K}}\boldsymbol{x}\|_2^2 < \|\boldsymbol{x}\|_2^2$ for all $\boldsymbol{x} \neq 0$, hence $\|\tilde{\mathcal{K}}\|_2 < 1$. Therefore $\rho(\tilde{\mathcal{K}}) < 1$, and since $\mathcal{K}$ and $\tilde{\mathcal{K}}$ are similar, $\rho(\mathcal{K}) = \rho(\tilde{\mathcal{K}}) < 1$. $\qquad\square$

For finite Lyapunov regularization weight, the penalty $\mathcal{L}_{\text{Lyap}}$ need not vanish exactly; nevertheless, Propositions 5–6 show that it biases the learned propagator toward the set of operators admitting a Lyapunov certificate of stability, in a way that complements the direct spectral constraints imposed by the ODO parameterization.

Taken together, Propositions 1–6 show that the learnable Koopman family used in `Learnable-DeepKoopFormer`: (i) provides a clear, tunable stability margin via $\rho_{\max}$ and the Lyapunov weight; (ii) retains the full expressiveness of spectrally bounded linear dynamics; (iii) admits low-rank specializations capturing dominant latent modes; and (iv) supports well-conditioned inverse propagation when the spectrum is bounded away from zero. These guarantees hold uniformly across all instantiations of `Learnable-DeepKoopFormer` built on top of the `PatchTST`, `Autoformer`, and `Informer` backbones.

The following result formalizes the role of the Koopman layer in the implemented direct-decoding architecture. Since the Koopman operator is applied only once, the result should be interpreted as a single-transition sensitivity bound, not as a finite-horizon rollout error bound.

**Proposition 7** (Single-step latent conditioning bound). *Let the forecasting model in `Learnable-DeepKoopFormer` be defined as*

$$\widehat{Y}_t(X_t) = \mathcal{D}_\varphi(\mathcal{K}_\phi\mathcal{E}_\theta(X_t)),$$

*where $\mathcal{E}_\theta$ denotes the encoder, $\mathcal{K}_\phi$ is the Koopman propagator, and $\mathcal{D}_\varphi$ is the decoder.*

*Assume that: (i) the encoder $\mathcal{E}_\theta$ is Lipschitz continuous with constant $L_\mathcal{E} > 0$, i.e.,*

$$\|\mathcal{E}_\theta(X) - \mathcal{E}_\theta(\tilde{X})\|_2 \le L_\mathcal{E}\|X - \tilde{X}\|_2,$$

*(ii) the decoder $\mathcal{D}_\varphi : \mathbb{R}^d \to \mathbb{R}^{H \cdot d}$ is Lipschitz continuous with constant $L_\mathcal{D} > 0$, and (iii) the Koopman propagator satisfies*

$$\|\mathcal{K}_\phi\|_2 \le \rho_{\max} < 1.$$

*Then for any two input histories $X_t, \tilde{X}_t$,*

$$
\begin{aligned}
\left\|\widehat{Y}_t(X_t) - \widehat{Y}_t(\tilde{X}_t)\right\|_2 &\le L_\mathcal{D}\|\mathcal{K}_\phi\|_2\|\mathcal{E}_\theta(X_t) - \mathcal{E}_\theta(\tilde{X}_t)\|_2 \\
&\le L_\mathcal{D}L_\mathcal{E}\|\mathcal{K}_\phi\|_2\|X_t - \tilde{X}_t\|_2 \\
&\le L_\mathcal{D}L_\mathcal{E}\rho_{\max}\|X_t - \tilde{X}_t\|_2.
\end{aligned}
$$

*Thus, spectral control of $\mathcal{K}_\phi$ bounds the instantaneous amplification of encoder-level perturbations through the single Koopman transition before decoding.*

*Proof.* Starting from the implemented model,

$$\widehat{Y}_t(X_t) = \mathcal{D}_\varphi(\mathcal{K}_\phi\mathcal{E}_\theta(X_t)).$$

Therefore,

$$\left\|\widehat{Y}_t(X_t) - \widehat{Y}_t(\tilde{X}_t)\right\|_2 = \left\|\mathcal{D}_\varphi(\mathcal{K}_\phi\mathcal{E}_\theta(X_t)) - \mathcal{D}_\varphi(\mathcal{K}_\phi\mathcal{E}_\theta(\tilde{X}_t))\right\|_2.$$

By Lipschitz continuity of the decoder,

$$\left\|\widehat{Y}_t(X_t) - \widehat{Y}_t(\tilde{X}_t)\right\|_2 \le L_\mathcal{D}\left\|\mathcal{K}_\phi(\mathcal{E}_\theta(X_t) - \mathcal{E}_\theta(\tilde{X}_t))\right\|_2.$$

Using the induced spectral norm inequality,

$$\left\|\widehat{Y}_t(X_t) - \widehat{Y}_t(\tilde{X}_t)\right\|_2 \le L_{\mathcal{D}}\|\mathcal{K}_\phi\|_2\|\mathcal{E}_\theta(X_t) - \mathcal{E}_\theta(\tilde{X}_t)\|_2.$$

Applying the Lipschitz property of the encoder gives

$$\left\|\widehat{Y}_t(X_t) - \widehat{Y}_t(\tilde{X}_t)\right\|_2 \le L_{\mathcal{D}}L_{\mathcal{E}}\|\mathcal{K}_\phi\|_2\|X_t - \tilde{X}_t\|_2.$$

Finally, since $\|\mathcal{K}_\phi\|_2 \le \rho_{\max} < 1$, we obtain

$$\left\|\widehat{Y}_t(X_t) - \widehat{Y}_t(\tilde{X}_t)\right\|_2 \le L_{\mathcal{D}}L_{\mathcal{E}}\rho_{\max}\|X_t - \tilde{X}_t\|_2.$$

This proves the claim. $\qquad\square$

It is worth to highlight that in the implemented architecture, the decoder may dominate the overall Lipschitz constant of the forecasting map; therefore, the role of the Koopman layer is not to fully determine end-to-end robustness, but to constrain and regularize the latent transition before decoding.

**Remark 1** (Role of the Koopman transition in the `Learnable-DeepKoopFormer` architecture). *In the* `Learnable-DeepKoopFormer` *architecture, the Koopman operator is applied once to the encoded context representation, and the resulting latent vector is decoded directly into the full $H$-step forecast. That is, inference uses*

$$\widehat{Y}_t(X_t) = \mathcal{D}_\varphi(\mathcal{K}_\phi\mathcal{E}_\theta(X_t)),$$

*rather than an autoregressive latent rollout of the form*

$$\widehat{Y}_t^{(h)}(X_t) = \mathcal{D}_\varphi\big(\mathcal{K}_\phi^h\mathcal{E}_\theta(X_t)\big).$$

*Therefore, Proposition 7 should be interpreted as a perturbation-sensitivity bound for the associated recursive latent-propagation model, or as an operator-level stability result motivating spectral control of $\mathcal{K}_\phi$, rather than as an end-to-end finite-horizon forecast-error guarantee for the direct-decoding implementation.*

*Direct multi-horizon decoding avoids error accumulation from autoregressive rollout, keeps the forecasting architecture aligned with modern direct forecasting backbones, and allows the decoder to learn horizon-specific output structure from a Koopman-stabilized latent representation. Consequently, the Koopman layer should be interpreted as imposing a spectrally controlled latent bottleneck that regularizes and conditions the representation passed to the forecast head, rather than as the sole mechanism responsible for evolving the latent state across every forecast step. Spectral constraints and Lyapunov regularization therefore limit perturbation amplification through this single transition, while the encoder and decoder retain responsibility for extracting temporal context and mapping the stabilized latent state to the complete forecast horizon.*

*In the implemented architecture, the Koopman operator is applied only once and the decoder directly outputs the complete prediction horizon. Therefore, the geometric factor $\rho_{\max}^h$ associated with recursive latent propagation does not apply to the direct-decoding model.*

*The proposition instead shows that the spectrally constrained Koopman layer bounds the instantaneous amplification of latent perturbations before decoding. A model without the Koopman layer satisfies the standard bound*

$$\left\|\widehat{Y}_t(X_t) - \widehat{Y}_t(\tilde{X}_t)\right\|_2 \le L_{\mathcal{D}}L_{\mathcal{E}}\|\mathcal{K}_\phi\|_2\|X_t - \tilde{X}_t\|_2$$
$$\le L_{\mathcal{D}}L_{\mathcal{E}}\rho_{\max}\|X_t - \tilde{X}_t\|_2.$$

*whereas the Koopman-augmented model inserts the factor $\|\mathcal{K}_\phi\|_2 \le \rho_{\max}$. Since the decoder is trained jointly, it may partially compensate for this contraction. Thus, the proposition should be viewed as an operator-level conditioning result motivating spectral control of the latent transition, rather than as a proof of a strict robustness advantage or uniformly improved forecasting accuracy.*

**Remark 2** (Why stability matters in the single-step Koopman setting). *Although the Koopman operator is applied only once in the implemented architecture, spectral stability and Lyapunov regularization remain practically relevant. First, the Koopman layer acts as a structured linear mixing operator across latent*

*dimensions, enabling the model to capture cross-channel correlations and coupling between variables through the learned operator $\mathcal{K}_\phi$. Unlike a purely diagonal or element-wise transformation, $\mathcal{K}_\phi$ introduces a globally coupled latent transition, allowing interactions between features to be explicitly modeled in a controlled linear subspace.*

*Second, even in the single-step setting, the spectral norm $\|\mathcal{K}_\phi\|$ directly governs the amplification of perturbations through the encoder–Koopman–decoder pipeline. In particular, stability constraints ($\|\mathcal{K}_\phi\| < 1$) ensure that the latent representation passed to the decoder is contractive and well-conditioned, preventing the decoder from receiving unstable or highly amplified features. This may improve robustness to noise and distribution shifts by reducing amplification of latent perturbations before decoding*

*Third, Lyapunov regularization further biases the learned representation toward energy decay in latent space, effectively acting as a regularizer on the geometry of the representation rather than on long-horizon rollout. From this perspective, the Koopman operator should be understood not only as a temporal propagator but also as a spectrally controlled latent mixing layer that stabilizes representations and encodes structured dependencies across variables. While repeated Koopman rollouts are theoretically meaningful, in the present architecture stability primarily improves conditioning, robustness, and cross-variable coupling in the single-step latent transition.*

**Interpretation and assumptions of Proposition 7.** Proposition 7 provides a perturbation-sensitivity bound under standard Lipschitz continuity assumptions on the encoder $\mathcal{E}_\theta$ and decoder $\mathcal{D}_\varphi$. In practice, these assumptions are not restrictive: neural networks composed of linear layers, attention blocks, and pointwise nonlinearities (e.g., GELU, ReLU, tanh) are Lipschitz continuous, with constants determined by the product of operator norms of their constituent layers. While these constants are not explicitly controlled during training, common design elements such as normalization layers (e.g., LayerNorm), bounded activations, and implicit regularization from optimization tend to prevent uncontrolled growth of these norms.

Importantly, Proposition 7 should be interpreted as an operator-level stability result rather than a tight end-to-end generalization bound. In the implemented `Learnable-DeepKoopFormer` architecture, the Koopman operator is applied only once, and the forecast is obtained via direct decoding

$$\widehat{Y}_t = \mathcal{D}_\varphi(\mathcal{K}_\phi \mathcal{E}_\theta(X_t)).$$

Therefore, the relevant case corresponds to $h = 1$, yielding the bound

$$\left\|\widehat{Y}_t(X_t) - \widehat{Y}_t(\tilde{X}_t)\right\|_2 \ \leq \ L_\mathcal{D} L_\mathcal{E} \|\mathcal{K}_\phi\|_2 \|X_t - \tilde{X}_t\|_2.$$

In this setting, spectral control of $\mathcal{K}_\phi$ does not govern long-horizon rollout error, but instead limits the amplification of perturbations through the single latent transition before decoding. Thus, the practical role of the Koopman operator is to provide a spectrally conditioned linear transformation that improves robustness and stabilizes the latent representation passed to the decoder. The proposition should therefore be understood as a justification for controlling $\|\mathcal{K}_\phi\|_2$, rather than as a direct guarantee on multi-step forecasting accuracy.

It shows that spectral control of the Koopman propagator yields direct control over the sensitivity of forecasts to perturbations in the input history. In particular, whenever $\|\mathcal{K}_\phi\|_2 < 1$, perturbations propagate through the encoder–Koopman–decoder pipeline with a bounded amplification factor.Importantly, the implemented `Learnable-DeepKoopFormer` architecture does not recursively propagate latent states across the prediction horizon. Therefore, the geometric decay factor associated with repeated Koopman rollout does not directly apply to the deployed forecasting model. In the implemented direct-decoding setting, the practical role of spectral control is instead to regularize and condition the latent representation passed to the decoder by limiting instantaneous perturbation amplification through the single Koopman transition.

### 3.1.5 Theoretical stability advantage of the Koopman layer

We compare the latent dynamics induced by the proposed Koopman layer with those of an unconstrained linear state–space model (SSM). We have shown how all `Learnable-DeepKoopFormer` variants propagate

a latent state $z_t \in \mathbb{R}^d$ according to $z_{t+1} = \mathcal{K}_\phi z_t$, where $\mathcal{K}_\phi$ is parameterised via an orthogonal–diagonal–orthogonal (ODO) factorisation, $\mathcal{K}_\phi = U_\phi \operatorname{diag}(\Sigma_\phi) V_\phi^\top$. Here $U_\phi, V_\phi \in \mathbb{R}^{d \times r}$ have orthonormal columns, obtained through QR retraction, and the spectral coefficients $\Sigma_\phi \in \mathbb{R}^r$ are generated from unconstrained raw parameters $S$ via a componentwise squashing map $\Sigma_{i,\phi}(\theta) = \rho_{\max} \sigma(S_i)$, $\quad \sigma : \mathbb{R} \to (0,1)$, $\quad 0 < \rho_{\max} < 1$. In contrast, the SSM baseline evolves a hidden state $h_t \in \mathbb{R}^{d_h}$ as

$$h_{t+1} = Ah_t + Bx_t, \qquad \hat{y}_t = Ch_t, \tag{14}$$

where $A, B, C$ are completely unconstrained. In particular, no analogous spectral and contraction constraints to the ones for $\mathcal{K}_\phi$ are imposed on the SSM transition matrix $A$ in (14). In the following Theorem we formally address core stability advantage of the Koopman parameterisation.

**Theorem 1.** *Let $\mathcal{K}_\phi(\rho_{\max})$ denote the class of Koopman operators realised by the ODO-based spectral–squashing parameterisation, such that $\|\mathcal{K}_\phi\|_2 < \rho_{\max} < 1$ for all $\mathcal{K}_\phi \in \mathcal{K}_\phi(\rho_{\max})$. Let $\mathcal{A}$ denote the class of unconstrained state–space transition matrices. Then: (i) For every $\mathcal{K}_\phi \in \mathcal{K}_\phi(\rho_{\max})$, the latent dynamics $z_{t+1} = \mathcal{K}_\phi z_t$ are uniformly exponentially stable at all training iterates. (ii) The class $\mathcal{A}$ contains matrices with arbitrarily large spectral radius; moreover, for any finite prediction horizon, there exist unstable $A \in \mathcal{A}$ that achieve the same finite-horizon training loss as a stable model.*

*Proof. (i) Stability of the Koopman parameterisation*

By Proposition 1 and Corollary 1, for all $\mathcal{K}_\phi \in \mathcal{K}_\phi(\rho_{\max})$ the latent dynamics satisfy (see (10)): $\|z_t\|_2 \le \rho_{\max}^t \|z_0\|_2$, which proves uniform exponential stability.

*(ii) Unstable but loss-equivalent SSM solutions* The unconstrained class $\mathcal{A}$ contains matrices with arbitrarily large spectral radius: for any $\gamma > 0$, $A = \gamma I$ satisfies $\rho(A) = \gamma$. Now consider a stable SSM $h_{t+1} = Ah_t + Bx_t$, $\hat{y}_t = Ch_t$ that achieves a given finite-horizon loss over $t = 0, \dots, H$. Define an augmented state $\tilde{h}_t = [h_t^\top, w_t^\top]^\top$ with dynamics

$$\tilde{h}_{t+1} = \begin{bmatrix} A & 0 \\ 0 & \gamma I \end{bmatrix} \tilde{h}_t + \begin{bmatrix} B \\ 0 \end{bmatrix} x_t, \qquad \hat{y}_t = \begin{bmatrix} C & 0 \end{bmatrix} \tilde{h}_t, \tag{15}$$

where $\gamma > 1$ and $\tilde{h}_0 = [h_0^\top, 0^\top]^\top$. Then $w_t \equiv 0$ for all $t \le H$, so the augmented model produces identical outputs and achieves the same finite-horizon loss, while its transition matrix has spectral radius $\gamma > 1$. Choosing $\gamma$ arbitrarily large completes the proof. $\square$

It is worth to highlight that, the above results concern the intrinsic stability properties of the latent transition operator itself and should not be interpreted as finite-horizon rollout guarantees for the implemented direct-decoding forecasting architecture.

### 3.1.6 Invertibility in finite dimensions and Banach space extension

Let $(\mathcal{X}, \|\cdot\|)$ be a Banach space and let $T : \mathcal{X} \to \mathcal{X}$ denote the (possibly nonlinear) time-1 evolution map of an underlying dynamical system. The Koopman operator acting on observables $J : \mathcal{X} \to \mathbb{R}$ is $(\mathcal{K}_\phi J)(x) = J(T(x))$, which is linear even if $T$ is nonlinear. If $T$ is bijective, then $\mathcal{K}_\phi$ is invertible with $\mathcal{K}_\phi^{-1} J = J \circ T^{-1}$. Thus invertibility of the latent propagator corresponds to bijectivity of the underlying state update. In `DeepKoopFormer` we learn a *finite-dimensional approximation* of $\mathcal{K}_\phi$ using an ODO-structured linear operator $\mathcal{K}_\phi = U_\phi \operatorname{diag}(\Sigma_\phi) V_\phi^\top$, $\quad U_\phi, V_\phi \in O(d)$, $\quad \Sigma_i \in (0, \rho_{\max})$.

Because $\Sigma_{i,\phi} > 0$ for all $i$, the operator $\mathcal{K}_\phi$ is invertible and belongs to

$$\mathrm{GL}(d) := \{M \in \mathbb{R}^{d \times d} \mid \det(M) \ne 0\}.$$

Its inverse is

$$\mathcal{K}_\phi^{-1} = V_\phi \operatorname{diag}(\Sigma_\phi^{-1}) U_\phi^\top,$$

and

$$\|\mathcal{K}_\phi\|_2 \le \rho_{\max}, \qquad \|\mathcal{K}_\phi^{-1}\|_2 = \frac{1}{\sigma_{\min}(\mathcal{K}_\phi)}.$$

Since $\sigma_{\min}(\mathcal{K}_\phi) < \rho_{\max}$, one has

$$\frac{1}{\sigma_{\min}(\mathcal{K}_\phi)} > \frac{1}{\rho_{\max}},$$

so the inverse need not be contractive.

Thus $\mathcal{K}_\phi^n \to 0$ as $n \to \infty$, while backward iterates $\mathcal{K}_\phi^{-n}$ are well-defined but generally grow in norm unless an additional lower spectral bound $\Sigma_{i,\phi} \geq \rho_{\min} > 0$ is imposed.

In contrast, standard SSMs with unconstrained transition matrices $A$—common in classical and deep state-space architectures—provide no explicit spectral guarantees: $A$ may be singular, ill-conditioned, or unstable, and even when invertible its spectrum is uncontrolled Gu et al. (2021a); Smith et al. (2022); Hamilton (2020). By comparison, the Koopman propagator in `Learnable-DeepKoopFormer` ensures stable, invertible latent dynamics with explicit spectral bounds and well-conditioned evolution.

## 4    Numerical Simulations

In this section, we present a comprehensive set of numerical simulations designed to evaluate the proposed `Learnable-DeepKoopFormer` framework across multiple datasets, forecasting settings, and architectural configurations. The experiments are organized into several complementary scenarios. We first conduct controlled ablation studies to isolate the effect of Koopman-based latent dynamics, spectral stability constraints, and Lyapunov regularization. We then compare plain forecasting backbones with their Koopman-enhanced counterparts across different Transformer and non-Transformer architectures. Additional simulations investigate the influence of learnable spectral parameterizations, low-rank Koopman structures, and latent stability properties on forecasting accuracy and robustness. Finally, we provide spectral analyses of the learned Koopman operators, including learned spectral-coefficient distributions, conditioning behavior, and analyses of how different spectral parameterizations shape the latent dynamics under explicit stability constraints across prediction horizons.

Together, these experiments aim to provide both predictive and dynamical insight into the role of structured Koopman latent evolution in modern time-series forecasting architectures.

**Benchmark Datasets**    To assess the forecasting performance of `Learnable-DeepKoopFormer`, we consider three heterogeneous and dynamically rich domains: *climate systems*, *financial markets*, and *electricity generation*. These settings exhibit nonstationarity, multiscale dependencies, and nonlinear interactions, challenging standard deep forecasting models. For climate analysis, we use CMIP6 atmospheric data[4] covering regional Wind Speed and surface pressure over Germany Makula & Zhou (2022); Hersbach et al. (2020); Forootani et al. (2025). As a nonlinear financial benchmark, we analyse a public multivariate Cryptocurrency dataset[5] including volatility, price, and trading indicators. Finally, we model Spanish national electricity generation[6] across fossil, wind, solar, hydro, and biomass sources, where physical and weather-driven couplings shape temporal dynamics. These datasets span chaotic, stochastic, and structured regimes, providing a rigorous testbed for evaluating Koopman-enhanced latent representations in nonlinear time-series forecasting.

The selected datasets were chosen to span multiple dynamical regimes and application domains commonly encountered in modern forecasting problems. In particular, the CMIP6 climate dataset exhibits high-dimensional chaotic and multiscale dynamics, the Cryptocurrency dataset represents highly stochastic and nonstationary financial behavior, and the electricity generation dataset contains structured seasonal and physically coupled temporal patterns. Together, these datasets provide complementary benchmarks for evaluating robustness, stability, and generalization of Koopman-enhanced forecasting models under heterogeneous temporal dynamics.

To improve clarity and reproducibility, we summarize the key characteristics and preprocessing steps for all datasets in Table 1. For each dataset, we report the number of samples, number of variables, temporal resolution, preprocessing pipeline, and train/test split protocol. Unless otherwise stated, all datasets are

---

[4]https://cds.climate.copernicus.eu/datasets/projections-cordex-domains-single-levels?tab=overview
[5]https://github.com/Chisomnwa/Cryptocurrency-Data-Analysis
[6]https://github.com/afshinfaramarzi/Energy-Demand-electricity-price-Forecasting/tree/main

Table 1: Summary of datasets used in the experiments.

| Dataset | Samples | Variables | Resolution | Preprocessing | Split |
|---|---|---|---|---|---|
| CMIP6 (Wind/Pressure) | 2928 | 6 (from 28,574) | 3-hour | Channel selection, MinMax scaling, sliding windows | 80/20 (chronological) |
| Cryptocurrency | 2900 | 2 | Daily | MinMax scaling, sliding windows | 80/20 (chronological) |
| Energy (Electricity) | 3500 | 6 | Hourly | MinMax scaling, sliding windows | 80/20 (chronological) |

normalized using a `MinMax` scaler prior to training, and sliding-window samples are constructed using a context length (P) and prediction horizon (H). No additional preprocessing steps, such as standardization, detrending, or feature engineering, are applied. For high-dimensional datasets, we restrict the number of input channels to maintain a controlled and comparable experimental setting across all models.

For the CMIP6 dataset, the original data consists of high-dimensional spatial fields with 28,574 variables per timestep. To ensure tractability and fair comparison across models, we select a subset of 6 representative channels. All datasets are truncated to a maximum of 2,500–3,500 samples depending on availability, and transformed into supervised learning format using sliding windows. No additional filtering, detrending, or subsampling beyond channel selection and normalization is applied.

**Hardware Configuration**   All experiments were executed on a high-performance computing (`HPC`) system with dual `AMD EPYC 9554` processors (128 cores per node, multithreaded) and 1.5 TB shared memory, suitable for large context windows and attention-based models. Training was accelerated using `NVIDIA L40S GPUs` supporting mixed-precision tensor operations.

**Code availability statement**   The `Learnable-DeepKoopFormer` framework—including source code, datasets, and representative figures—is accessible at is accessible at `Zenodo` [7]. The entire implementation is written in `Python` and builds upon standard scientific computing libraries, including `PyTorch`, `NumPy`, and `SciPy`.

**Model hyperparameters**   To ensure fair comparison, model capacity is fixed across experiments. `PatchTST`, `Informer`, `iTransformer`, and `TimesNet` use $L = 3$ encoder/block layers with $d_{\text{model}} = 96$, $h = 4$ attention heads where applicable, and $d_{\text{ff}} = 96$. `Autoformer` uses the same depth and heads with $d_{\text{model}} = 96$ and $d_{\text{ff}} = 64$. Patch size for `PatchTST` and `Informer`, the moving-average window for `Autoformer`, and the input length for the remaining Koopman-compatible backbones are tied to the history length $P$. Token representations are pooled into a single latent vector. For `iTransformer`, temporal tokens are embedded with a linear projection followed by Transformer encoding and average pooling. For `TimesNet`, temporal representations are obtained using $L = 3$ lightweight temporal convolution blocks followed by adaptive average pooling. For constrained and learnable Koopman regimes, the latent dimension is $d_{\text{lat}} = d_{\text{model}}$ for `PatchTST`, `Informer`, `iTransformer`, `TimesNet`, `DLinear`, and SSM backbones, and $d_{\text{lat}} = H$ for `Autoformer`. Stable and learnable variants enforce $\rho(\mathcal{K}_\phi) < \rho_{\max}$ with $\rho_{\max} = 0.99$, while the unconstrained case uses a free matrix in $\mathbb{R}^{d_{\text{lat}} \times d_{\text{lat}}}$. Within `Learnable-DeepKoopFormer`, `scalar-gated` and `per-mode gated` use global and per-mode $(\alpha, \beta)$ gates; `MLP-shaped` spectral mapping employs a one-hidden-layer MLP for spectral shaping; and `low-rank` Koopman uses rank $r = 16$, $\mathcal{K}_\phi = U_{r,\phi} \operatorname{diag}(\Sigma_{r,\phi}) V_{r,\phi}^\top$, with $U_r, V_r \in \mathbb{R}^{d_{\text{lat}} \times 16}$. Among baselines, LSTM uses $L = 2$ layers with hidden size 96; `DLinear` applies a shared linear map from the length-$P$ history to the $H$-step forecast; and the SSM employs a 96-dimensional latent state with linear readout to $\mathbb{R}^{H \cdot d}$. All models are trained with Adam (learning rate $3 \times 10^{-4}$) for 4,000 steps per $(P, H)$ configuration; the Lyapunov weight is $\lambda_{\text{Lyap}} = 0.1$ for Koopman variants unless stated otherwise.

## 4.1   Scenario I: Ablation study to isolate the impact of Koopman-based latent dynamics

This subsection presents a controlled ablation analysis designed to clarify the role of Koopman-based latent dynamics in the proposed forecasting framework. We examine two complementary settings: first, we isolate the effects of latent propagation, spectral stability, and Lyapunov regularisation using a fixed

---

[7]https://doi.org/10.5281/zenodo.17988424, https://doi.org/10.5281/zenodo.18115612, https://doi.org/10.5281/zenodo.18491274

`PatchTST` backbone on the pressure surfce dataset; second, we compare plain forecasting backbones with their Koopman-enhanced counterparts on the Wind Speed dataset. Together, these experiments allow us to distinguish improvements due to backbone representation learning from those arising from explicit latent dynamical evolution.

### 4.1.1 Spectral stability and Lyapunov regularisation influence

To demonstrate the potential of `DeepKoopFormer`, we perform a controlled ablation study designed to isolate the impact of Koopman-based latent dynamics. By keeping the `PatchTST` encoder fixed and varying only the latent propagator and its associated constraints, we explicitly evaluate how spectral stability and Lyapunov regularisation influence forecasting performance.

This allows us to distinguish between improvements due to the forecasting backbone, improvements due to adding a latent linear transition, and improvements specifically attributable to the spectrally constrained Koopman parameterisation.

We compare the following six models:

- `DLinear` Zeng et al. (2023): a lightweight channel-wise linear forecasting baseline applied directly to the input window. This model does not use attention or an explicit latent propagator.

- `LSTM`: a two-layer recurrent baseline with hidden dimension $h = 96$, chosen to provide a recurrent comparison with a latent width comparable to the Transformer-based models.

- `PatchTST` (*plain Transformer backbone*): a `PatchTST` encoder followed directly by a linear forecasting head. This is the unaugmented backbone and contains no latent propagation layer.

- `PatchTST`–`LinearLatent` (*unconstrained latent linear layer*): the same `PatchTST` encoder followed by an unconstrained latent transition $z_{t+1} = W z_t$ and a linear decoder. This ablation tests whether a generic latent linear layer already explains part of the empirical gain, without imposing Koopman structure, spectral constraints, or Lyapunov regularisation.

- `DeepKoopFormer`-`PatchTST` (No Lyap) (*spectrally constrained Koopman without Lyapunov regularisation*): the `PatchTST` encoder followed by an ODO-parameterised Koopman operator $\mathcal{K}_\phi$ satisfying $\rho(\mathcal{K}) < 0.99$, with the Lyapunov penalty disabled, i.e. $\lambda_{\mathrm{Lyap}} = 0$.

- `DeepKoopFormer`-`PatchTST` (With Lyap) (*full DeepKoopFormer variant*): the same ODO-parameterised Koopman operator with $\rho(\mathcal{K}_\phi) < 0.99$, together with the Lyapunov-inspired regulariser using $\lambda_{\mathrm{Lyap}} = 0.1$.

All `PatchTST`-based models use the same encoder architecture: a one-dimensional patch embedding layer followed by a Transformer encoder with three layers, four attention heads, model width $d_{\mathrm{model}} = d_{\mathrm{latent}} = 96$, and feedforward dimension $d_{\mathrm{ff}} = 96$. The `LSTM` baseline uses two recurrent layers with hidden size 96, providing a comparable latent dimensionality. The `DLinear` model is intentionally smaller and serves as a simple linear reference baseline.

To keep the comparison controlled, all models are trained and evaluated using the same chronological split of the time series, with the first 80% used for training and the remaining 20% used for testing. We restrict the inputs to a fixed set of five channels. The features are scaled with a `MinMax` transform fitted on the training portion only, and the same transformation is then applied to the test portion. For each patch length $p$ and forecast horizon $H$, we construct a sliding-window forecasting dataset with context length equal to the patch length. We sweep

$$P \in \{70, 80, 90, 100, 110, 120, 130\}, \qquad H \in \{2, 4, 6, 8, 10, 12, 14, 16\}.$$

All models are trained with the Adam optimiser using learning rate $3 \times 10^{-4}$ for a fixed budget of 4000 epochs, without early stopping. The primary training objective is the mean squared forecasting error on the

| Model | Latent propagator | Lyapunov term |
|---|---|---|
| PatchTST | none (direct linear head) | – |
| PatchTST– LinearLatent | unconstrained linear map $\boldsymbol{z}_{t+1} = W\boldsymbol{z}_t$ | $\lambda_{\text{Lyap}} = 0$ |
| DeepKoopFormer-PatchTST (No Lyap) | ODO Koopman, $\rho(K) < 0.99$ | $\lambda_{\text{Lyap}} = 0$ |
| DeepKoopFormer-PatchTST (With Lyap) | ODO Koopman, $\rho(K) < 0.99$ | $\lambda_{\text{Lyap}} = 0.1$ |

Table 2: PatchTST-based ablation configurations used to isolate the effect of latent propagation, spectral Koopman constraints, and Lyapunov regularisation.

normalised scale. For Koopman-augmented variants, the total loss is

$$\mathcal{L}_{\text{total}} = \mathcal{L}_{\text{forecast}} + \lambda_{\text{Lyap}}\mathcal{L}_{\text{Lyap}}, \tag{16}$$

where $\mathcal{L}_{\text{Lyap}}$ is the Lyapunov-inspired penalty. We set $\lambda_{\text{Lyap}} = 0.1$ only for KoopPatchTST_Full; all other variants use $\lambda_{\text{Lyap}} = 0$. For every model and every $(p, H)$ configuration, we report mean squared error (MSE) and mean absolute error (MAE). Table 2 summarises the ablation configurations.

Figure 2 summarizes the train–test error distributions for the pressure dataset. Across all models, the training errors are substantially smaller and more concentrated than the test errors, especially at longer forecast horizons. Among the PatchTST-based variants, the Koopman-augmented models remain broadly competitive with the direct PatchTST baseline and the unconstrained latent-layer variant. The full DeepKoopFormer variant with Lyapunov regularisation often achieves low training error, but it does not uniformly reduce the test error relative to the no-Lyapunov and unconstrained variants. This suggests that the Koopman constraint and Lyapunov penalty improve the structure and stability of the latent transition, but their predictive benefit is dataset- and horizon-dependent. The DLinear baseline shows larger errors overall, while the LSTM exhibits the widest test-error spread, particularly in MAE and MSE at longer horizons.

Overall, these results support that Koopman augmentation provides a stable and competitive latent-dynamics mechanism compare to the other baselines. We emphasize that the empirical forecasting improvements introduced by the Koopman layer are generally modest and not uniform across all configurations. In several settings, the backbone architecture itself remains the dominant factor determining predictive performance. The primary role of the Koopman module is therefore better interpreted as imposing structured and spectrally controlled latent dynamics, rather than consistently improving forecasting accuracy. Similarly, the Lyapunov regularizer does not consistently reduce test error, although it biases optimization toward contractive latent dynamics and improved spectral conditioning.

To analyze the forecasting performance across different configurations, we visualize the error metrics (MSE and MAE) as functions of the prediction horizon for multiple architectures and patch lengths in Figs. 3 and 4. Specifically, for each dataset split (Train/Test), we group the results by patch length, horizon, and model architecture, and compute the average error over all runs. The resulting plots are organized as a single row of subplots, where each subplot corresponds to a fixed patch length, enabling direct comparison across scales of temporal context. Within each subplot, curves represent different architectures, distinguished by both color and marker style to enhance visual clarity. The x-axis denotes the prediction horizon, while the y-axis shows the corresponding error metric. This allows to simultaneously assess (i) how prediction error grows with horizon, (ii) how performance varies across architectures, and (iii) how the choice of patch length influences forecasting accuracy.

### 4.1.2 Plain Backbones versus Koopman-Enhanced Forecasting Models

In this section, we systematically compare each forecasting backbone in its plain form against its Koopman-enhanced counterpart. The plain models use the backbone representation directly for multi-step prediction, while the combined Koopformer variants insert a Koopman operator between the learned latent representation and the forecasting head. This design allows us to isolate the contribution of Koopman-based latent dynamics from the representational capacity of the backbone itself. By evaluating PatchTST, Autoformer, Informer, iTransformer, and TimesNet-style architectures under identical input lengths and forecast hori-

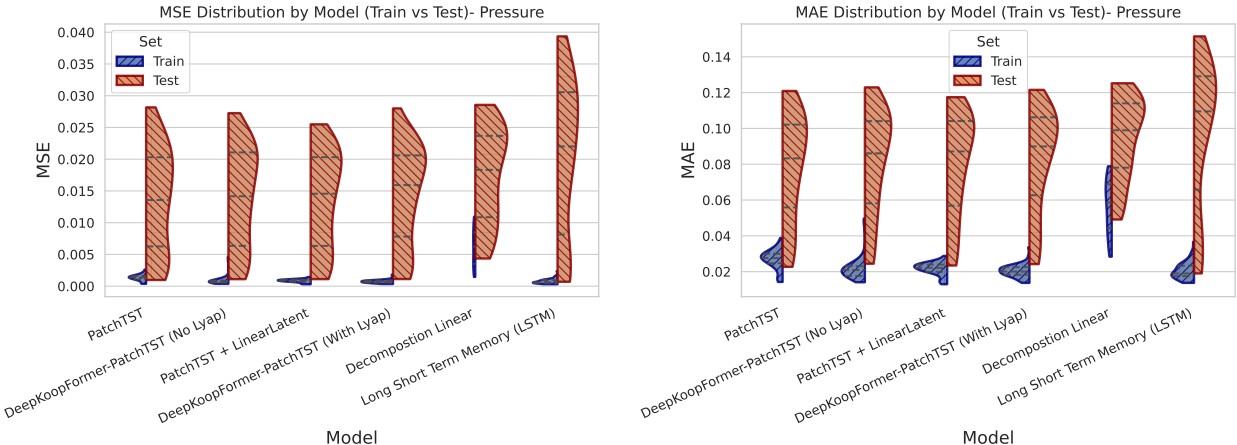

Figure 2: **PatchTST**-based ablation configurations used to isolate the effect of latent propagation, spectral Koopman constraints, and Lyapunov regularisation 4.1.1.

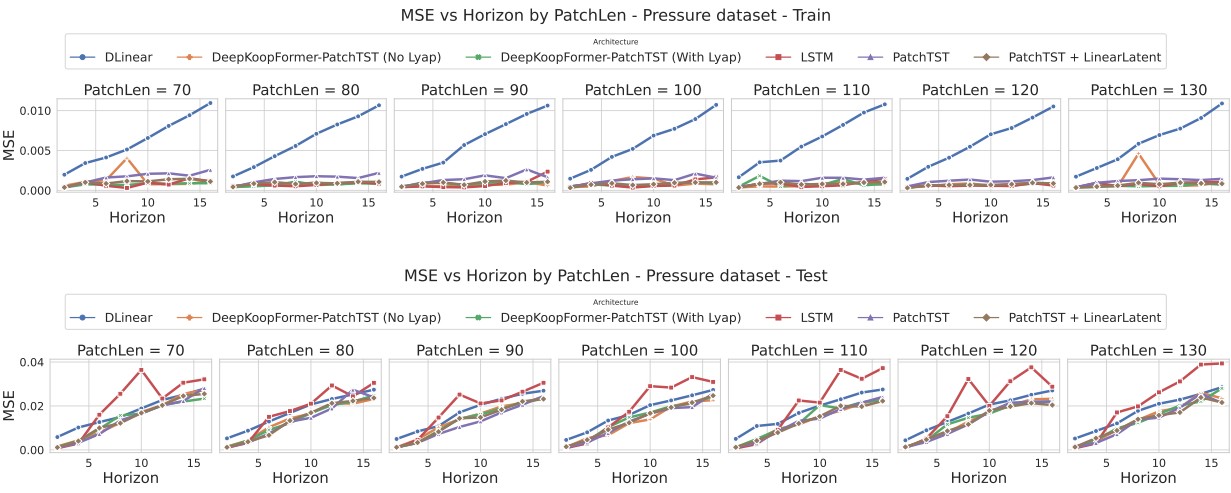

Figure 3: MSE vs. prediction horizon across different patch lengths for multiple architectures. Each subplot corresponds to a fixed patch length, while curves represent different models. Results show that prediction error generally increases with horizon, with variations in robustness across architectures in subsection 4.1.1

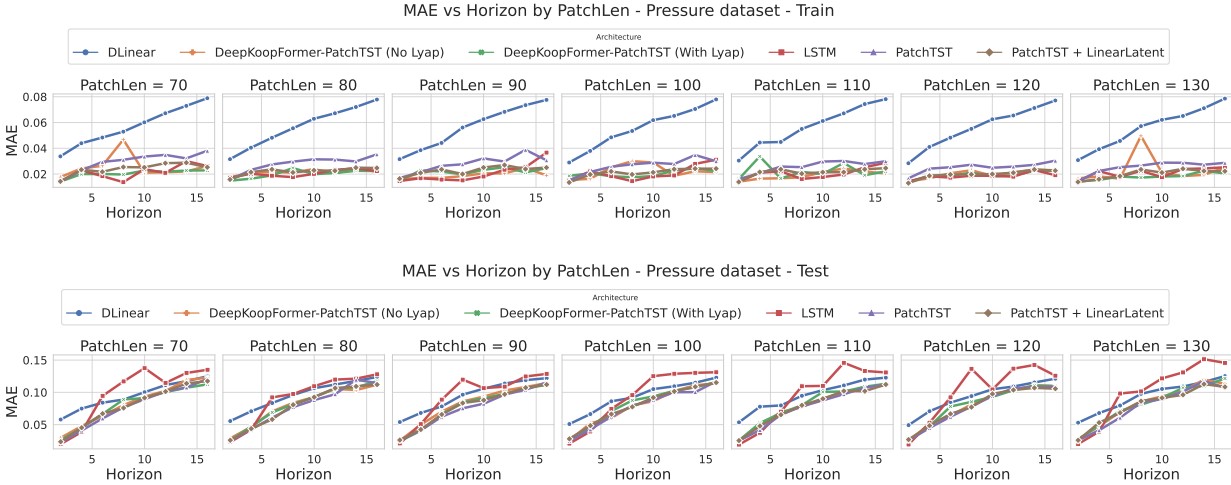

Figure 4: MAE vs. prediction horizon across different patch lengths for multiple architectures. Each subplot corresponds to a fixed patch length, with curves indicating different models. The results highlight how model accuracy degrades with increasing horizon and how patch length influences performance in subsection 4.1.1.

zons, we assess whether explicit linear evolution in latent space improves forecasting accuracy, stability, and generalization compared with standard backbone-only predictors.

We evaluate the proposed framework on the CMIP6 Wind Speed forecasting dataset to investigate the effectiveness of Koopman-enhanced latent dynamics under practical multivariate temporal prediction settings. In the simulation, sliding windows of length $P$ are paired with forecast targets of length $H$, over a grid $P \in \{80, 100, 120, 140\}$ and $H \in \{4, 8, 12, 16\}$. Inputs are feature-wise normalised and split chronologically into 80%/20% train/test sets.

The Wind Speed forecasting experiments reveal a consistent distinction between optimization performance and generalization behavior across the considered backbone architectures. In nearly all cases, the Koopman-enhanced variants achieve lower training MSE and MAE than their plain backbone counterparts, indicating that the additional latent dynamical operator substantially improves optimization efficiency and temporal fitting capability.

This effect is especially visible for `Informer`, `PatchTST`, and `iTransformer`, where most Koopman variants shift the training-error distributions downward relative to the plain backbone. The strongest reductions are often obtained by the more flexible Koopman formulations, such as the unconstrained and `low-rank` Koopman variants, indicating that the learned operator increases the model capacity to capture trajectory-specific latent dynamics. Overall, the training behavior confirms that Koopman augmentation improves in-sample fitting by explicitly modeling evolution in the learned latent space.

In contrast to the training results, the test performance demonstrates that lower optimization error does not necessarily translate into improved generalization. While several Koopman-enhanced models maintain competitive or slightly improved test MSE and MAE relative to the plain backbones, the gains are considerably smaller and much less consistent across architectures. Structured Koopman variants, particularly the constrained, `scalar-gated`, and `per-mode gated` formulations, generally exhibit the most stable behavior, with test-error distributions remaining close to or marginally below those of the plain models. This indicates that moderate constraints on latent spectral evolution can improve temporal propagation while avoiding severe overfitting.

The test results indicate that Koopman-enhanced latent dynamics are particularly effective for the `iTransformer`, `Autoformer`, and `TimesNet` backbones. In these architectures, several Koopman variants achieve lower or more stable test MSE and MAE distributions compared with the corresponding plain backbone models, demonstrating improved temporal generalization. The improvements are especially visible

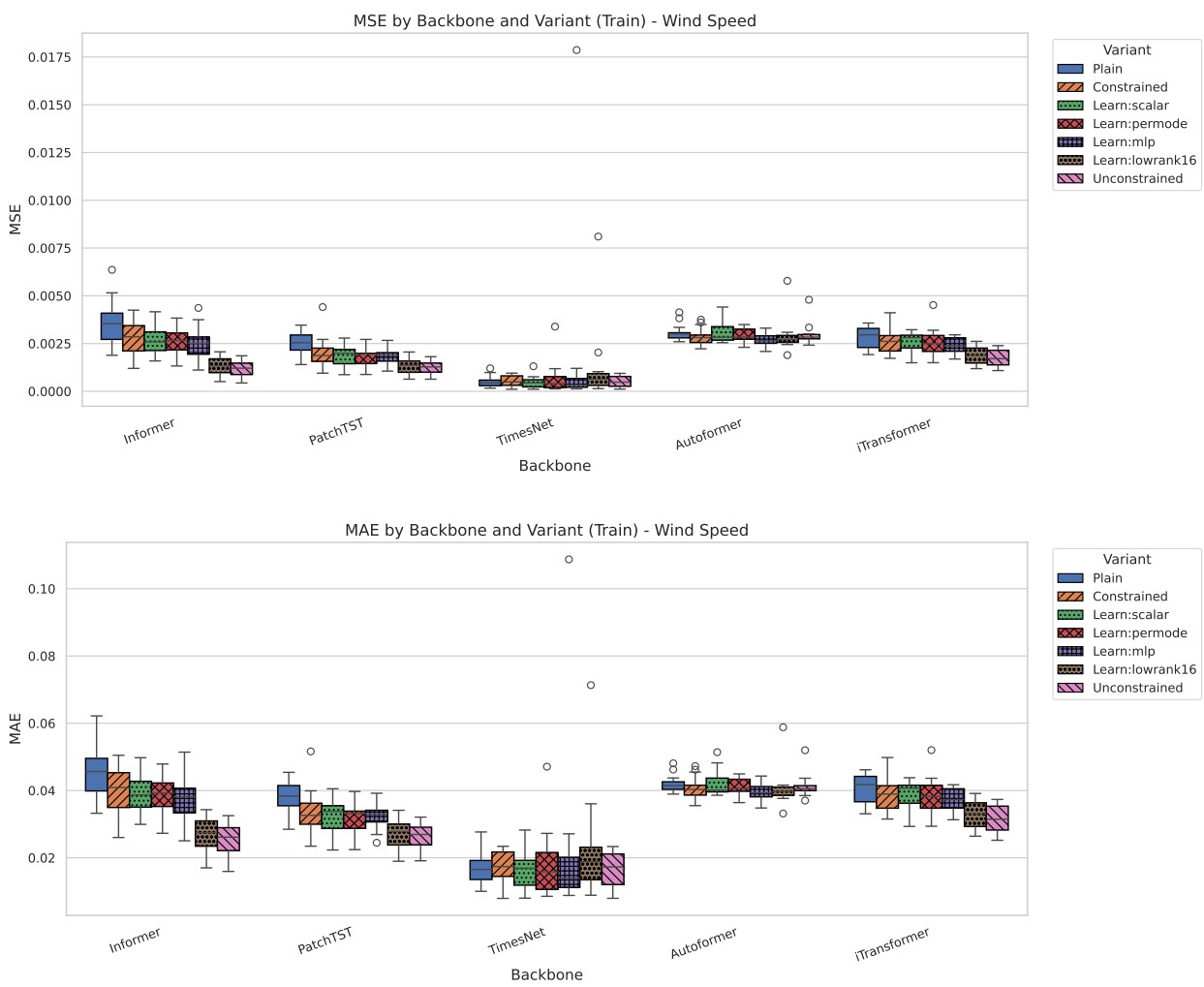

Figure 5: Training MSE and MAE distributions for the considered backbone architectures and Koopman variants on the Wind speed dataset. Koopman-enhanced models consistently achieve lower training errors than the corresponding plain backbones, indicating improved latent temporal fitting and optimization capability 4.1.2

for the constrained, `scalar-gated`, and `per-mode gated` formulations, suggesting that structured latent spectral evolution provides a beneficial regularization effect during forecasting. For `iTransformer` and `Autoformer`, the Koopman layer appears to complement the transformer-based temporal representation by introducing a more explicit dynamical propagation mechanism in latent space. Similarly, although `TimesNet` exhibits strong training overfitting, its structured Koopman variants still provide competitive test behavior relative to the plain model, indicating that controlled spectral evolution can enhance periodic temporal modeling when sufficient regularization is imposed. Overall, these observations suggest that Koopman augmentation is most beneficial for backbones whose latent representations can effectively exploit explicit dynamical evolution while remaining sufficiently constrained to avoid spectral overfitting. Architectures that already incorporate strong linear propagation mechanisms (e.g., `DLinear` or SSM-style models) naturally leave less room for improvement from an additional structured latent transition.

## 4.2 Scenario II: Comparing Transformer backbones and Baselines

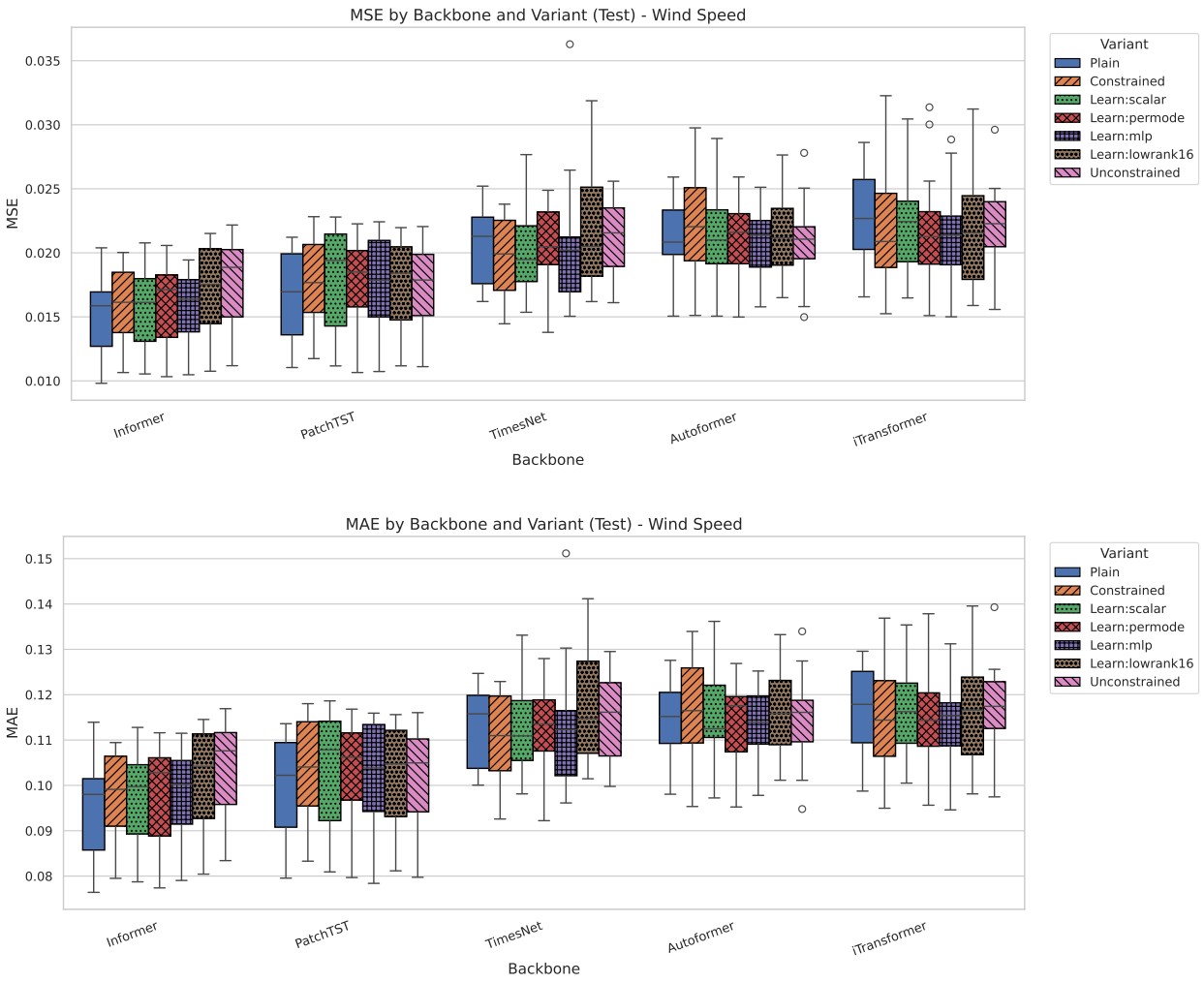

Figure 6: Test MSE and MAE distributions for the considered backbone architectures and Koopman variants on the Wind Speed dataset. Structured Koopman formulations generally provide more stable and competitive generalization performance, whereas highly flexible variants exhibit larger variance and occasional spectral overfitting 4.1.2

In this scenario, we evaluate `Learnable-DeepKoopFormer` in a controlled and scalable setting, we use a unified Koopformer benchmark on multivariate real-valued time series. From each dataset, sliding windows of length $P$ are paired with forecast targets of length $H$, over a grid $P \in \{80, 100, 120, 140\}$ and $H \in \{4, 8, 12, 16\}$. Inputs are feature-wise normalised and split chronologically into 80%/20% train/test sets.

To quantitatively assess the bias–variance characteristics of the proposed models, we additionally report summary statistics of the forecasting errors across all $(P, H)$ configurations, including mean, median, standard deviation, interquartile range (IQR), and worst-case error for both MSE and MAE. Here, $Q_1$ and $Q_3$ denote the first and third quartiles of the error distribution, corresponding to the 25th and 75th percentiles, respectively. The interquartile range (IQR), defined as $Q_3 - Q_1$, measures the dispersion of the central 50% of forecasting errors and provides a robust estimate of variability less sensitive to outliers than the standard deviation.

**Spectral logging and diagnostics** For all models with an explicit linear propagator, we log spectral quantities to characterise stability and latent energy dynamics. In Koopman-based architectures, latent

evolution $z_{t+1} = \mathcal{K}_\phi z_t$ is governed by the spectrum of $\mathcal{K}_\phi$, which controls contraction or amplification over time. In constrained and learnable ODO variants, $\mathcal{K}_\phi = U_\phi \Sigma_\phi V_\phi^\top$, with orthonormal $U_\phi, V_\phi$ and diagonal $\Sigma_\phi = \mathrm{diag}(\sigma_{i,\phi})$ obtained via a squashing map enforcing $\sigma_{i,\phi} \in (0, \rho_{\max})$, $\rho_{\max} < 1$. We log the singular values $\{\sigma_i(\mathcal{K}_\phi)\}$ and in particular $\max_i \sigma_i(\mathcal{K}_\phi)$ as a proxy for the spectral radius, allowing us to visualise how different parametrisations (`scalar`, `per-mode`, `MLP`, `low-rank`) populate the disc of radius $\rho_{\max}$.

We compare three linear-transition regimes: (i) *constrained Koopman* (`constr`), enforcing $\max_i |\Sigma_{\phi,i}| < \rho_{\max} < 1$ for uniform contraction; (ii) *unconstrained Koopman* (`unconstr`), replacing the ODO structure with a free matrix $\mathcal{K}_\phi \in \mathbb{R}^{d \times d}$ and monitoring its eigenvalues $\{\lambda_i(\mathcal{K}_\phi)\}$ and spectral radius $\rho(\mathcal{K}_\phi) = \max_i |\lambda_i(\mathcal{K}_\phi)|$; and (iii) a linear state-space model (SSM), $h_{t+1} = Ah_t + Bx_t$, $\hat{y}_t = Ch_t$, where $A$ is unconstrained. Since $A$ is generally non-normal, we log its singular values—particularly $\|A\|_2$—as a stability and conditioning diagnostic. This logging procedure maps each trained model to a spectral data point, enabling distributional comparisons (e.g., violin plots) that reveal stability–accuracy trade-offs across Koopman parametrisations and Transformer backbones.

### 4.2.1 Discussion on Wind Speed dataset

For the Wind Speed forecasting task, Fig. 7 summarises train/test MSE and MAE across all $(P, H)$ configurations. The results show that the effect of Koopman augmentation is backbone-dependent rather than uniformly dominant. Among the Transformer-based models, `Informer` and `PatchTST` generally form the lowest and most compact error distributions, indicating favourable accuracy–stability behaviour across patch lengths and horizons. `Autoformer`, `iTransformer`, and `TimesNet` exhibit broader spreads, with several long-horizon cases producing heavier upper tails; `TimesNet` in particular attains very low training errors but comparatively larger test errors, suggesting a larger generalisation gap.

The classical baselines also show distinct behaviour. LSTM has consistently high test errors and broad distributions, indicating limited robustness across forecasting settings. In contrast, `DLinear` and SSM achieve competitive test errors, especially at shorter horizons, although their train errors are substantially higher than those of the `DeepKoopFormer`–Transformer variants. Within the Koopman family, constrained operators often improve stability, but the advantage is not universal across all backbones: learnable variants such as `scalar-gated`, `per-mode gated`, `MLP-shaped` spectral mapping, and `low-rank` Koopman can match or exceed the constrained variant in selected configurations, while the unconstrained operator occasionally yields competitive medians but more pronounced tails. Overall, the strongest Koopman gains are observed for `PatchTST` and `Informer`, whereas `iTransformer` and `TimesNet` introduce greater variability, highlighting that Koopman spectral structure is most effective when paired with a compatible forecasting backbone.

Table 3 reports aggregated statistics across backbones, summarising performance over all patch lengths and forecasting horizons. Among the baselines, `DLinear` and SSM achieve the lowest test errors on average, with comparable MSE (around $1.1 \times 10^{-2}$) and MAE (around $8.3 \times 10^{-2}$–$8.5 \times 10^{-2}$), while LSTM exhibits substantially higher test error and variance despite lower training error, indicating overfitting. The Koopman-augmented models show consistent and stable behaviour across different backbone architectures, with relatively low standard deviations, reflecting robustness to configuration changes. In particular, `PatchTST` and `Informer` achieve competitive performance among the Koopman variants, although their average test errors remain slightly higher than the strongest linear baselines. Overall, these aggregated results suggest that while Koopman augmentation improves stability and consistency across architectures, its advantage is more pronounced in regularisation and structured dynamics rather than uniformly reducing the average forecasting error.

Ranking the backbones by mean test MSE, *Baseline DLinear* performs best (0.01092), followed closely by *Baseline SSM* (0.01133). The next tier includes *Koop-Informer* (0.01627) and *Koop-PatchTST* (0.01734), followed by *Koopman-TimesNet* (0.02067), *Koopman-Autoformer* (0.02147), and *Koopman-iTransformer* (0.02181). *Baseline LSTM* ranks lowest among standard models (0.03280).

**Koopman spectra across backbones — Wind Speed** For each configuration, we pool latent spectral magnitudes $\rho = |\lambda|$ across all backbones and variants, including `PatchTST`, `Autoformer`, `Informer`, `iTransformer`, and `TimesNet`. Figure 8 reports these distributions alongside the SSM singular-value spec-

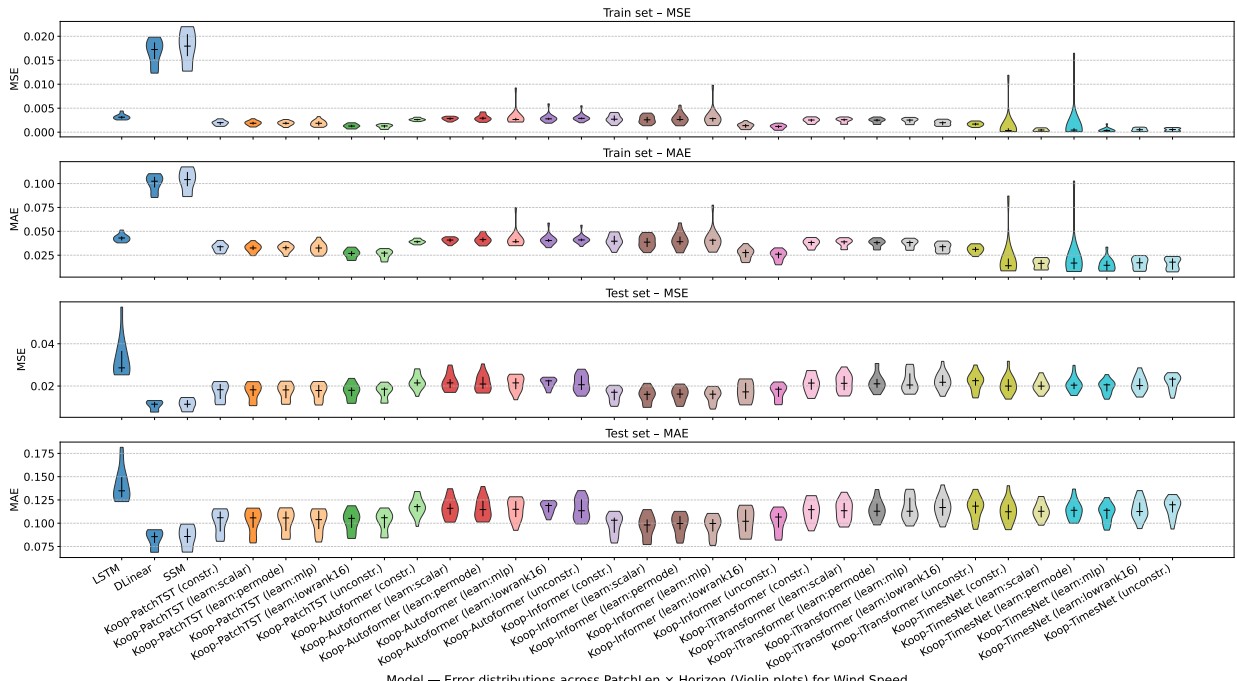

Figure 7: Violin plots of train/test MSE and MAE for all architectures on the Wind Speed forecasting task, aggregated over all patch lengths and horizons.

Table 3: Backbone-level statistical summary of MSE and MAE for the Wind Speed test dataset. Lower values are better.

| Backbone | Set | Metric | Mean ± Std | Median | Q1 | Q3 |
|---|---|---|---|---|---|---|
| DLinear | Test | MSE | $0.0085 \pm 0.0032$ | 0.0086 | 0.0063 | 0.0108 |
| DLinear | Test | MAE | $0.0696 \pm 0.0147$ | 0.0706 | 0.0598 | 0.0813 |
| LSTM | Test | MSE | $0.0179 \pm 0.0128$ | 0.0140 | 0.0092 | 0.0244 |
| LSTM | Test | MAE | $0.0907 \pm 0.0376$ | 0.0859 | 0.0697 | 0.1182 |
| SSM | Test | MSE | $0.0061 \pm 0.0031$ | 0.0061 | 0.0039 | 0.0085 |
| SSM | Test | MAE | $0.0538 \pm 0.0168$ | 0.0557 | 0.0433 | 0.0669 |
| Koop-Autoformer | Test | MSE | $0.0144 \pm 0.0068$ | 0.0147 | 0.0086 | 0.0197 |
| Koop-Autoformer | Test | MAE | $0.0881 \pm 0.0234$ | 0.0915 | 0.0689 | 0.1075 |
| Koop-Informer | Test | MSE | $0.0116 \pm 0.0068$ | 0.0116 | 0.0060 | 0.0167 |
| Koop-Informer | Test | MAE | $0.0751 \pm 0.0257$ | 0.0778 | 0.0552 | 0.0955 |
| Koop-PatchTST | Test | MSE | $0.0123 \pm 0.0073$ | 0.0123 | 0.0066 | 0.0183 |
| Koop-PatchTST | Test | MAE | $0.0773 \pm 0.0280$ | 0.0792 | 0.0585 | 0.0996 |
| Koop-TimesNet | Test | MSE | $0.1158 \pm 0.8831$ | 0.0196 | 0.0096 | 0.0321 |
| Koop-TimesNet | Test | MAE | $0.1288 \pm 0.2667$ | 0.0983 | 0.0751 | 0.1243 |
| Koop-iTransformer | Test | MSE | $0.0153 \pm 0.0078$ | 0.0154 | 0.0082 | 0.0212 |
| Koop-iTransformer | Test | MAE | $0.0876 \pm 0.0264$ | 0.0913 | 0.0650 | 0.1081 |

trum. Constrained Koopman models (`constr`) exhibit compact spectra concentrated in $\rho \approx 0.3$–$0.8$, with virtually no mass above 1, confirming effective enforcement of the spectral bound across all architectures.

Learnable variants (`scalar-gated`, `per-mode gated`, `MLP-shaped` spectral mapping, `low-rank` Koopman) occupy a similar range but with broader support, indicating that they exploit a larger portion of the admissible spectral disc while remaining predominantly contractive. This behaviour is consistent across Transformer-based backbones as well as the more recent `iTransformer` and `TimesNet` architectures, suggesting that the Koopman parametrisation governs the spectral structure more strongly than the choice of encoder.

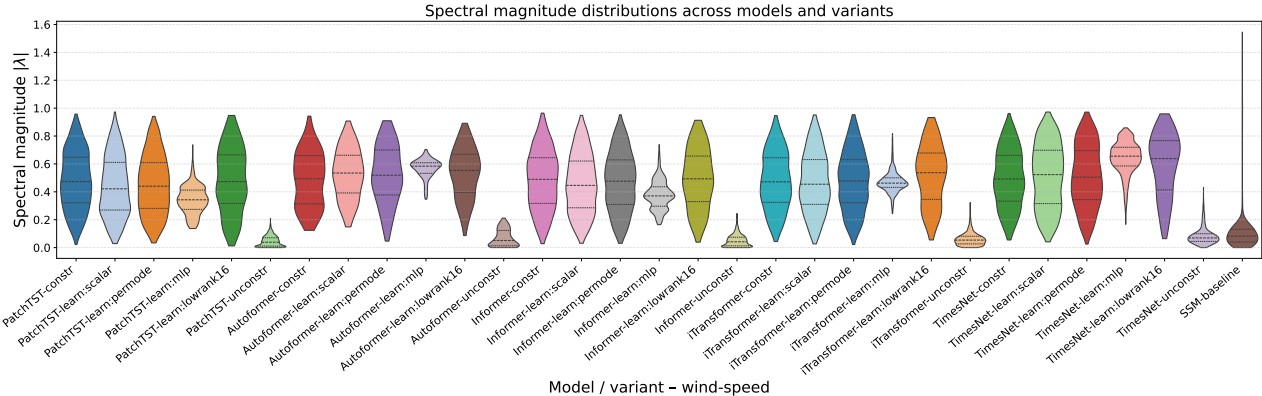

Figure 8: Spectral distributions of latent Koopman operators for the Wind Speed forecasting task.

In contrast, `unconstr` Koopman operators collapse toward small radii ($\rho \lesssim 0.2$), indicating strongly damped latent dynamics driven primarily by data fitting and regularisation effects. The SSM baseline similarly concentrates at small magnitudes ($\rho \approx 0.05$–$0.2$), but exhibits a thin tail extending beyond the unit circle, reflecting occasional unstable or poorly conditioned transitions. Overall, these results indicate that spectral behaviour is largely controlled by the Koopman design, with constrained and learnable variants achieving a balance between expressivity and stability across a diverse set of backbones.

### 4.2.2 Discussion on Pressure Surface dataset

For the CMIP6 pressure–surface task, Fig. 9 reports train/test MSE and MAE across all $(P, H)$ configurations. Koopman–Transformer models based on `PatchTST`, `Autoformer`, `Informer`, and `iTransformer` generally exhibit low-centred and compact error distributions, indicating stable performance across forecasting settings. In contrast, LSTM, `DLinear`, and SSM often show higher-centred and broader distributions, reflecting greater variability, especially at longer horizons.

Among Koopman variants, the constrained operator (`constr`) typically yields the most compact and low-variance errors across backbones. Learnable families (`scalar-gated`, `per-mode gated`, `MLP-shaped` spectral mapping, `low-rank` Koopman) achieve comparable medians, although with slightly wider spreads, reflecting increased flexibility with a limited loss of robustness. The unconstrained operator (`unconstr`) displays heavier tails—particularly for `PatchTST`, `Informer`, and `iTransformer`—indicating reduced reliability without spectral regularisation.

The additional `TimesNet` backbone shows a markedly different behaviour. While some `TimesNet` configurations achieve competitive errors, several settings produce very large outliers, especially for longer horizons and certain learnable or unconstrained Koopman operators. These failures substantially distort the violin ranges and obscure the comparison among the remaining models. For this reason, `TimesNet` results are omitted from Fig. 9, but they are retained in the numerical results for completeness.

Although certain baseline configurations, notably `DLinear` and SSM, attain slightly lower mean test errors in some cases, these improvements are accompanied by substantially wider spreads, suggesting less consistent generalisation. In contrast, Koopman models built on stable Transformer backbones maintain competitive mean errors with markedly reduced variability, highlighting the stabilising effect of structured latent dynamics.

Table 4 summarises the aggregated mean, standard deviation, and median errors across all $(P, H)$ configurations. Among baseline models, SSM achieves the lowest mean test error (MSE $\approx 0.0061$, MAE $\approx 0.0538$), followed by `DLinear`, while LSTM exhibits substantially higher variance and worse generalisation (test MSE $\approx 0.0179$ with large standard deviation).

Koopman-based Transformer backbones show competitive performance with improved stability. In particular, `Informer` and `PatchTST` achieve mean test MSE values of approximately 0.0116 and 0.0123, respectively,

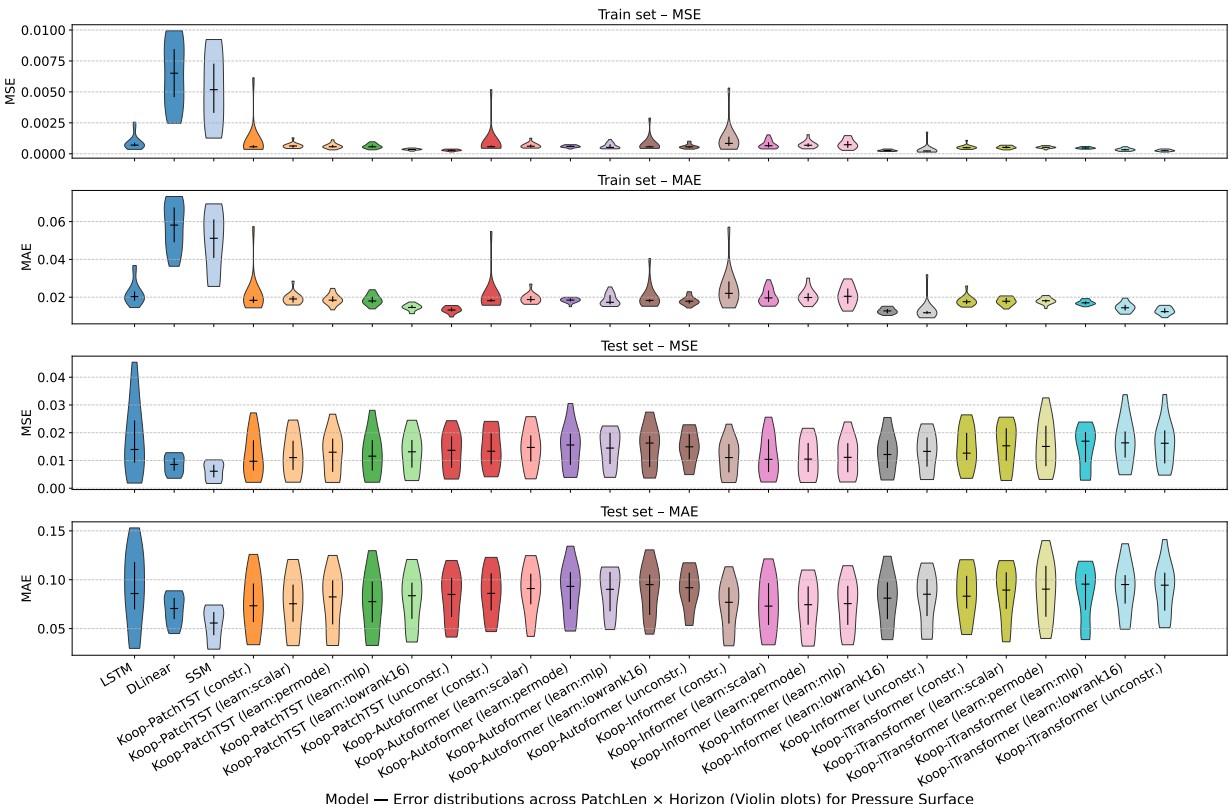

Figure 9: Violin plots of train/test MSE and MAE for all architectures on the Pressure Surface forecasting task, aggregated over all patch lengths and horizons.

Table 4: Backbone-level statistical summary of MSE and MAE for the Pressure Surface test dataset. Lower values are better.

| Backbone | Set | Metric | Mean $\pm$ Std | Median | Q1 | Q3 |
|---|---|---|---|---|---|---|
| DLinear | Test | MSE | $0.0085 \pm 0.0032$ | 0.0086 | 0.0063 | 0.0108 |
| DLinear | Test | MAE | $0.0696 \pm 0.0147$ | 0.0706 | 0.0598 | 0.0813 |
| LSTM | Test | MSE | $0.0179 \pm 0.0128$ | 0.0140 | 0.0092 | 0.0244 |
| LSTM | Test | MAE | $0.0907 \pm 0.0376$ | 0.0859 | 0.0697 | 0.1182 |
| SSM | Test | MSE | $0.0061 \pm 0.0031$ | 0.0061 | 0.0039 | 0.0085 |
| SSM | Test | MAE | $0.0538 \pm 0.0168$ | 0.0557 | 0.0433 | 0.0669 |
| Koop-Autoformer | Test | MSE | $0.0144 \pm 0.0068$ | 0.0147 | 0.0086 | 0.0197 |
| Koop-Autoformer | Test | MAE | $0.0881 \pm 0.0234$ | 0.0915 | 0.0689 | 0.1075 |
| Koop-Informer | Test | MSE | $0.0116 \pm 0.0068$ | 0.0116 | 0.0060 | 0.0167 |
| Koop-Informer | Test | MAE | $0.0751 \pm 0.0257$ | 0.0778 | 0.0552 | 0.0955 |
| Koop-PatchTST | Test | MSE | $0.0123 \pm 0.0073$ | 0.0123 | 0.0066 | 0.0183 |
| Koop-PatchTST | Test | MAE | $0.0773 \pm 0.0280$ | 0.0792 | 0.0585 | 0.0996 |
| Koop-TimesNet | Test | MSE | $0.1158 \pm 0.8831$ | 0.0196 | 0.0096 | 0.0321 |
| Koop-TimesNet | Test | MAE | $0.1288 \pm 0.2667$ | 0.0983 | 0.0751 | 0.1243 |
| Koop-iTransformer | Test | MSE | $0.0153 \pm 0.0078$ | 0.0154 | 0.0082 | 0.0212 |
| Koop-iTransformer | Test | MAE | $0.0876 \pm 0.0264$ | 0.0913 | 0.0650 | 0.1081 |

with moderate variance, while `iTransformer` performs similarly (test MSE $\approx 0.0153$) but with slightly higher dispersion. `Autoformer` shows somewhat higher mean errors (test MSE $\approx 0.0144$), consistent with its broader distributions observed earlier.

A notable outlier is `TimesNet`, which exhibits dramatically larger mean errors and extremely high variance (test MSE mean $\approx 0.116$ with standard deviation $\approx 0.88$). Importantly, its median test MSE ($\approx 0.0196$) is much lower than its mean, indicating that the poor average performance is driven by a small number of catastrophic failures rather than consistently poor predictions. This heavy-tailed behaviour is also reflected in MAE statistics and aligns with the instability observed in the per-configuration results.

Overall, Koopman models built on stable Transformer backbones (`PatchTST`, `Informer`, `iTransformer`) maintain a favourable trade-off between accuracy and robustness, achieving competitive mean errors with relatively controlled variance. In contrast, `TimesNet`-based Koopman models suffer from severe instability in certain regimes, which significantly degrades their aggregate performance despite reasonable median behaviour.

Ranking the backbones by mean test MSE, *Baseline SSM* performs best (0.00606), followed by *Baseline DLinear* (0.00850). The next tier includes *Koopman-Informer* (0.01165) and *Koopman-PatchTST* (0.01228), followed by *Koopman-Autoformer* (0.01445) and *Koopman-iTransformer* (0.01530). *Baseline LSTM* ranks lower (0.01788), while *Koopman-TimesNet* performs worst overall (0.11582), showing high variance and instability.

**Koopman spectra across backbones—Pressure Surface** For each configuration, we pool latent spectral magnitudes $\rho = |\lambda|$ by backbone and variant (Fig. 10). Constrained Koopman models (`constr`) exhibit compact spectra concentrated mainly in $\rho \approx 0.3$–0.8, with no mass above 1, confirming effective enforcement of the spectral bound. Learnable variants (`scalar-gated`, `per-mode gated`, `MLP-shaped` spectral mapping, `low-rank` Koopman) occupy a similar sub-unit range, but with broader support across modes, indicating additional flexibility while remaining largely contractive.

The unconstrained variants display a markedly different pattern. For `PatchTST`, `Autoformer`, `Informer`, `iTransformer`, and `TimesNet`, the `unconstr` spectra collapse toward very small radii, often around $\rho \lesssim 0.1$, suggesting strongly damped latent dynamics rather than richer long-range propagation. The `MLP-shaped` spectral mapping variants tend to form narrow, mid-radius spectra, whereas the low-rank variants generally preserve wider spectral support comparable to the constrained case.

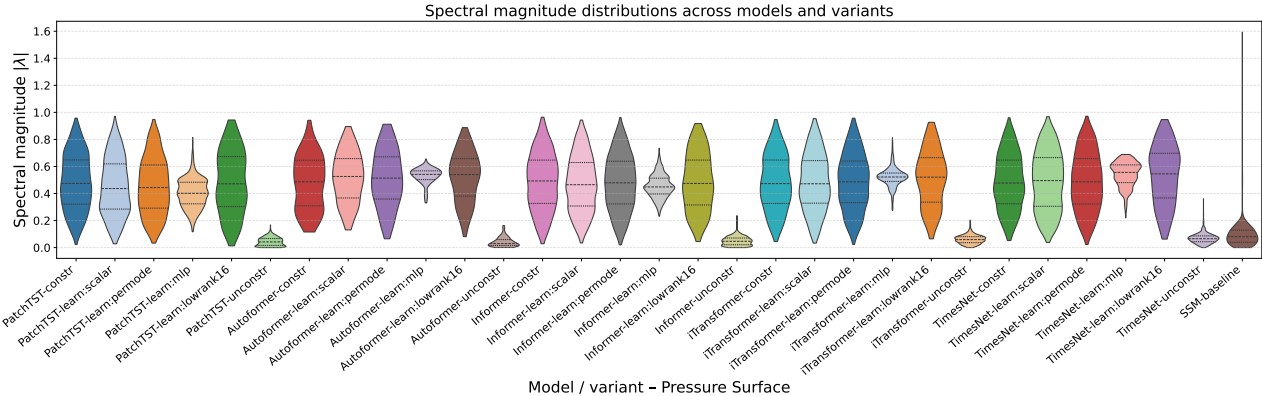

Figure 10: Spectral distributions of latent Koopman operators for the Pressure Surface forecasting task.

The newly included `iTransformer` and `TimesNet` backbones follow the same overall trend: stable, sub-unit spectra for constrained and learnable Koopman operators, and near-zero spectra for unconstrained operators. However, the SSM baseline differs substantially, concentrating most mass near small magnitudes while exhibiting a pronounced tail that extends well beyond the unit circle. This indicates occasional unstable modes and contrasts with the explicitly bounded Koopman spectra. Overall, spectral behaviour is shaped more strongly by the Koopman parametrisation than by the choice of Transformer backbone, with constrained and learnable variants providing stable latent dynamics for pressure–surface forecasting.

## 4.3 Scenario III: Augmenting Koopman latent with Non Transformer Backbones

In this scenario, we extend the Koopman augmentation beyond Transformer-based forecasting backbones and examine its interaction with non-Transformer architectures. In particular, we consider `DLinear` and SSM as lightweight and dynamically motivated alternatives, and compare their plain versions with Koopman-enhanced counterparts. This allows us to assess whether the proposed latent Koopman evolution is specifically beneficial for attention-based models or whether it can also improve simpler linear and state-space forecasting structures.

### 4.3.1 Discussion on Cryptocurrency dataset

For the Cryptocurrency forecasting task, Fig. 11 provides a general overview of train/test MSE and MAE across all $(P, H)$ configurations. Among the non-Transformer baselines, SSM is the strongest and most stable model, yielding the lowest test errors with compact distributions, while `DLinear` remains competitive but exhibits slightly higher errors. In contrast, `LSTM` shows noticeably broader test-error distributions and larger MAE, indicating stronger sensitivity to window and horizon choices. Within the Koopman family, Koopman-SSM achieves the best overall performance, closely matching or improving upon the SSM baseline and producing the most compact test-error violins. The Koopman–Transformer variants, including `PatchTST`, `Autoformer`, `Informer`, and `iTransformer`, generally reduce training error but show larger test spreads, suggesting weaker generalisation than Koopman-SSM on this volatile task. Among them, `Informer` and `PatchTST` are relatively more stable, whereas `Autoformer` and `iTransformer` exhibit higher test errors. The `TimesNet` backbone performs poorly, with substantially inflated test MSE/MAE and wide distributions, therefore we omitted its results. Across variants, constrained and learnable Koopman operators usually provide better-controlled errors than the `unconstr` operator, whose heavier tails indicate reduced robustness without spectral control.

Across backbone-level aggregates as reported in Table 5, Koopman-SSM gives the best overall test performance, achieving the lowest mean MSE and MAE among all models and slightly outperforming the SSM baseline. Among the non-Transformer baselines, SSM is clearly strongest, followed by `DLinear`, while LSTM has substantially higher test error and variability despite lower training error, indicating weaker generalisation. The Koopman-enhanced Transformer backbones show much lower training losses than the baselines,

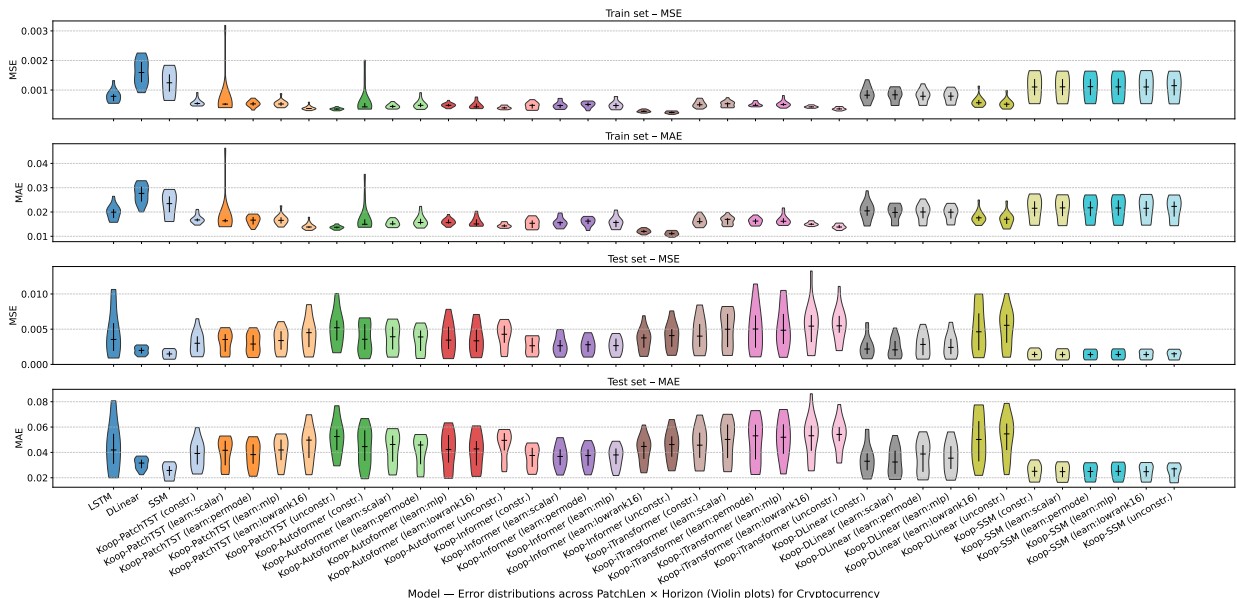

Figure 11: Violin plots of train/test MSE and MAE for all architectures on the Cryptocurrency forecasting task, aggregated over all patch lengths and horizons.

but their test errors are generally higher than SSM and Koopman-SSM, suggesting overfitting or sensitivity to the Cryptocurrency forecasting setting. Within this group, `Informer` is the most competitive, followed by `DLinear`, `PatchTST`, and `Autoformer`, whereas `iTransformer` performs worse with larger test errors. The `TimesNet` backbone is a clear outlier: although its training error remains low, its test MSE and MAE are an order of magnitude larger, confirming poor robustness on the Cryptocurrency task. Overall, the results indicate that Koopman structure is most beneficial when paired with the SSM backbone, while more expressive Transformer-style backbones do not consistently translate their low training errors into better test accuracy.

Figures **??**–**??** further illustrate the distribution of test MSE and MAE across backbones and Koopman variants. The Koopman-SSM models exhibit the tightest and lowest error ranges across all operator choices, confirming both strong accuracy and robustness. The baseline SSM also remains highly competitive, outperforming the baseline LSTM and `DLinear` models by a clear margin. Among the Transformer-style Koopman backbones, `Informer` consistently yields the most compact distributions and lowest medians, while `PatchTST` and `Autoformer` show broader spreads and moderate increases in error. The `iTransformer` variants exhibit the highest medians and largest variability among the Transformer backbones, indicating reduced stability across $(P, H)$ settings. Across nearly all architectures, the constrained operator generally produces lower and more concentrated errors, whereas the `unconstr` and some learnable variants display heavier tails and more outliers, reflecting weaker robustness under volatile Cryptocurrency dynamics.

Ranking the backbones by mean test MSE (lower is better), *Koopman-SSM* performs best (0.00141), followed closely by *Baseline SSM* (0.00147) and *Baseline `DLinear`* (0.00198). A middle tier includes *Koopman-`Informer`* (0.00303), *Koopman-`DLinear`* (0.00340), *Koopman-`PatchTST`* (0.00364), and *Koopman-`Autoformer`* (0.00370). Lower-ranked models are *Baseline LSTM* (0.00450) and *Koopman-`iTransformer`* (0.00506), while *Koopman-`TimesNet`* performs worst (0.02417), showing a substantial gap from the rest.

**Koopman spectra across backbones — Cryptocurrency** Pooling latent spectral magnitudes $\rho = |\lambda|$ across configurations (Fig. 12), the constrained Koopman variants (`constr`) for `PatchTST`, `Autoformer`, `Informer`, `iTransformer`, `DLinear`, and SSM remain within the stable region and show broad but controlled spectra, typically centred around $\rho \approx 0.3$–$0.7$. The learnable variants (`scalar-gated`, `per-mode gated`, `MLP-shaped` spectral mapping, `low-rank` Koopman) occupy a similar contractive range, with `MLP-shaped` spectral mapping often producing narrower spectra and `low-rank` Koopman retaining wider modal variation because of its low-rank structure. In contrast, the `unconstr` Koopman operators collapse toward very small

Table 5: Backbone-level statistical summary of MSE and MAE for the Cryptocurrency test dataset. Lower values are better.

| Backbone | Set | Metric | Mean ± Std | Median | Q1 | Q3 |
|---|---|---|---|---|---|---|
| DLinear | Test | MSE | 0.0020 ± 0.0005 | 0.0020 | 0.0016 | 0.0024 |
| DLinear | Test | MAE | 0.0310 ± 0.0043 | 0.0316 | 0.0275 | 0.0343 |
| LSTM | Test | MSE | 0.0045 ± 0.0031 | 0.0035 | 0.0019 | 0.0059 |
| LSTM | Test | MAE | 0.0449 ± 0.0186 | 0.0419 | 0.0310 | 0.0547 |
| SSM | Test | MSE | 0.0015 ± 0.0005 | 0.0015 | 0.0011 | 0.0018 |
| SSM | Test | MAE | 0.0252 ± 0.0054 | 0.0258 | 0.0216 | 0.0292 |
| Koop-Autoformer | Test | MSE | 0.0037 ± 0.0019 | 0.0037 | 0.0019 | 0.0054 |
| Koop-Autoformer | Test | MAE | 0.0425 ± 0.0129 | 0.0450 | 0.0316 | 0.0535 |
| Koop-DLinear | Test | MSE | 0.0034 ± 0.0024 | 0.0031 | 0.0016 | 0.0046 |
| Koop-DLinear | Test | MAE | 0.0403 ± 0.0155 | 0.0391 | 0.0276 | 0.0502 |
| Koop-Informer | Test | MSE | 0.0030 ± 0.0014 | 0.0029 | 0.0018 | 0.0041 |
| Koop-Informer | Test | MAE | 0.0387 ± 0.0098 | 0.0387 | 0.0300 | 0.0457 |
| Koop-PatchTST | Test | MSE | 0.0036 ± 0.0019 | 0.0035 | 0.0020 | 0.0048 |
| Koop-PatchTST | Test | MAE | 0.0422 ± 0.0124 | 0.0423 | 0.0320 | 0.0504 |
| Koop-SSM | Test | MSE | 0.0014 ± 0.0006 | 0.0015 | 0.0010 | 0.0018 |
| Koop-SSM | Test | MAE | 0.0247 ± 0.0056 | 0.0267 | 0.0206 | 0.0295 |
| Koop-TimesNet | Test | MSE | 0.0242 ± 0.0215 | 0.0207 | 0.0097 | 0.0311 |
| Koop-TimesNet | Test | MAE | 0.1084 ± 0.0478 | 0.1104 | 0.0764 | 0.1338 |
| Koop-iTransformer | Test | MSE | 0.0051 ± 0.0027 | 0.0051 | 0.0027 | 0.0069 |
| Koop-iTransformer | Test | MAE | 0.0496 ± 0.0149 | 0.0520 | 0.0373 | 0.0611 |

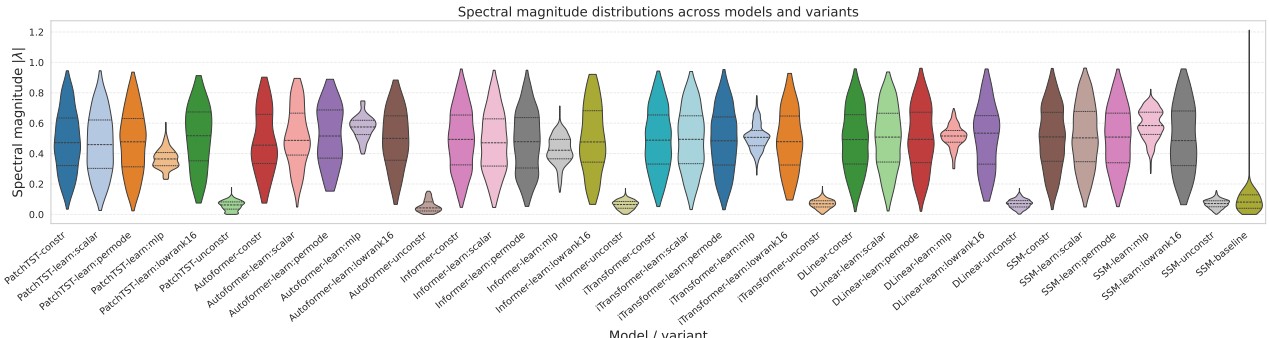

Figure 12: Spectral distributions of latent Koopman operators for the Cryptocurrency forecasting task.

magnitudes across most backbones, indicating strongly damped latent dynamics rather than expressive long-range propagation. This pattern is visible not only for Transformer-style backbones but also for the Koopman-DLinear and Koopman-SSM models, suggesting that spectral behaviour is mainly determined by the Koopman parametrisation rather than the backbone family. The plain SSM baseline also concentrates near small spectral magnitudes, with a thin high-radius tail, indicating occasional less stable modes. Overall, constrained and learnable Koopman parametrisations provide stable yet expressive latent evolution, while unconstrained operators tend to become over-damped in the volatile Cryptocurrency setting.

### 4.3.2 Discussion on Energy Systems Dataset

Figure 14 reports train/test MSE and MAE across all $(P, H)$ settings. Unlike the previous domains, performance on the energy dataset is more heterogeneous across both transformer and non-transformer backbones. Among the non-transformer baselines, DLinear and SSM are the strongest: they obtain low test errors with relatively compact distributions, with SSM and Koopman-SSM giving some of the best overall test performance. In contrast, LSTM achieves very low training errors but much larger test errors, indicating clear overfitting and weaker generalization.

For transformer-style backbones, `PatchTST` and `Informer` remain competitive, while `iTransformer` and `Autoformer` show higher errors and broader spreads. `TimesNet` performs poorly on this dataset, with substantially higher test MSE/MAE and wide variability despite low training errors, suggesting poor robustness for these energy forecasting settings.

Across backbones, Koopman-enhanced models often improve stability, especially for SSM, `PatchTST`, and `Informer`. The constrained operator (`constr`) generally produces compact, well-centred error distributions, supporting the benefit of spectral control for slow-varying energy dynamics. Learnable variants (`scalar-gated`, `per-mode gated`, `MLP-shaped` spectral mapping, `low-rank` Koopman) can achieve competitive medians but often introduce broader spreads, reflecting an expressiveness–robustness trade-off. Overall, the results show that low training error alone does not ensure reliable forecasting, while spectrally controlled Koopman models tend to yield more robust error distributions across hyperparameter settings.

Table 6 summarizes the aggregate train and test statistics across all $(P, H)$ configurations for the energy forecasting task. Among the baseline models, `DLinear` and SSM achieve the strongest overall generalization, with low test MSE and MAE together with relatively moderate variability. In contrast, Lstm attains substantially lower training errors but considerably worse test performance, confirming strong overfitting behaviour.

Across Koopman-enhanced backbones, Koopman-SSM achieves the best overall test performance, obtaining the lowest mean test MSE and MAE while maintaining relatively controlled variance. Koopman-`PatchTST` and Koopman-`Informer` also perform competitively, producing test errors close to the strongest baselines while preserving low training losses. Koopman-`DLinear` and Koopman-`iTransformer` achieve moderate performance, improving training accuracy substantially but with weaker test robustness compared to Koopman-SSM or Koopman-`PatchTST`. In contrast, Koopman-`TimesNet` performs significantly worse than all other approaches, with the highest test MSE/MAE and very large standard deviations despite extremely low training errors, indicating severe overfitting and unstable behaviour across configurations. Similarly, Koopman-`Autoformer` exhibits relatively high test errors and limited robustness.

Overall, the table highlights a consistent trend: architectures with very low training errors do not necessarily generalize well on the energy dataset. Instead, Koopman-enhanced models built on stable backbones such as SSM, `PatchTST`, and `Informer` provide the best balance between accuracy, robustness, and consistency across hyperparameter settings.

Figures 14a and 14b further compare the distribution of test MSE and MAE across Koopman variants for each backbone. The results confirm that SSM-based models achieve the most stable and lowest overall error distributions, with relatively compact interquartile ranges and limited outliers across variants. `PatchTST` and `Informer` also maintain competitive and consistent behaviour, although with slightly higher medians and broader spreads than Koopman-SSM. In contrast, `iTransformer`, Koopman-`DLinear`, and especially Koopman-`Autoformer` exhibit higher medians and larger variability, indicating reduced robustness to Koopman parameterization choices.

Models ranked by backbone using mean test MSE (lower is better) show that *Koopman-SSM* performs best (0.00988), followed by *Baseline `DLinear`* (0.01073). Next are *Koopman-`PatchTST`* (0.01141), *Baseline SSM* (0.01141), and *Koopman-`Informer`* (0.01166), which form a competitive middle group. *Koopman-`iTransformer`* (0.01491) and *Koopman-`DLinear`* (0.01550) show moderate performance, while *Koop-`Autoformer`* (0.01832) and *Baseline LSTM* (0.02192) lag behind. *Koop-`TimesNet`* performs worst (0.05432), indicating a significant drop relative to other backbones.

**Koopman spectra across backbones— Energy Systems** Pooling latent spectral magnitudes $\rho = |\lambda|$ across configurations (Fig. 15), the constrained Koopman variants (`constr`) for `PatchTST`, `Autoformer`, `Informer`, `iTransformer`, `DLinear`, and SSM consistently remain within the stable contractive regime,

with broad spectral support typically concentrated around $\rho \approx 0.3$–$0.7$. Their distributions are relatively wide and multimodal, indicating richer latent dynamics while still avoiding unstable eigenvalues near or beyond the unit circle.

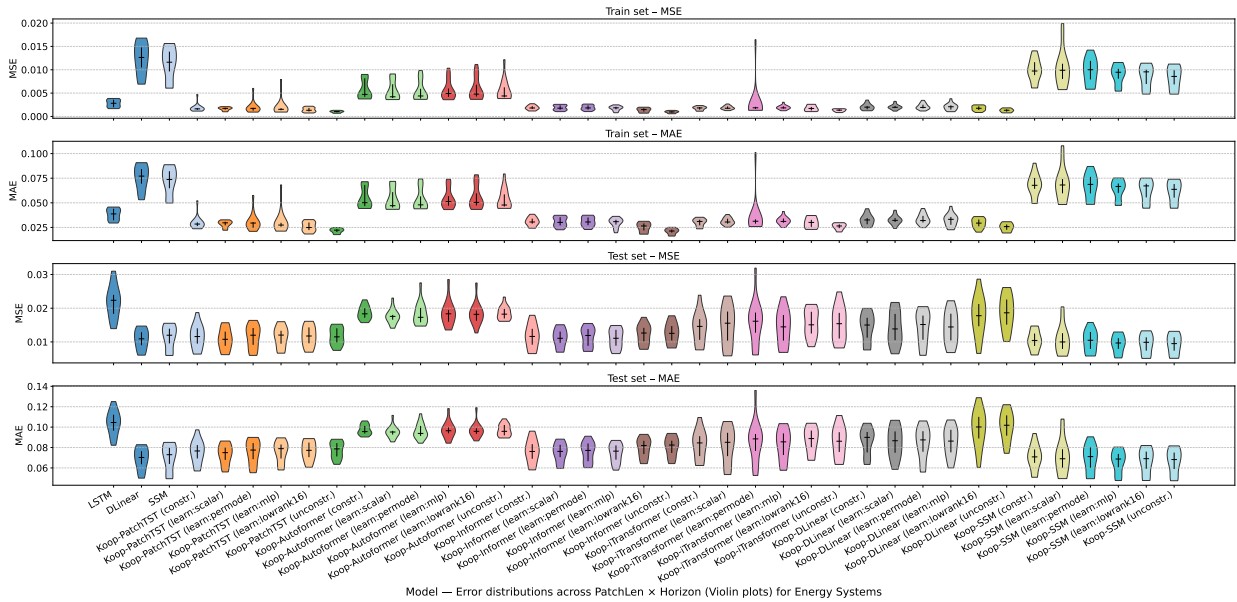

Figure 13: Violin plots of train/test MSE and MAE for all architectures on the Energy Systems forecasting task, aggregated over all patch lengths and horizons.

Table 6: Backbone-level statistical summary of MSE and MAE for the Energy Systems test dataset. Lower values are better.

| Backbone | Set | Metric | Mean ± Std | Median | Q1 | Q3 |
|---|---|---|---|---|---|---|
| DLinear | Test | MSE | $0.0107 \pm 0.0028$ | 0.0109 | 0.0090 | 0.0129 |
| DLinear | Test | MAE | $0.0690 \pm 0.0103$ | 0.0703 | 0.0636 | 0.0764 |
| LSTM | Test | MSE | $0.0219 \pm 0.0046$ | 0.0224 | 0.0183 | 0.0240 |
| LSTM | Test | MAE | $0.1040 \pm 0.0114$ | 0.1045 | 0.0966 | 0.1122 |
| SSM | Test | MSE | $0.0114 \pm 0.0034$ | 0.0120 | 0.0096 | 0.0139 |
| SSM | Test | MAE | $0.0703 \pm 0.0127$ | 0.0730 | 0.0639 | 0.0796 |
| Koop-Autoformer | Test | MSE | $0.0183 \pm 0.0027$ | 0.0181 | 0.0165 | 0.0199 |
| Koop-Autoformer | Test | MAE | $0.0965 \pm 0.0065$ | 0.0954 | 0.0923 | 0.1004 |
| Koop-DLinear | Test | MSE | $0.0155 \pm 0.0051$ | 0.0156 | 0.0110 | 0.0190 |
| Koop-DLinear | Test | MAE | $0.0899 \pm 0.0162$ | 0.0909 | 0.0768 | 0.1005 |
| Koop-Informer | Test | MSE | $0.0117 \pm 0.0029$ | 0.0118 | 0.0092 | 0.0139 |
| Koop-Informer | Test | MAE | $0.0769 \pm 0.0098$ | 0.0773 | 0.0695 | 0.0844 |
| Koop-PatchTST | Test | MSE | $0.0114 \pm 0.0030$ | 0.0114 | 0.0092 | 0.0140 |
| Koop-PatchTST | Test | MAE | $0.0756 \pm 0.0099$ | 0.0771 | 0.0695 | 0.0843 |
| Koop-SSM | Test | MSE | $0.0099 \pm 0.0031$ | 0.0102 | 0.0081 | 0.0123 |
| Koop-SSM | Test | MAE | $0.0682 \pm 0.0129$ | 0.0705 | 0.0622 | 0.0778 |
| Koop-TimesNet | Test | MSE | $0.0543 \pm 0.0314$ | 0.0448 | 0.0316 | 0.0657 |
| Koop-TimesNet | Test | MAE | $0.1594 \pm 0.0397$ | 0.1496 | 0.1287 | 0.1814 |
| Koop-iTransformer | Test | MSE | $0.0149 \pm 0.0051$ | 0.0153 | 0.0111 | 0.0186 |
| Koop-iTransformer | Test | MAE | $0.0842 \pm 0.0153$ | 0.0864 | 0.0745 | 0.0945 |

Among the learnable parametrisations, `per-mode gated` and `scalar-gated` preserve similarly broad spectra, whereas `MLP-shaped` spectral mapping produces noticeably tighter and more concentrated distributions, generally centred around $\rho \approx 0.4$–$0.6$. The low-rank parametrisation `low-rank` Koopman exhibits the widest spread among the learnable variants, frequently extending toward larger magnitudes ($\rho \gtrsim 0.8$), reflecting increased modal diversity induced by the low-rank latent structure.

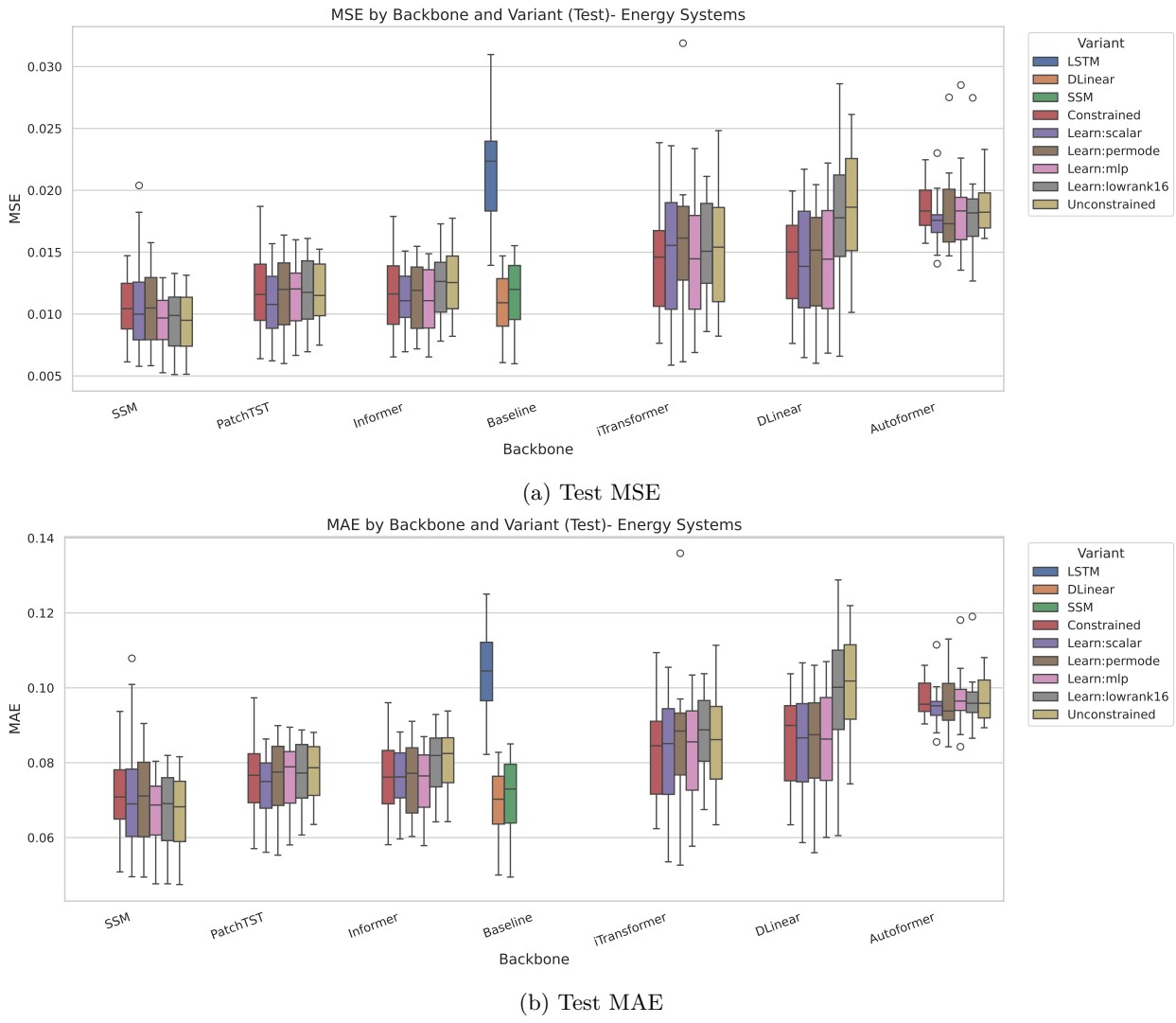

(a) Test MSE

(b) Test MAE

Figure 14: Grouped box plots for all architectures on the Energy Systems forecasting task, aggregated over all patch lengths and horizons. (a) Test MSE. (b) Test MAE.

In contrast, the `unconstr` Koopman operators collapse toward very small spectral magnitudes across nearly all backbones, with most density concentrated below $\rho \approx 0.15$. This behaviour indicates excessively damped latent transitions that suppress long-range temporal propagation and reduce dynamical expressivity. The same tendency appears consistently for Transformer-based architectures as well as Koopman-`DLinear` and Koopman-SSM variants, suggesting that the observed spectral characteristics are driven primarily by the Koopman parametrisation itself rather than by the underlying forecasting backbone.

The plain SSM baseline also concentrates heavily near small magnitudes, although it exhibits a thin high-radius tail extending close to $\rho \approx 1$, indicating the presence of a few less stable or weakly damped modes. Overall, constrained and learnable Koopman parametrisations achieve a favourable balance between stability and expressive latent evolution in the volatile Cryptocurrency forecasting regime, whereas unconstrained operators tend to become overly contractive and dynamically restrictive.

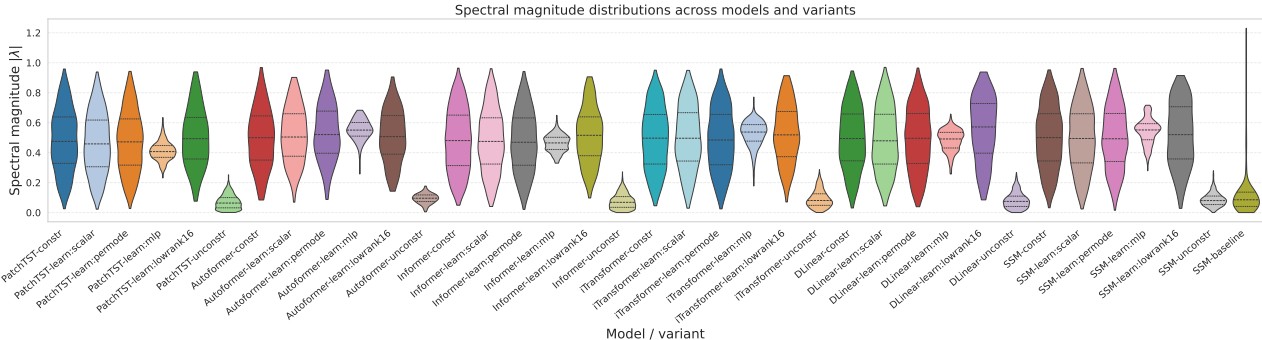

Figure 15: Spectral distributions of latent Koopman operators for the Energy Systems forecasting task.

### 4.4 Perturbation-Based Sensitivity Analysis of the Koopman Operator

The robustness and sensitivity of the Koopman-based forecasting models were evaluated through controlled perturbations of the learned Koopman operator. After training, the latent Koopman matrix $K$ was extracted and perturbed according to $\mathcal{K}_\varepsilon = \mathcal{K} + \varepsilon\Delta$, where $\varepsilon$ denotes the perturbation magnitude and $\Delta$ is a randomly generated normalized perturbation matrix. In this work, Gaussian perturbations were used, with the perturbation matrix scaled such that its Frobenius norm matched that of the original Koopman operator, ensuring comparable perturbation energy across experiments. The perturbed operator $K_\varepsilon$ was then used during inference while keeping all other network parameters fixed.

Sensitivity experiments were performed for perturbation levels $\varepsilon \in \left\{0, 10^{-4}, 10^{-3}, 10^{-2}, 10^{-1}\right\}$, and each experiment was repeated over multiple randomized trials to account for stochastic variability. For every perturbation realization, the forecasting performance was evaluated using the mean squared error (MSE) and mean absolute error (MAE), while the relative degradation was quantified through the normalized metric

$$\text{Relative MSE} = \frac{\text{MSE}_\varepsilon}{\text{MSE}_0},$$

where $\text{MSE}_0$ denotes the unperturbed baseline error.

In addition, the spectral radius

$$\rho(\mathcal{K}_\varepsilon) = \max_i |\lambda_i(K_\varepsilon)|$$

of the perturbed Koopman operator was monitored to analyze the relationship between spectral stability and predictive robustness. This perturbation framework provides a quantitative measure of how sensitive the learned Koopman dynamics are to operator uncertainty and numerical disturbances.

Each reported robustness statistic corresponds to an average over n_trials = 10 stochastic perturbation realizations of the Koopman operator.

Table 7 summarizes the overall robustness ranking of the `PatchTST`-Koopman variants across all perturbation magnitudes. The unconstrained Koopman formulation achieves the best overall robustness, yielding the lowest average relative MSE (1.0069), the smallest maximum degradation (1.0278), and the lowest variability across perturbation trials. These results indicate that the unconstrained operator remains remarkably stable even under progressively stronger stochastic perturbations.

The `low-rank` Koopman parameterization provides the second most robust configuration, with only a modest increase in average relative MSE (1.0325) and low perturbation variance. This suggests that `low-rank` Koopman Koopman parameterizations can effectively regularize the latent dynamics while preserving robustness properties. The `MLP-shaped` spectral mapping variant demonstrates intermediate robustness, exhibiting larger degradation and variability compared to the unconstrained and `low-rank` Koopman models, but still outperforming the constrained and `per-mode gated` parameterizations.

In contrast, the constrained, `scalar-gated`, and `per-mode gated` variants exhibit substantially higher sensitivity to Koopman perturbations, reflected by both larger average relative MSE values and increased variance

across trials. Among these approaches, the `per-mode gated` variant shows the weakest robustness, reaching the largest average and maximum degradation levels. Notably, the robustness ranking closely follows the magnitude of the average Koopman spectral radius $\rho(\mathcal{K})$. Variants with smaller spectral radius consistently demonstrate improved perturbation stability, whereas larger spectral radii are associated with amplified sensitivity and increased forecasting degradation. This observation suggests a strong relationship between Koopman spectral amplification and robustness behavior in latent dynamical forecasting models.

Table 8 reports the perturbation sensitivity statistics of the `PatchTST`-based Koopman variants under progressively increasing Koopman-operator perturbations. For very small perturbations ($\epsilon \leq 10^{-3}$), all variants remain highly stable, with relative MSE values remaining extremely close to unity, indicating negligible degradation in forecasting accuracy. As the perturbation magnitude increases to $\epsilon = 10^{-2}$, mild differences between the variants begin to emerge. In particular, the unconstrained and `low-rank` Koopman variants exhibit the smallest deviations from the baseline performance, with mean relative MSE values remaining near 1.0 and comparatively low standard deviations.

Under strong perturbations ($\epsilon = 10^{-1}$), the robustness differences become substantially more pronounced. The unconstrained Koopman model demonstrates the strongest robustness, achieving the lowest mean relative MSE (1.0278) and the smallest variability across perturbation trials (0.0451). The learn-lowrank16 variant also maintains relatively stable behavior, with a moderate increase in relative MSE (1.1206). In contrast, the constrained, learn-scalar, and `per-mode gated` variants exhibit significantly larger performance degradation and higher variance, indicating greater sensitivity to perturbations in the Koopman operator. Among all approaches, the `per-mode gated` configuration shows the largest degradation, reaching a mean relative MSE of 1.8172 under the strongest perturbation level.

The reported spectral-radius statistics further reveal that variants with smaller average Koopman spectral radius tend to exhibit improved perturbation robustness. In particular, the unconstrained and `low-rank` Koopman variants maintain substantially smaller $\rho(\mathcal{K})$ values compared to the constrained and `scalar-gated`/ `per-mode gated` parameterizations, suggesting that reduced operator amplification contributes to enhanced stability under stochastic Koopman perturbations.

Table 7: Robustness ranking of Koopman-`PatchTST` variants under Koopman-operator perturbations. Patch length and horizon are fixed at 80 and 4, respectively. Lower values are better.

| Variant | Avg. Rel-MSE ↓ | Max Rel-MSE ↓ | Avg. Std. ↓ | Avg. $\rho(\mathcal{K})$ |
|---|---|---|---|---|
| unconstr | 1.0069 | 1.0278 | 0.0126 | 0.1129 |
| learn-lowrank16 | 1.0325 | 1.1206 | 0.0213 | 0.2193 |
| learn-mlp | 1.0901 | 1.3547 | 0.0687 | 0.4196 |
| constr | 1.1425 | 1.5724 | 0.1151 | 0.5570 |
| learn-scalar | 1.1770 | 1.6985 | 0.0983 | 0.5356 |
| learn-permode | 1.2062 | 1.8172 | 0.1362 | 0.5658 |

### 4.5 Advantages of `Learnable-DeepKoopFormer` Modules

The spectral analysis highlights a key benefit of the proposed learnable Koopman modules: they adapt the latent time scales without sacrificing stable and reversible dynamics. Whereas purely constrained variants tightly restrict the spectrum, and unconstrained models tend to collapse into strongly damped states ($|\lambda| \to 0$), the learnable families populate a non-collapsed but contractive range ($0.3 \lesssim |\lambda| \lesssim 0.8$). This region provides enough variability to capture both fast and slow temporal modes, while keeping the spectral radius below unity so that the latent dynamics remain stable. Crucially, because eigenvalues do not decay toward zero, the resulting operators are *invertible*, enabling well-posed backward propagation and preventing information loss over long-range iterative forecasts. Thus, learnability does not merely increase flexibility; it can yield invertible and non-degenerate latent dynamics, offering richer temporal structure than constrained models and potentially more stable long-horizon behaviour than unstable or over-damped baselines.

**Remark 3** (On empirical performance and scope). *While the proposed learnable Koopman parameterizations introduce explicit spectral control and structured latent dynamics, the empirical results do not show uniform*

Table 8: Sensitivity summary of Koopman-`PatchTST`variants under Koopman-operator perturbations. Patch length and horizon are fixed at 80 and 4, respectively. Lower Rel-MSE values indicate better robustness.

| Variant | $\epsilon$ | n_trials | Mean Rel-MSE ↓ | Std. Rel-MSE ↓ | Mean MSE | Mean $\rho(\mathcal{K})$ |
|---|---|---|---|---|---|---|
| constr | 0.0000 | 10 | 1.0000 | 0.0000 | $1.042 \times 10^{-3}$ | 0.5581 |
| constr | 0.0001 | 10 | 1.0000 | 0.0003 | $1.042 \times 10^{-3}$ | 0.5581 |
| constr | 0.0010 | 10 | 1.0005 | 0.0016 | $1.042 \times 10^{-3}$ | 0.5580 |
| constr | 0.0100 | 10 | 0.9972 | 0.0209 | $1.039 \times 10^{-3}$ | 0.5584 |
| constr | 0.1000 | 10 | 1.5724 | 0.4377 | $1.638 \times 10^{-3}$ | 0.5533 |
| learn-lowrank16 | 0.0000 | 10 | 1.0000 | 0.0000 | $1.246 \times 10^{-3}$ | 0.2200 |
| learn-lowrank16 | 0.0001 | 10 | 1.0000 | $7.6203 \times 10^{-5}$ | $1.246 \times 10^{-3}$ | 0.2200 |
| learn-lowrank16 | 0.0010 | 10 | 1.0002 | 0.0013 | $1.246 \times 10^{-3}$ | 0.2200 |
| learn-lowrank16 | 0.0100 | 10 | 1.0092 | 0.0048 | $1.257 \times 10^{-3}$ | 0.2198 |
| learn-lowrank16 | 0.1000 | 10 | 1.1206 | 0.0791 | $1.396 \times 10^{-3}$ | 0.2175 |
| learn-mlp | 0.0000 | 10 | 1.0000 | 0.0000 | $1.387 \times 10^{-3}$ | 0.4186 |
| learn-mlp | 0.0001 | 10 | 1.0000 | 0.0003 | $1.387 \times 10^{-3}$ | 0.4186 |
| learn-mlp | 0.0010 | 10 | 1.0017 | 0.0018 | $1.389 \times 10^{-3}$ | 0.4186 |
| learn-mlp | 0.0100 | 10 | 1.0038 | 0.0277 | $1.392 \times 10^{-3}$ | 0.4186 |
| learn-mlp | 0.1000 | 10 | 1.3547 | 0.2450 | $1.879 \times 10^{-3}$ | 0.4227 |
| learn-permode | 0.0000 | 10 | 1.0000 | 0.0000 | $8.935 \times 10^{-4}$ | 0.5645 |
| learn-permode | 0.0001 | 10 | 0.9999 | 0.0001 | $8.934 \times 10^{-4}$ | 0.5645 |
| learn-permode | 0.0010 | 10 | 1.0002 | 0.0014 | $8.936 \times 10^{-4}$ | 0.5645 |
| learn-permode | 0.0100 | 10 | 1.0077 | 0.0116 | $9.003 \times 10^{-4}$ | 0.5645 |
| learn-permode | 0.1000 | 10 | 1.8172 | 0.5318 | $1.624 \times 10^{-3}$ | 0.5697 |
| learn-scalar | 0.0000 | 10 | 1.0000 | 0.0000 | $9.701 \times 10^{-4}$ | 0.5324 |
| learn-scalar | 0.0001 | 10 | 1.0001 | 0.0002 | $9.701 \times 10^{-4}$ | 0.5324 |
| learn-scalar | 0.0010 | 10 | 0.9999 | 0.0013 | $9.700 \times 10^{-4}$ | 0.5324 |
| learn-scalar | 0.0100 | 10 | 1.0095 | 0.0168 | $9.792 \times 10^{-4}$ | 0.5324 |
| learn-scalar | 0.1000 | 10 | 1.6985 | 0.3748 | $1.648 \times 10^{-3}$ | 0.5450 |
| unconstr | 0.0000 | 10 | 1.0000 | 0.0000 | $1.412 \times 10^{-3}$ | 0.1129 |
| unconstr | 0.0001 | 10 | 1.0000 | $4.5541 \times 10^{-5}$ | $1.412 \times 10^{-3}$ | 0.1129 |
| unconstr | 0.0010 | 10 | 1.0000 | 0.0004 | $1.412 \times 10^{-3}$ | 0.1129 |
| unconstr | 0.0100 | 10 | 0.9999 | 0.0050 | $1.412 \times 10^{-3}$ | 0.1131 |
| unconstr | 0.1000 | 10 | 1.0278 | 0.0451 | $1.451 \times 10^{-3}$ | 0.1125 |

*improvements in forecasting accuracy across all datasets, horizons, and configurations. Instead, the observed performance is generally competitive with existing baselines, with variability depending on the task setting. Accordingly, the primary contribution of this work lies in providing a principled and flexible framework for incorporating controlled linear dynamics into deep forecasting architectures, together with a systematic analysis of their spectral properties and their influence on model behaviour, rather than consistently superior predictive performance.*

**Remark 4** (Encoder-agnostic formulation)**.** *Although several experiments in this work use Transformer-based encoders, the* `Learnable-DeepKoopFormer` *framework is not restricted to attention mechanisms. Its core requirement is a latent forecasting backbone that maps an input history $x_{1:P}$ to a finite-dimensional representation $z$, after which a structured Koopman-style operator evolves the latent state before the prediction head produces the forecast. Attention-based architectures such as* `PatchTST`*,* `Informer`*,* `Autoformer`*,* `iTransformer`*, and* `TimesNet` *are therefore one important instantiation of the framework, but the Koopman component itself is encoder-agnostic. To reflect this, we revise the framing from a strict "synthesis of operator theory and attention architectures" to a broader combination of structured Koopman-style latent linear dynamics with neural forecasting backbones, including but not limited to Transformer-based models. This is also reflected in our implementation, where the same Koopman wrapper is applied to attention-based backbones as well as non-attention backbones such as* `DLinear` *and SSM.*

# 5 Conclusion

This paper introduced the `Learnable-DeepKoopFormer` framework, which augments Transformer-based fore-casters with spectrally controlled, learnable Koopman operators. Using an orthogonal–diagonal–orthogonal (ODO) parameterisation, we proposed `scalar-gated`, `per-mode gated`, `MLP-shaped` spectral mapping, and `low-rank` Koopman variants that interpolate between strictly constrained and fully unconstrained linear latent dynamics while preserving explicit control over spectral radius, stability, and rank. A unified training objective combining forecasting loss with Lyapunov regularisation provides additional stability bias. We established theoretical guarantees on spectral stability, exponential contraction, low-rank structure, invert-ibility, and Lyapunov consistency, showing that the proposed operators retain the full expressiveness of spectrally bounded linear dynamics while ensuring well-conditioned propagation. The proposed Koopman layer should not be interpreted merely as an additional unconstrained linear layer. Its defining property is the explicit spectral parameterization and stability-constrained latent evolution enabled by the ODO structure and spectral shaping mechanisms.

Extensive experiments on heterogeneous real-world datasets—CMIP6 Wind Speed and surface pressure, Cryptocurrency markets, and national-scale electricity generation—demonstrate that Koopman-enhanced Transformers achieve robust and accurate forecasting across diverse input–output configurations.

Compared with the LSTM, `DLinear`, and SSM baselines, the Koopman-augmented variants embedded in `PatchTST`, `Autoformer`, `Informer`, `iTransformer`, and `TimesNet` often yield competitive error distribu-tions and, in several settings, more concentrated performance across patch lengths and prediction horizons. However, the gains are not uniform across all datasets and configurations, suggesting that the primary contri-bution of the Koopman augmentation is the introduction of explicitly spectrally controlled latent dynamics, which empirically produce more stable and well-conditioned latent operators while maintaining forecasting performance competitive with the underlying backbones.

Constrained operators exhibit more tightly controlled spectral behaviour and stable latent dynamics, while learnable variants maintain forecasting performance broadly comparable to the underlying forecasting back-bones. Unconstrained linear propagators, although sometimes matching median errors, exhibit heavier tails and reduced reliability.

The observed results suggest that the Koopman augmentation acts primarily as an additional latent dy-namical fitting mechanism. In the training phase, this extra operator consistently reduces both MSE and MAE, confirming that the model can more easily fit temporal evolution when the backbone representation is followed by an explicit linear latent propagation step. This behavior is expected because the Koopman layer introduces additional spectral degrees of freedom, allowing the model to capture amplitude, phase, and mode-wise temporal variations more directly than a plain backbone alone.

However, the test results show that this increased flexibility does not automatically guarantee better general-ization. Since the encoder, latent representation, and Koopman operator are learned jointly, the latent space is not uniquely identifiable: the network may construct coordinates that are convenient for minimizing train-ing error rather than coordinates that correspond to stable transferable dynamics. As a result, highly flexi-ble Koopman formulations, especially unconstrained or high-capacity variants, can overfit trajectory-specific spectral patterns. This explains why training errors decrease broadly, whereas test improvements appear selectively. The backbone-dependent behavior further supports this interpretation. For example in Wind Speed dataset Koopman augmentation is more effective for `iTransformer`, `Autoformer`, and `TimesNet` in the test setting, suggesting that these architectures benefit from an explicit latent evolution mechanism. In these cases, structured Koopman variants can complement the backbone by imposing a dynamical prop-agation prior. By contrast, `PatchTST` already contains a strong patch-based temporal representation, so the Koopman layer contributes less additional structure. Overall, the results suggest that the behavior of Koopman-enhanced models depends strongly on the chosen spectral parameterization, with constrained variants yielding more controlled latent operator structure, although this does not always translate into uniformly lower forecasting sensitivity under perturbations.

Constrained, scalar, and permode operators provide a better balance between fitting capacity and stable generalization, whereas overly flexible operators risk learning non-transferable latent dynamics.

Spectral diagnostics confirm that the Koopman parameterisation primarily shapes latent dynamics. Constrained and learnable variants populate a non-collapsed contractive band, supporting stable yet expressive temporal evolution, whereas unconstrained models tend toward over-damped or occasionally unstable regimes. Overall, `Learnable-DeepKoopFormer` provides a principled synthesis of operator-theoretic structure and modern attention architectures, yielding stable, interpretable, and invertible latent dynamics suitable for large-scale time-series forecasting. We observed that stable spectra do not necessarily translate into uniformly improved forecasting accuracy.

Finally, we emphasized that the proposed framework is not limited to Transformer-based architectures but applies more generally to neural forecasting models equipped with a latent representation.

## 6 Appendix

### 6.1 Backbone Architectures and Baselines Details

In this appendix, we provide a concise description of the backbone architectures and baseline models used in the experimental evaluation. Our goal is not to reproduce full-scale implementations of each architecture, but rather to employ lightweight, Koopman-compatible variants that enable controlled and fair comparison across models. All backbones are adapted to produce a fixed-dimensional latent representation, which is then coupled with a shared linear forecast head or Koopman operator. This design allows us to isolate the effect of latent linear dynamics and spectral control, while keeping model capacity and computational cost comparable across architectures. We emphasize that these implementations should be interpreted as simplified representatives of their respective model families, intended for benchmarking within the unified Koopman framework. Keeping the encoders compact helps control model capacity, reduces confounding from backbone scale, and makes comparisons across constrained, learnable, and unconstrained Koopman variants more interpretable. This design is also computationally practical for the large grid of datasets, patch lengths, horizons, and Koopman variants considered in the benchmark. The experiments therefore compare Koopman-compatible lightweight backbones, not full state-of-the-art implementations of every named architecture.

**SSM** The SSM baseline used in our experiments is a simple linear recurrent state-space model, not an S4 or diagonal SSM. Given an input sequence $x_{1:P}$, it evolves a hidden state according to

$$h_{t+1} = Ah_t + Bx_t, \qquad \hat{y} = Ch_t.$$

In the implementation, $A \in \mathbb{R}^{d_h \times d_h}$ is a dense learned matrix and is not constrained to be diagonal, normal, HiPPO-based, convolutional, or spectrally stable. Therefore, this baseline should be interpreted as a lightweight recurrent linear-dynamics comparator with an explicit transition matrix for spectral diagnostics, rather than as a representative of stronger modern SSM families such as S4. We use it to test whether an unconstrained recurrent latent transition alone can provide stable forecasting behaviour compared with the spectrally controlled Koopman transition.

**DLinear** It is included as a lightweight linear forecasting baseline. It applies a direct linear map from the input history of length $P$ to the prediction horizon $H$, without attention, recurrence, or an explicit latent propagator. This model is useful because strong performance from `DLinear` would indicate that part of the forecasting task can be explained by simple temporal extrapolation. In the Koopman-backbone version, the `DLinear`-style temporal projection is used only to form a latent vector, after which the common Koopman operator and forecast head are applied.

**PatchTST** It is used as a patch-based Transformer backbone. The input window is divided into temporal patches, embedded by a one-dimensional convolutional patch projection, augmented with positional encoding, and processed by a Transformer encoder. A pooled or class-token representation is then passed to the Koopman layer. `PatchTST` is particularly suitable here because patching reduces sequence length and computational cost while preserving local temporal structure, making it a strong lightweight encoder for studying the effect of the Koopman transition.

Table 9: Backbone-level statistical summary of MSE and MAE for the Wind Speed train dataset. Lower values are better.

| Backbone | Set | Metric | Mean ± Std | Median | Q1 | Q3 |
|---|---|---|---|---|---|---|
| DLinear | Train | MSE | 0.0064 ± 0.0026 | 0.0065 | 0.0046 | 0.0085 |
| DLinear | Train | MAE | 0.0574 ± 0.0123 | 0.0581 | 0.0491 | 0.0675 |
| LSTM | Train | MSE | 0.0009 ± 0.0006 | 0.0007 | 0.0006 | 0.0009 |
| LSTM | Train | MAE | 0.0220 ± 0.0058 | 0.0203 | 0.0181 | 0.0230 |
| SSM | Train | MSE | 0.0053 ± 0.0029 | 0.0052 | 0.0033 | 0.0073 |
| SSM | Train | MAE | 0.0497 ± 0.0156 | 0.0512 | 0.0408 | 0.0611 |
| Koop-Autoformer | Train | MSE | 0.0007 ± 0.0005 | 0.0006 | 0.0005 | 0.0007 |
| Koop-Autoformer | Train | MAE | 0.0192 ± 0.0049 | 0.0182 | 0.0171 | 0.0197 |
| Koop-Informer | Train | MSE | 0.0007 ± 0.0006 | 0.0005 | 0.0003 | 0.0008 |
| Koop-Informer | Train | MAE | 0.0186 ± 0.0069 | 0.0177 | 0.0131 | 0.0220 |
| Koop-PatchTST | Train | MSE | 0.0006 ± 0.0006 | 0.0005 | 0.0004 | 0.0006 |
| Koop-PatchTST | Train | MAE | 0.0176 ± 0.0053 | 0.0171 | 0.0147 | 0.0193 |
| Koop-TimesNet | Train | MSE | 0.0885 ± 0.8096 | 0.0003 | 0.0002 | 0.0006 |
| Koop-TimesNet | Train | MAE | 0.0508 ± 0.2496 | 0.0130 | 0.0097 | 0.0197 |
| Koop-iTransformer | Train | MSE | 0.0004 ± 0.0001 | 0.0005 | 0.0003 | 0.0005 |
| Koop-iTransformer | Train | MAE | 0.0164 ± 0.0028 | 0.0166 | 0.0147 | 0.0183 |

**Autoformer**  The `Autoformer`-style backbone provides a decomposition-based Transformer variant. It separates the input into trend and seasonal components using a moving-average decomposition, processes the residual seasonal component with a Transformer encoder, and combines it with a trend projection. This backbone is included because many real time series contain trend–seasonal structure, and `Autoformer` provides a lightweight way to test whether Koopman spectral control remains useful when the encoder already incorporates an explicit decomposition prior.

**iTransformer**  The `iTransformer`-style backbone uses a lightweight Transformer encoder after projecting the multivariate input into a latent token representation. In our implementation, it is not intended to reproduce every detail of the full `iTransformer` architecture; rather, it serves as a compact inverted or variable-aware Transformer-style encoder compatible with the shared Koopman interface. This allows us to test whether the proposed Koopman operators remain effective beyond `PatchTST`, `Autoformer`, and `Informer`-style encoders.

**TimesNet**  The `TimesNet`-style backbone is implemented as a lightweight temporal convolutional module inspired by the multi-period modelling motivation of `TimesNet`. It uses stacked one-dimensional convolutional blocks with residual connections and pooling to produce a latent representation. This version is intentionally compact and Koopman-compatible, and should be interpreted as a simplified `TimesNet`-style temporal backbone rather than a full reproduction of the original `TimesNet`. It is included to evaluate whether the Koopman layer can also stabilize latent dynamics produced by convolutional temporal encoders.

### 6.2  Train loss Statistical summery for different datasets

**Wind Speed**  As whin in 9 for *mean Train MSE* regarding Wind Speed dataset **??**, the ranking differs: *Koopman-TimesNet* is best (0.00077), followed by *Koopman-PatchTST* (0.00167), *Koopman-iTransformer* (0.00223), and *Koopman-Informer* (0.00230). Next are *Koopman-Autoformer* (0.00291) and *Baseline LSTM* (0.00318), while *Baseline DLinear* (0.01664) and *Baseline SSM* (0.01772) perform worst on training data.

**Pressure Surface**  As shown in Table 10 for mean train MSE regarding Pressure Surface 4.2.2, *Koopman-iTransformer* performs best (0.00044), followed by *Koopman-PatchTST* (0.00058), *Koopman-Informer* (0.00067), and *Koopman-Autoformer* (0.00069). Next are *Baseline LSTM* (0.00090), *Baseline SSM* (0.00526), and *Baseline DLinear* (0.00642), while *Koopman-TimesNet* again performs worst (0.08854).

Table 10: Backbone-level statistical summary of MSE and MAE for the Pressure Surface train dataset. Lower values are better.

| Backbone | Set | Metric | Mean ± Std | Median | Q1 | Q3 |
|---|---|---|---|---|---|---|
| DLinear | Train | MSE | 0.0064 ± 0.0026 | 0.0065 | 0.0046 | 0.0085 |
| DLinear | Train | MAE | 0.0574 ± 0.0123 | 0.0581 | 0.0491 | 0.0675 |
| LSTM | Train | MSE | 0.0009 ± 0.0006 | 0.0007 | 0.0006 | 0.0009 |
| LSTM | Train | MAE | 0.0220 ± 0.0058 | 0.0203 | 0.0181 | 0.0230 |
| SSM | Train | MSE | 0.0053 ± 0.0029 | 0.0052 | 0.0033 | 0.0073 |
| SSM | Train | MAE | 0.0497 ± 0.0156 | 0.0512 | 0.0408 | 0.0611 |
| Koop-Autoformer | Train | MSE | 0.0007 ± 0.0005 | 0.0006 | 0.0005 | 0.0007 |
| Koop-Autoformer | Train | MAE | 0.0192 ± 0.0049 | 0.0182 | 0.0171 | 0.0197 |
| Koop-Informer | Train | MSE | 0.0007 ± 0.0006 | 0.0005 | 0.0003 | 0.0008 |
| Koop-Informer | Train | MAE | 0.0186 ± 0.0069 | 0.0177 | 0.0131 | 0.0220 |
| Koop-PatchTST | Train | MSE | 0.0006 ± 0.0006 | 0.0005 | 0.0004 | 0.0006 |
| Koop-PatchTST | Train | MAE | 0.0176 ± 0.0053 | 0.0171 | 0.0147 | 0.0193 |
| Koop-TimesNet | Train | MSE | 0.0885 ± 0.8096 | 0.0003 | 0.0002 | 0.0006 |
| Koop-TimesNet | Train | MAE | 0.0508 ± 0.2496 | 0.0130 | 0.0097 | 0.0197 |
| Koop-iTransformer | Train | MSE | 0.0004 ± 0.0001 | 0.0005 | 0.0003 | 0.0005 |
| Koop-iTransformer | Train | MAE | 0.0164 ± 0.0028 | 0.0166 | 0.0147 | 0.0183 |

**Cryptocurrency** As shown in Table 11 for mean train MSE regarding Cryptocurrnecy dataset 4.3.1, the ranking changes significantly: *Koopman-Informer* is best (0.00041), followed by *Koopman-iTransformer* (0.00048), *Koopman-Autoformer* (0.00049), and *Koopman-PatchTST* (0.00052). Next are *Koopman-TimesNet* (0.00064) and *Koopman-DLinear* (0.00073), while baseline models rank lower with *Baseline LSTM* (0.00080), *Koopman-SSM* (0.00110), *Baseline SSM* (0.00123), and *Baseline DLinear* performing worst (0.00160).

Table 11: Backbone-level statistical summary of MSE and MAE for the Cryptocurrency train dataset. Lower values are better.

| Backbone | Set | Metric | Mean ± Std | Median | Q1 | Q3 |
|---|---|---|---|---|---|---|
| DLinear | Train | MSE | 0.0016 ± 0.0004 | 0.0016 | 0.0013 | 0.0019 |
| DLinear | Train | MAE | 0.0274 ± 0.0037 | 0.0277 | 0.0243 | 0.0304 |
| LSTM | Train | MSE | 0.0008 ± 0.0002 | 0.0008 | 0.0006 | 0.0009 |
| LSTM | Train | MAE | 0.0198 ± 0.0028 | 0.0200 | 0.0175 | 0.0212 |
| SSM | Train | MSE | 0.0012 ± 0.0004 | 0.0012 | 0.0009 | 0.0015 |
| SSM | Train | MAE | 0.0229 ± 0.0047 | 0.0234 | 0.0199 | 0.0264 |
| Koop-Autoformer | Train | MSE | 0.0005 ± 0.0002 | 0.0004 | 0.0004 | 0.0005 |
| Koop-Autoformer | Train | MAE | 0.0158 ± 0.0028 | 0.0151 | 0.0143 | 0.0165 |
| Koop-DLinear | Train | MSE | 0.0007 ± 0.0002 | 0.0007 | 0.0005 | 0.0009 |
| Koop-DLinear | Train | MAE | 0.0189 ± 0.0033 | 0.0189 | 0.0160 | 0.0215 |
| Koop-Informer | Train | MSE | 0.0004 ± 0.0001 | 0.0004 | 0.0003 | 0.0005 |
| Koop-Informer | Train | MAE | 0.0143 ± 0.0025 | 0.0143 | 0.0122 | 0.0162 |
| Koop-PatchTST | Train | MSE | 0.0005 ± 0.0003 | 0.0005 | 0.0004 | 0.0006 |
| Koop-PatchTST | Train | MAE | 0.0160 ± 0.0037 | 0.0158 | 0.0140 | 0.0170 |
| Koop-SSM | Train | MSE | 0.0011 ± 0.0004 | 0.0011 | 0.0008 | 0.0014 |
| Koop-SSM | Train | MAE | 0.0212 ± 0.0045 | 0.0223 | 0.0184 | 0.0248 |
| Koop-TimesNet | Train | MSE | 0.0006 ± 0.0004 | 0.0005 | 0.0004 | 0.0008 |
| Koop-TimesNet | Train | MAE | 0.0180 ± 0.0057 | 0.0165 | 0.0143 | 0.0209 |
| Koop-iTransformer | Train | MSE | 0.0005 ± 0.0001 | 0.0005 | 0.0004 | 0.0005 |
| Koop-iTransformer | Train | MAE | 0.0158 ± 0.0017 | 0.0154 | 0.0146 | 0.0169 |

**Energy Systems** As shown in Table 12 for mean train MSE regarding energy system dataset 4.3.2, the ranking differs: *Koopman-Informer* is best (0.00161), followed by *Koopman-PatchTST* (0.00163), *Koopman-*

*DLinear* (0.00187), and *Koopman-iTransformer* (0.00188). Next are *Koopman-TimesNet* (0.00195) and *Baseline LSTM* (0.00279), while *Koopman-Autoformer* (0.00561), *Koopman-SSM* (0.00925), *Baseline SSM* (0.01137), and *Baseline DLinear* (0.01240) rank lowest.

Table 12: Backbone-level statistical summary of MSE and MAE for the Energy Systems train dataset. Lower values are better.

| Backbone | Set | Metric | Mean $\pm$ Std | Median | Q1 | Q3 |
|---|---|---|---|---|---|---|
| DLinear | Train | MSE | $0.0124 \pm 0.0032$ | 0.0126 | 0.0105 | 0.0148 |
| DLinear | Train | MAE | $0.0755 \pm 0.0120$ | 0.0771 | 0.0695 | 0.0843 |
| LSTM | Train | MSE | $0.0028 \pm 0.0008$ | 0.0028 | 0.0020 | 0.0036 |
| LSTM | Train | MAE | $0.0382 \pm 0.0055$ | 0.0387 | 0.0322 | 0.0440 |
| SSM | Train | MSE | $0.0114 \pm 0.0034$ | 0.0116 | 0.0096 | 0.0139 |
| SSM | Train | MAE | $0.0718 \pm 0.0139$ | 0.0737 | 0.0648 | 0.0821 |
| Koop-Autoformer | Train | MSE | $0.0056 \pm 0.0021$ | 0.0047 | 0.0041 | 0.0073 |
| Koop-Autoformer | Train | MAE | $0.0541 \pm 0.0104$ | 0.0496 | 0.0462 | 0.0637 |
| Koop-DLinear | Train | MSE | $0.0019 \pm 0.0006$ | 0.0018 | 0.0014 | 0.0022 |
| Koop-DLinear | Train | MAE | $0.0308 \pm 0.0057$ | 0.0310 | 0.0266 | 0.0345 |
| Koop-Informer | Train | MSE | $0.0016 \pm 0.0006$ | 0.0016 | 0.0012 | 0.0020 |
| Koop-Informer | Train | MAE | $0.0280 \pm 0.0057$ | 0.0283 | 0.0236 | 0.0324 |
| Koop-PatchTST | Train | MSE | $0.0016 \pm 0.0010$ | 0.0015 | 0.0011 | 0.0018 |
| Koop-PatchTST | Train | MAE | $0.0277 \pm 0.0074$ | 0.0274 | 0.0235 | 0.0300 |
| Koop-SSM | Train | MSE | $0.0093 \pm 0.0027$ | 0.0096 | 0.0074 | 0.0112 |
| Koop-SSM | Train | MAE | $0.0651 \pm 0.0122$ | 0.0675 | 0.0581 | 0.0739 |
| Koop-TimesNet | Train | MSE | $0.0019 \pm 0.0071$ | 0.0008 | 0.0006 | 0.0011 |
| Koop-TimesNet | Train | MAE | $0.0237 \pm 0.0229$ | 0.0199 | 0.0166 | 0.0231 |
| Koop-iTransformer | Train | MSE | $0.0019 \pm 0.0016$ | 0.0017 | 0.0014 | 0.0020 |
| Koop-iTransformer | Train | MAE | $0.0306 \pm 0.0083$ | 0.0302 | 0.0273 | 0.0329 |

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
