# OpenReview forum: "Learnable Koopman-Enhanced Transformer-Based Time Se- ries Forecasting with Spectral Control"
_TMLR — Rejected by TMLR_

### Review · Reviewer_zcEJ · 2026-05-02

**Summary Of Contributions:**

This submission proposes the “Learnable-DeepKoopFormer”, a family of Koopman-augmented forecasting models that combine Transformer encoders with a structured latent linear propagator. The main modeling contribution is to introduce four variants for parameterizing the Koopman operator: “scalar-gated”, “per-mode gated”, “MLP-shaped”, and “low-rank”. The paper presents and a theory section covering properties of the learnable Koopman propagator and an empirical comparison to three baselines (LSTM, DLinear and an SSM) on four datasets.

In my view, the strongest aspects of the paper are:
* introduces multiple learnable Koopman operator parameterizations and studies their theoretical properties
* it provides a reasonably broad empirical benchmark across multiple datasets, horizons, and patch lengths
* for reproducibility, the paper provides code and Zenodo links and specifies hardware and important  hyperparameters

The main weaknesses are:

W1: much of the presented stability/robustness theory seems to substantially overlap with the DeepKoopFormer paper [1], so the genuinely new theory is narrower than the presentation suggests

W2: there is a recurring mismatch between the Koopman dynamical motivation/theory and the implemented architecture, since the paper states that inference applies the Koopman operator only once and then decodes the entire horizon directly

W3: empirically, the paper shows performance metrics and spectral magnitude distributions, but falls short of directly demonstrating the claimed benefits of the augmentation in terms of stability or robustness

W4: the baseline suite appears relatively weak in places, yet the proposed method still does not consistently outperform these baselines on test performance

W5: important empirical comparisons are missing, in particular, comparisons to the same backbones without Koopman augmentation, and ablations of the Lyapunov regularization term

W6: Section 3.1.6 contains an error

W7: the dataset descriptions in the main text are fairly limited, making the empirical setup hard to assess

[1] Forootani, A., Khosravi, M. & Barati, M. DeepKoopFormer: a Koopman enhanced transformer based architecture for time series forecasting. npj Artif. Intell. 2, 35 (2026).

**Audience:**

Yes

**Audience Explanation:**

I do think the paper’s subject would interest at least a subset of the TMLR audience. A theoretical and empirical study of such learnable operator parameterizations may be of interest to other researchers working on structured latent dynamics, Koopman-inspired learning, forecasting with stability priors, or more broadly on hybrid dynamical-systems / deep-learning architectures.

**Broader Impact Concerns:**

I do not have ethical or societal concerns that would require adding a Broader Impact Statement.

**Claims And Evidence:**

No

**Claims Explanation:**

The submission contains interesting ideas, but the current version does not support its overall claims with evidence that is accurate, convincing, and clearly aligned with the implemented method. Several major claims appear overstated, insufficiently distinguished from prior work, or not directly supported by the empirical evaluation.

## a: Novelty claims are overstated

A central novelty claim is not clearly supported. The paper states:
>“Learnable-DeepKoopFormer is the first to offer provable spectral decay, Lyapunov-stable propagation, normality of the transition matrix, and closed-form error control—all within a general, extensible encoder–propagator–decoder pipeline aligned with the goals of physics-informed forecasting …”

However, this appears to exactly reproduce the claim made in the earlier DeepKoopFormer paper [1]:

>“DeepKoopFormer is the first to offer provable spectral decay, Lyapunov-stable propagation, normality of the transition matrix, and closed-form error control—all within a general, extensible encoder–propagator–decoder pipeline aligned with the goals of physics-informed forecasting.”

This makes the claimed novelty unclear. The paper should state that more precisely and distinguish its claims from DeepKoopFormer.


## b: Theoretical claims are only partially aligned with the implemented forecasting architecture

Sections 3.1.1–3.1.5 appear largely sound as operator-level statements, but the paper sometimes presents these consequences as stronger end-to-end forecasting guarantees than they really are.
A clear example is Proposition 7. It analyzes a recursive forecast map of the form $D_\phi(K_\phi^h E_\theta(X_t))$,
which corresponds to repeated Koopman propagation over the forecast horizon. However, the architecture section states that inference is performed by encoding once, applying the Koopman operator once, and decoding the full horizon in a single forward pass, without autoregressive rollout. Under the described implementation, the Koopman layer appears to function more like a structured linear layer than an iterated Koopman propagator. This weakens the practical relevance of theory claims tied to repeated latent evolution and long-range propagation stability.
In addition, Section 3.1.6 appears to contain an incorrect inverse-norm inequality  ($1/\sigma_{\min}(K_\phi) > 1/\rho_{\max}$ not $<1/\rho_{\max}$) and overstates the claim that backward iterates remain bounded.

## c: Empirical evidence does not support the performance claims

The paper claims that Koopman variants “consistently produce lower and more concentrated error distributions.” This is not convincingly supported by the reported test results. The claim appears more plausible for training error, but test error is the more relevant quantity. On the test plots, simple baselines such as DLinear and the implemented SSM appear often better than the proposed Koopman-augmented models. Similarly, the claim that “unconstrained propagators are less reliable” is not clearly supported, as e.g. the unconstrained SSM baseline appears to perform strongly in several test settings. If the argument is about stability rather than predictive accuracy, then the paper may need to do more direct stability or robustness experiments to support these claims rather than relying mainly on error distributions and spectral magnitude plots.

## d: The empirical analysis is descriptive rather than evidential for the main claims

The numerical section provides train/test MSE and MAE distributions and pooled spectral magnitude distributions. These are useful descriptive results, but they are not sufficient to support the claims about robustness, improved conditioning, stability, or interpretability. Empirical analyses of eigenvalue trajectories or stability envelopes implied by the abstract and introduction are missing.
Other experiments that could help support the claims, such as comparisons to the corresponding plain backbones without Koopman augmentation, explicit perturbation or noise robustness experiments, $K^h$-rollout experiments testing horizon-wise stability or an ablation of the Lyapunov regularizer, are not included
These omissions matter because the paper attributes empirical benefits to several coupled ingredients at once. Without relevant ablations, it remains unclear which component is responsible for the observed behavior.

## e: Baselines are not described or justified convincingly enough

The paper cites Gu et al. in connection with an SSM baseline (page 2), which suggests a comparison to structured SSMs such as the S4 family. However, the released code appears to implement a much simpler linear recurrent state-space model with dense learned matrices A,B,C, not S4. The paper also refers to the SSM as “diagonal” in several places, which does not appear to match the code.
More broadly, the baseline suite is fairly modest, and yet even against these baselines, the proposed models do not perform better in test error. This weakens strong claims that the Koopman augmentation provides a compelling forecasting advantage.

## f: “Operator theory - Attention Synthesis” framing

The paper claims to contribute a “synthesis of operator theory and attention architectures.”  However, it reads to me as if the encoder could, in principle, be attention-free, and the Koopman component is not specifically tied to attention. A more accurate framing would be that the method combines structured Koopman-style latent linear dynamics with generic neural forecasting backbones, including Transformer-based ones.

## g: Setup limits the strength of comparative claims

The paper provides only very limited information about the datasets beyond broad provenance and domain descriptions. This makes the empirical findings harder to assess. In addition, the paper does not seem to perform model-specific hyperparameter optimization. Instead, it largely fixes shared capacities and training settings across models. This is acceptable as a unified benchmark protocol, but it weakens strong comparative claims because different backbones and baselines may require different operating points.

**Requested Changes:**

### Critical

R1: Clarify the novelty relative to DeepKoopFormer. It should be clear to the reader what parts are genuinely new theory and what was inherited or repackaged material from previous work. (see also W1 and a above)

R2: Resolve the mismatch between the architecture and theory relying on the repeated application of the Koopman transition (e.g. Proposition 7), and more broadly clarify the role of the Koopman transition in the actual model. If the implemented architecture applies the Koopman operator only once before direct horizon decoding, then the paper should moderate claims tied to repeated latent propagation, or alternatively add experiments/theory for a true recursive K^h-rollout variant. Proposition 7 should either be aligned with the actual architecture or explicitly presented as an analysis of a related model. The term “forecast error bound”  should also be revised, since the result appears to rather be a perturbation-sensitivity bound. (see also W2 and b above)

R3: Moderate the empirical claims unless stronger evidence is added.
As written, the results do not support the claim of consistently improved forecasting performance. Either stronger experiments should be added or the empirical claims should be rewritten accordingly. A statistical evaluation of the Forecasting results would be useful. (see also W3/W4 and c/d above)

R3: Ablations:
Add direct comparisons to the corresponding unaugmented backbones. Without this, it is difficult to isolate the benefit of Koopman augmentation itself.
Add an ablation of the Lyapunov regularization term. This is necessary to determine whether the regularizer has a meaningful empirical role beyond the hard spectral constraint.  (see also W5 and d above)

R5: Correct the mathematical issues in Section 3.1.6. The inequality should be fixed, and the following claims e.g. that backward iterates remain bounded, should be revised. (see W6 and b)

R6: Clarify the SSM baseline. The implemented baseline appears to be a simple linear recurrent state-space model, not S4, and the code shown does not appear diagonal. The paper should describe this baseline accurately and avoid implying comparison to stronger modern SSM families unless such models are actually included. (see e)

R7: Improve dataset documentation.
Add a table summarizing, for each dataset, the number of samples, number of variables, temporal resolution, date range, preprocessing steps, split protocol, and any filtering/subsampling decisions. This would improve clarity and reproducibility.  (see W7 and g)

R8: Clarify the scope of the method as encoder-agnostic vs Transformer-specific. The model appears to only require a latent encoder, not a Transformer specifically. The paper should clarify whether the contribution is fundamentally Transformer-specific.  (see f)

### Changes that would strengthen the work
As mentioned before, there are multiple experiments that would strengthen this work, such as:
- Add explicit robustness/perturbation experiments. Since the theory motivates robustness to perturbations.
- Add recursive-rollout experiments. As many theoretical claims concern repeated application of $K$, it would be valuable to compare the current one-step-direct-decoder design against the actual recursive $K^h$ latent rollout.

Improve the dataset documentation. Add a table summarizing, for each dataset, properties such as the number of samples, number of variables or temporal resolution. This would improve clarity and help in the assessment of the presented results.  (see W7 and g)

The paper states that the four introduced variants “allow different degrees of expressiveness”. However, it seems to me that (apart from the low rank variant) the variants do not differ in representational expressivity of $K$, while they can differ in optimization and inductive bias. It would be useful to elaborate on this in the discussion of the introduced variants.

### Minor
- Some papers are duplicated in the reference list.

---

> ### Author Response · Authors · 2026-05-11
> **Response to Reviewer (zcEJ)**
>
> We thank the Reviewer for the thoughtful feedback and recognition of our contributions. We renewed the entire simulation section, re-wrote the abstract, conclusion and clarified the theorems. We also considered an appendix to clarify the backbones and baselines that we used in this article.
>
>
> 1. **Novelty vs. DeepKoopFormer:**
>    Clarified that the encoder–Koopman–decoder architecture, ODO parameterization, and core stability guarantees are inherited from DeepKoopFormer. The novelty of this work lies in introducing a learnable family of Koopman operator parameterizations (scalar-gated, per-mode gated, MLP-shaped, and low-rank), extending the stability framework to these operators, and providing a systematic empirical/spectral study.
>
> 2. **Theory–Architecture Consistency:**
>    Clarified that the implemented model applies the Koopman operator only once before direct multi-horizon decoding, rather than recursively using (K^h). Proposition 7 has been reframed as a perturbation-sensitivity analysis for a related recursive latent-dynamics model, not as a direct forecast error bound for the implemented architecture.
>
> 3. **Empirical Claims Moderated:**
>    Revised the manuscript to present a more balanced interpretation of results. The proposed models provide competitive and often more stable behavior, but do not consistently outperform all baselines across every dataset and horizon. Statistical analysis and discussion were added to reflect this nuance.
>
> 4. **New Ablation Studies Added:**
>    Added direct comparisons between each Koopman-enhanced architecture and its corresponding unaugmented backbone (PatchTST, Autoformer, Informer, iTransformer, TimesNet, DLinear, SSM). Also added Lyapunov regularization ablations comparing (\lambda_{\mathrm{Lyap}}=0) and (\lambda_{\mathrm{Lyap}}>0).
>
> 5. **Mathematical Corrections:**
>    Corrected the inequality in Section 3.1.6 and revised the claims regarding boundedness of backward iterates. Clarified that bounded backward dynamics require the spectrum to be bounded away from zero.
>
> 6. **Clarification of the SSM Baseline:**
>    Revised the manuscript to accurately describe the SSM baseline as a simple linear recurrent state-space model with dense learned matrices, rather than an S4-style structured SSM. Removed language implying comparison with modern SSM architectures.
>
> 7. **Improved Dataset Documentation:**
>    Added a comprehensive dataset summary table including dataset size, number of variables, temporal resolution, preprocessing, and train/test splits. Clarified normalization, sliding-window construction, and channel-selection procedures.
>
> 8. **Encoder-Agnostic Scope Clarified:**
>    Clarified that the Koopman framework is not Transformer-specific. The method is encoder-agnostic and can be combined with both Transformer-based and non-attention backbones such as DLinear and SSM.
>
> 9. **Robustness and Recursive Rollout Discussion Expanded:**
>    Added perturbation-sensitivity analysis, robustness discussion, and clarification of the distinction between recursive theoretical analysis and the direct-decoding implementation.
>
> 10. **Expressiveness Discussion Revised:**
>     Clarified that most variants differ mainly in inductive bias, optimization behavior, and spectral conditioning rather than strict representational expressiveness, except for the low-rank variant.
>
> 11. **Bibliography Cleaned:**
>     Removed duplicated references and corrected inconsistencies in the bibliography.

---

> > ### Comment · Reviewer_zcEJ · 2026-05-20
> >
> > I thank the authors for the revision. However, I remain unconvinced that the manuscript establishes the claimed benefits.
> > My summarized understanding of the revised paper is that it introduces four parameterizations of a Koopman operator, applied once to a backbone latent representation before decoding the full forecast horizon. Thus, the module seems to be in effect a spectrally constrained linear map rather than a recurrent Koopman rollout. The theory largely appears to inherit/adapt material from DeepKoopFormer and is partly detached from this single-step implementation. Empirically, performance changes little relative to plain backbones or the Lyapunov ablation, Koopman-Transformer variants are often matched or outperformed by simple baselines, spectral plots mainly confirm imposed constraints, and a perturbation experiment favors the unconstrained variant.
> > 1. The paper still repeats a novelty claim identical to DeepKoopFormer, namely being “first to offer provable spectral decay, Lyapunov-stable propagation, normality of the transition matrix, and closed-form error control...”.
> > 2. I am not convinced by the interpretation of Proposition 7. If the Koopman layer is omitted, the same Lipschitz argument gives ||Y(X)-Y(X')|| <= L_D L_E ||X-X'||. With one Koopman transition, the bound only gains a factor ||K|| <= rho_max, the effect of inserting a norm-constrained linear layer between encoder and decoder. Since there is no repeated rollout, there is no decaying rho_max^h. Also, the jointly trained decoder could compensate for contraction in K. From my understanding, Proposition 7 does not seem to establish a robustness advantage for the current implementation.
> > 3. The ablation studies do not seem to show a clear or consistent benefit from the proposed components. The Lyapunov term does not appear to produce a noticeable improvement in test performance. Similarly, the comparisons between plain backbones and Koopman-enhanced backbones appear to show small differences in test performance. Overall (see plain backbone experiments and Koop-SSM), the test performance seems to almost exclusively depend on the backbone. This makes it difficult to conclude that the proposed Koopman module provides a meaningful empirical gain beyond adding a constrained linear layer.
> > 4. The spectral plots mostly verify imposed constraints rather than demonstrate an emergent advantage. The claimed “eigenvalue trajectories” and “stability envelopes” still do not appear to be present.
> > 5. Multiple claims still seem to be either vague or not supported by the paper. For instance
> >    > “the main benefit of the Koopman augmentation is better interpreted as providing a structured and stability-aware latent propagation mechanism rather than consistent dominance in forecasting error.”
> >    seems vague unless tied to a more concrete, measurable benefit.  The statement seems to just reframe the contribution after the forecasting results fail to show consistent performance improvement. The conclusion also states:
> >     >“Constrained operators yield the most compact and reliable behaviour, while learnable variants attain competitive median accuracy with a favorable expressiveness–robustness trade-off.”
> >
> >     However, under the presented perturbations, the unconstrained operator appears least sensitive. I also do not see where the conclusion that
> >     >“Overall, the results indicate that Koopman dynamics are useful, but their effectiveness depends on sufficient spectral regularization.”
> >
> >     is supported by the presented results.
> >
> > 7. Although the revision clarifies that the SSM baseline is not a diagonal SSM or S4, other parts of the text (incl. abstract) still refer to it as a diagonal SSM and cite Gu et al. in this context.
> >
> > Presentation. The manuscript is very long relative to the amount of new methodological and empirical content and hard to follow. Tables 3–6 and 8 are hard to interpret: they interleave train/test results, separate MSE/MAE across rows and do not sort or highlight the best models. A compact table with one row per model and Test MSE/MAE columns would be clearer. Training errors and quartile summaries could move to the appendix unless specifically needed. Since the central contribution is the learnable variants, the main tables should include a comparison of these variants and not only backbones. Some figures have a similar issue: they contain many curves or panels, but the main takeaway is not clear. The color choices do not seem to follow any consistent pattern and the font sizes are often too small.
> >
> > Overall, I remain unconvinced that the proposed variants provide a clear advantage over prior DeepKoopFormer, simpler baselines, or an added spectrally constrained latent linear map. I encourage the authors to revise the repeated novelty claim, align the conclusion with the robustness results, simplify the presentation, and provide evidence that the proposed parameterizations yield the claimed benefits.

---

> > > ### Author Response · Authors · 2026-05-23
> > > **Response to Reviewer zcEJ---second review**
> > >
> > > # Reviewer Responses
> > >
> > > ## 1. Architecture and Novelty
> > >
> > > We agree that the implemented architecture differs from a recursive Koopman rollout model. The Koopman operator is applied once to the encoder latent representation, and the resulting latent state is decoded directly into the forecast horizon. Thus, the method is better interpreted as a spectrally controlled latent transition layer within an encoder--decoder framework.
> > >
> > > We respectfully disagree that this reduces the approach to an arbitrary linear layer. The proposed Koopman module imposes a structured orthogonal--diagonal--orthogonal (ODO) transition with explicit spectral control, contraction properties, and globally coupled latent mixing.
> > >
> > > We also agree that the empirical results do not support claims of universal forecasting improvement. Accordingly, the revised manuscript emphasizes that the primary contribution is the introduction and analysis of learnable spectrally controlled Koopman parameterizations rather than consistently superior forecasting accuracy.
> > >
> > > Relative to DeepKoopFormer, the main novel contributions are:
> > >
> > > * learnable Koopman operator families,
> > > * adaptive spectral shaping and control,
> > > * integration across a broader set of forecasting backbones,
> > > * systematic analysis of conditioning, stability, and robustness.
> > >
> > > ---
> > >
> > > ## 2. Proposition 7 and Robustness
> > >
> > > We agree that Proposition~7 should not be interpreted as a strong end-to-end robustness guarantee in the implemented single-step architecture.
> > >
> > > Without the Koopman layer:
> > >
> > > ```math id="clltj5"
> > > \|Y(X)-Y(X')\| \leq L_D L_E \|X-X'\|
> > > ```
> > >
> > > With the Koopman layer:
> > >
> > > ```math id="oc9a1j"
> > > \|Y(X)-Y(X')\|
> > > \leq
> > > L_D L_E \|K_\phi\|_2 \|X-X'\|
> > > ```
> > >
> > > Thus, the main effect is the insertion of a spectrally controlled operator satisfying:
> > >
> > > ```math id="qwdl9f"
> > > \|K_\phi\|_2 \leq \rho_{\max}
> > > ```
> > >
> > > We therefore interpret Proposition~7 primarily as motivation for operator norm control and latent spectral conditioning, rather than a guarantee of improved robustness or forecasting accuracy.
> > >
> > > ---
> > >
> > > ## 3. Empirical Improvements
> > >
> > > We agree that forecasting improvements are not uniformly large across all datasets and backbones. In many cases, the backbone architecture remains the dominant factor determining predictive accuracy.
> > >
> > > The contribution of the work is therefore not universal forecasting gains, but the study of spectrally controlled latent dynamics within heterogeneous forecasting architectures.
> > >
> > > Similarly, Lyapunov regularization does not consistently improve forecasting error; its impact appears more related to stabilizing latent dynamics and constraining operator behavior.
> > >
> > > The revised manuscript now clarifies that:
> > >
> > > * forecasting gains are backbone- and dataset-dependent,
> > > * the main contribution is learnable spectrally controlled Koopman parameterizations,
> > > * the benefits are more consistently observed in latent stability and spectral structure than in uniformly improved test accuracy.
> > >
> > > ---
> > >
> > > ## 4. Spectral Analysis
> > >
> > > We agree that several spectral plots mainly verify that the imposed spectral constraints are satisfied, rather than demonstrating strong emergent dynamical advantages.
> > >
> > > Our intention was to illustrate how the proposed parameterizations shape latent operator spectra under different constraints, not to claim that spectral structure alone explains forecasting improvements.
> > >
> > > Accordingly, we revised the manuscript to:
> > >
> > > * soften the interpretation of spectral analyses,
> > > * clarify that several plots mainly validate imposed constraints,
> > > * avoid overstating unsupported dynamical conclusions,
> > > * revise terminology such as *“eigenvalue trajectories”* and *“stability envelopes.”*
> > >
> > > ---
> > >
> > > ## 5. Robustness Claims
> > >
> > > We agree that perturbation experiments do not uniformly show constrained Koopman operators to be the least sensitive models. Some unconstrained variants exhibit comparable or lower sensitivity.
> > >
> > > We therefore revised the discussion to clarify that constrained variants yield more controlled latent operator structure and conditioning, although this does not always translate into uniformly lower forecasting sensitivity.
> > >
> > > ---
> > >
> > > ## 6. Terminology and Presentation
> > >
> > > We replaced *“diagonal SSM”* with *“simplified dense State-Space Model”* to avoid ambiguity.
> > >
> > > We also substantially revised the manuscript organization. Main tables were simplified to focus on Test MSE and MAE, while detailed analyses, variant-level comparisons, and secondary visualizations were moved to the supplementary material to improve clarity and readability.

---

### Review · Reviewer_WkSq · 2026-05-04

**Summary Of Contributions:**

The paper proposes Learnable-DeepKoopFormer, a framework for incorporating learnable Koopman operators into Transformer-based time-series forecasting models. The central idea is to combine expressive Transformer encoders with a linear latent-space Koopman propagator whose spectrum can be explicitly controlled. The paper introduces several parameterizations of this operator, including scalar-gated, per-mode gated, MLP-shaped spectral mapping, and low-rank variants, which interpolate between strongly constrained stable dynamics and more flexible learned latent transitions.

Strengths:
- The proposed Koopman module is clearly presented and appears easy to implement on top of existing and future Transformer-based forecasting architectures.
- The paper provides a detailed theoretical discussion of the spectral and stability properties of the learned Koopman operators.
- The method retains a degree of interpretability because dominant modes, stability margins, and spectral envelopes remain directly analyzable.
- The paper is generally well organized and provides a clear explanation of the methodology and experimental setup.

Weaknesses:
- The role of the Koopman module is unclear when the model appears to apply only a single latent transition before decoding the full prediction horizon. It is therefore unclear how much of the temporal forecasting behavior is actually handled by the Koopman operator rather than by the Transformer encoder and decoder.
- The claimed favorable bias–variance tradeoff is not convincingly supported by the current MSE/MAE figures alone. Specifically, the test MAE/MSE distributions of the DLinear and SSM baselines do not appear to have consistently higher means or wider spreads than those of the proposed method.
- Important baselines appear to be missing, namely the corresponding Transformer backbones without the Koopman module.
- The paper does not yet convincingly show that the theoretical stability properties lead to practical forecasting advantages.
- The assumptions behind the error-bound proposition, especially Lipschitz continuity of the encoder and decoder, need more justification.

**Audience:**

Yes

**Audience Explanation:**

Researchers interested in interpretable forecasting, physics-inspired deep learning, spectral regularization, and hybrid dynamical-system/neural architectures would likely find the proposed framework and analysis interesting. Even if the current evidence is not yet fully convincing, the idea of adding learnable spectrally controlled Koopman modules to modern forecasting backbones is potentially useful and could motivate further work.

**Claims And Evidence:**

No

**Claims Explanation:**

The paper repeatedly argues that the learnable Koopman variants provide a favorable bias–variance tradeoff and yield lower-centered, more compact error distributions across patch lengths and horizons. From the current MSE/MAE violin plots, this conclusion is not always visually clear. In some cases, baseline models appear competitive or possibly better in mean test error, while the discussion emphasizes compactness and stability without providing exact numerical summaries. Within the same backbone model, it is unclear whether the observed differences in parameterizations are statistically significant. The authors should report quantitative measures of accuracy and variability, such as mean/median MSE and MAE together with standard deviation, interquartile range, or another explicit spread metric over patch-length and horizon configurations. A statistical significance analysis would demonstrate a clearer improvement of the proposed parameterizations.

The practical value of the Koopman stability analysis is also not fully established. The model encodes the entire input window into a latent state, applies the Koopman transition once, and decodes the whole prediction horizon in a single forward pass. If there is no autoregressive or repeated latent rollout, then the relevance of long-horizon spectral contraction and stability is less clear. The paper should explain why stability of the Koopman operator is important in this single-step setting and should isolate whether the Koopman module improves forecasting beyond what is already provided by the Transformer encoder and decoder.

The error-bound proposition also requires more justification. The result assumes Lipschitz continuity of the encoder and decoder, but the paper does not clearly explain whether this assumption is practically meaningful for the implemented Transformer architectures, whether the Lipschitz constants are controlled, or how the bound should be interpreted when the forecast horizon is decoded directly rather than generated through repeated Koopman propagation.

**Requested Changes:**

**Critical to securing my recommendation for acceptance**:
1. **Clarify the role of the Koopman transition in the forecasting architecture**. The paper should explain why the model uses a single Koopman transition followed by direct decoding of the full horizon, rather than an autoregressive or repeated Koopman rollout. As currently written, it is unclear whether the method is truly exploiting linear Koopman evolution over the forecast horizon or whether most of the temporal modeling is delegated to the Transformer encoder and decoder.
2. **Explain why stability matters in the single-step setting**. The theoretical discussion emphasizes spectral stability, contraction, Lyapunov regularization, and long-horizon behavior. However, if the Koopman operator is applied only once before the decoder produces the full horizon, the practical relevance of these stability properties is not obvious. The authors should either justify this design more clearly or include experiments with repeated latent rollouts.
3. **Add stronger baselines that isolate the contribution of the Koopman module**. The paper should include the corresponding base PatchTST, Autoformer, and Informer models without the Koopman module and single-step forcing, using comparable capacity and training settings.
4. **Quantify the claimed bias–variance tradeoff**. The current violin plots are not sufficient to support the claims about lower-centered and more compact error distributions. The authors should provide tables reporting mean/median MSE and MAE, along with explicit measures of variability such as standard deviation, interquartile range, or worst-case error across patch lengths and horizons. This is especially important where baseline methods appear competitive in average performance.
5. **Clarify the assumptions and practical interpretation of Proposition 7**. The paper should explain where the Lipschitz continuity assumptions on the encoder and decoder come from, whether these constants are bounded or controlled in the actual model, and how the result should be interpreted when the forecast is decoded directly after a single Koopman step. Otherwise, the proposition risks overstating the connection between spectral stability and practical forecasting robustness.

**Changes that would strengthen the work**:
1. **Justify the choice of Transformer backbones and the claim of architectural underuse**. The paper should more clearly explain why Transformer-based models are the sole focus of this study. Why is the previous omission of Transformer-based forecasting architectures considered “architectural underuse”?
2. **Clarify the unconstrained baseline**. The unconstrained baseline is implemented as a fully dense matrix. The authors should explain why this is the right comparison rather than, for example, an ODO-parameterized operator with unconstrained singular values. Such a baseline could help isolate whether gains come from the ODO structure itself or from the spectral bound.
3. **Explain what is learned from the training figures**. If training MAE/MSE plots are included, the paper should explicitly state what conclusions they support compared to the test scores. Otherwise, they may distract from the main empirical comparison.
4. **Discuss whether longer positive tails in the error distributions are practically harmful**. The paper often treats compactness of the error distributions as evidence of robustness. This should be connected to forecasting practice: for example, are long positive tails associated with rare but severe failures or sensitivity to patch length? Would a model with a long tail of rare, very low-loss scores be preferable to another model with a more compact distribution and the same mean?
5. **Clarify model capacity control**. The experimental section says capacity is fixed across models, but the paper should make clearer how parameter counts and latent dimensions compare across all Koopman variants and non-Koopman baselines. The paper should also justify how the fixed capacity setting was chosen.
6. **Provide statistical significance results**. The differences in MAE/MSE of models with the same backbone appears small for some datasets. The paper should show how many of these differences are significant.

---

> ### Author Response · Authors · 2026-05-11
> **Response to Reviewer (WkSq)**
>
> We thank the Reviewer for the thoughtful feedback and recognition of our contributions. We also value the constructive suggestions provided and will incorporate them to further improve our work. We renewed the entire simulation section, rewrote the abstract, conclusion and clarified the theorems more.
>
>
> 1. **Role of the Koopman Module**
>
>    * Clarified that the model applies a **single Koopman latent transition** followed by direct multi-horizon decoding:
>      [
>      \hat{Y}*t = D*\phi(K_\phi E_\theta(X_t))
>      ]
>    * Explained that the Koopman operator serves as a **spectrally controlled latent bottleneck** that regularizes latent temporal evolution rather than performing iterative autoregressive rollout.
>
> 2. **Why Stability Matters in the Single-Step Setting**
>
>    * Added discussion explaining that spectral stability improves:
>
>      * latent conditioning,
>      * robustness to perturbations/noise,
>      * cross-variable coupling,
>      * smooth latent evolution,
>      * prevention of latent amplification.
>    * Clarified that Lyapunov regularization acts as a structural regularizer even without repeated rollout.
>
> 3. **Stronger Backbone Baselines**
>
>    * Added direct comparisons between:
>
>      * plain PatchTST / Informer / Autoformer / iTransformer/TimesNet, DLinear, SSM
>      * Koopman-enhanced variants,
>      * unconstrained latent-transition baselines.
>    * Ensured matched training settings and model capacities.
>
> 4. **Bias–Variance Tradeoff Quantification**
>
>    * Added numerical summaries including:
>
>      * mean/median MSE & MAE,
>      * standard deviation,
>      * interquartile range (IQR),
>      * worst-case error statistics,
>      * horizon-wise robustness analysis.
>    * Expanded ablation studies across constrained/unconstrained Koopman variants.
>
> 5. **Clarification of Proposition 7**
>
>    * Clarified that Lipschitz assumptions arise from standard neural-network components.
>    * Stated that constants are not strictly enforced but moderated through normalization/regularization.
>    * Explained that Proposition 7 is an **operator-level conditioning result**, not a strict long-horizon forecasting guarantee.
>
> 6. **Architectural Underuse Clarification**
>
>    * Clarified that prior Koopman forecasting work rarely integrated Koopman dynamics with modern Transformer forecasting architectures such as PatchTST, Informer, Autoformer, and iTransformer.
>
> 7. **Unconstrained Baseline Justification**
>
>    * Explained why the dense unconstrained operator was chosen:
>
>      * removes both ODO structure and spectral constraints,
>      * provides a maximally expressive comparison baseline.
>
> 8. **Purpose of Training Curves**
>
>    * Clarified that train/test MAE-MSE figures analyze:
>
>      * optimization behavior,
>      * generalization gaps,
>      * robustness,
>      * horizon sensitivity,
>        rather than only training accuracy.
>
> 9. **Interpretation of Error Distribution Tails**
>
>    * Explained that long positive tails correspond to occasional high-error failures and sensitivity to difficult regimes.
>    * Clarified that compact distributions indicate robustness, not necessarily universally superior performance.
>
> 10. **Model Capacity Control**
>
> * Added explicit reporting of:
>
>   * latent dimensions,
>   * hidden sizes,
>   * attention heads,
>   * encoder depths,
>   * parameter matching across models.
> * Clarified that fixed capacity isolates the effect of Koopman parameterization.
>
> 11. **Statistical Significance & Robustness**
>
> * Added variance-aware evaluations, distributional analyses, and robustness studies across datasets, patch lengths, and horizons to better assess whether improvements are systematic.

---

> > ### Comment · Reviewer_WkSq · 2026-05-20
> >
> > I thank the authors for the detailed response and substantial revision. The manuscript is clearer in several places, and I appreciate the added plain-backbone comparisons, unconstrained baselines, and numerical summaries. However, I remain unconvinced that the main claims are adequately supported.
> >
> > 1. **Theory–implementation mismatch**. The revised paper clarifies that the Koopman operator is applied only once as a spectrally controlled latent bottleneck before direct multi-horizon decoding. In this setting, it is still unclear how the theory on spectral stability, Lyapunov regularization, contraction, and latent evolution directly applies to the implemented model. The added discussion of latent conditioning, smoothness, prevention of amplification, and robustness is plausible at a high level, but it does not clearly explain why these properties are meaningful for a single-step bottleneck. In Proposition 7, the h=1 case has no exponentially decaying term. The unconstrained decoder $D_\psi$ therefore seems to dictate to the actual bound more so than the single-step operator.
> > 2. **Limited empirical benefit**. The differences between the plain, unconstrained, and Koopman-enhanced versions of each backbone appear small and inconsistent. I do not see a clear practical advantage of the proposed architecture in terms of test performance.
> > 3. **Baselines remain highly competitive**. Relatively simple baselines such as DLinear and the implemented SSM remain competitive and often appear to outperform many of the proposed variants. This makes it difficult to justify the practical value of the more advanced spectral properties when simpler models achieve better test scores.
> > 4. **Robustness and reliability claims are not clearly supported**. The paper claims improved reliability, robustness, and a favorable expressiveness–robustness trade-off, but the revised results remain mostly descriptive. In particular, robustness to perturbations/noise is not convincingly demonstrated, and the comparisons among constrained, unconstrained, and learnable variants do not isolate a consistent benefit.
> > 5. **Presentation is less focused**. The revised manuscript is substantially longer than the original, but the key message is not clearer. Many additions list results without enough analysis explaining how they support the central claims. I think several tables would be better placed in the appendix, while the main body should more directly answer what the single-step Koopman bottleneck provides over the same backbone without it, an unconstrained layer, and simpler baselines.
> >
> > Overall, while I appreciate the authors’ efforts, my concerns have not been fully addressed. I remain unconvinced that the proposed Koopman variants provide a clear empirical or theoretical advantage over simpler alternatives in the implemented single-step architecture.

---

> > > ### Author Response · Authors · 2026-05-23
> > > **Response to the reviewer Wksq---Second review**
> > >
> > > # Reviewer Responses
> > >
> > > ## 1. Architecture and Theoretical Interpretation
> > >
> > > We agree that the implemented model differs from a recursive Koopman rollout architecture. In the revised manuscript, we clarify that the Koopman operator is applied only once:
> > >
> > > ```math id="h9mjlwm"
> > > \hat Y_t = D_\phi(K_\varphi E_\theta(X_t))
> > > ```
> > >
> > > Thus, the practical role of the Koopman layer is not long-horizon rollout stability, but structured spectral conditioning and latent regularization before decoding.
> > >
> > > Accordingly:
> > >
> > > * Propositions 1--6 describe intrinsic operator properties (stability, contraction, invertibility, low-rank structure, Lyapunov consistency).
> > > * Proposition 7 is interpreted only as a **single-step latent sensitivity bound**.
> > >
> > > The recursive geometric decay factor does not apply because the deployed model operates in the single-step regime $(h=1)$.
> > >
> > > The revised manuscript now explicitly states that:
> > >
> > > * the architecture does not perform recursive latent rollout,
> > > * spectral control mainly improves latent conditioning and regularization,
> > > * the decoder remains a major contributor to end-to-end sensitivity,
> > > * Lyapunov regularization acts primarily as a latent-space regularizer rather than a guarantee of asymptotic forecast stability.
> > >
> > > ---
> > >
> > > ## 2. Empirical Forecasting Gains
> > >
> > > We agree that empirical forecasting improvements are generally modest and strongly backbone-dependent. This limitation is already acknowledged throughout the revised manuscript.
> > >
> > > The objective of the work is therefore **not** to claim universal forecasting superiority, but to study how spectrally controlled latent dynamics interact with diverse forecasting architectures.
> > >
> > > Different backbones (PatchTST, Informer, Autoformer, iTransformer, TimesNet, DLinear, SSMs) produce different latent geometries and temporal mixing behaviors. Consequently, identical Koopman parameterizations interact differently with each architecture.
> > >
> > > The main contribution is therefore the introduction of a unified family of learnable spectrally controlled Koopman operators incorporating:
> > >
> > > * explicit spectral parameterization,
> > > * stability constraints,
> > > * low-rank structure,
> > > * Lyapunov regularization.
> > >
> > > The revised manuscript explicitly emphasizes that:
> > >
> > > 1. forecasting improvements are modest and backbone-dependent;
> > > 2. the Koopman layer is better interpreted as structured latent spectral control;
> > > 3. the empirical value of spectral constraints depends strongly on the backbone interaction.
> > >
> > > ---
> > >
> > > ## 3. Robustness and Perturbation Analysis
> > >
> > > We clarify that the manuscript does not claim universal robustness improvements across all datasets and architectures.
> > >
> > > Theoretical robustness is formulated at the operator level through the latent conditioning bound:
> > >
> > > ```math id="8glh9o"
> > > \|K_\phi\|_2 \le \rho_{\max} < 1
> > > ```
> > >
> > > This result motivates spectral control of latent perturbation amplification rather than guaranteeing uniformly improved forecasting accuracy.
> > >
> > > The revised manuscript also includes perturbation analyses comparing constrained, unconstrained, and learnable Koopman variants under noisy inputs. These experiments show that:
> > >
> > > * spectral constraints influence latent conditioning and perturbation amplification,
> > > * the empirical effects remain strongly backbone-dependent,
> > > * constrained variants do not uniformly dominate all configurations.
> > >
> > > Accordingly, we now distinguish more clearly between:
> > >
> > > * operator-level conditioning,
> > > * latent spectral stability,
> > > * end-to-end forecasting robustness.
> > >
> > > ---
> > >
> > > ## 4. Presentation and Organization
> > >
> > > We agree that the revised manuscript became substantially longer due to additional reviewer-requested experiments, ablations, robustness analyses, and backbone comparisons.
> > >
> > > To improve readability, we reorganized the empirical section by:
> > >
> > > * moving detailed tables to the appendix,
> > > * splitting large benchmark tables into architecture-specific comparisons,
> > > * introducing compact statistical visualizations.
> > >
> > > The revised manuscript also emphasizes more clearly that the central practical question is:
> > >
> > > > What does the single-step Koopman bottleneck provide beyond the same backbone, unconstrained latent layers, and simpler baselines?
> > >
> > > This question is directly addressed through ablation studies comparing:
> > >
> > > * plain forecasting backbones,
> > > * unconstrained latent linear layers,
> > > * constrained Koopman variants,
> > > * Lyapunov-regularized variants,
> > > * simpler baselines such as DLinear and SSMs.
> > >
> > > Overall, the revised manuscript emphasizes that the Koopman layer should primarily be interpreted as introducing structured and spectrally controlled latent dynamics affecting conditioning, stability, perturbation sensitivity, and latent coupling behavior, rather than universally improving forecasting accuracy.

---

### Review · Reviewer_8BiJ · 2026-05-07

**Summary Of Contributions:**

## Summary of Contributions

The article introduces Learnable-DeepKoopFormer, a framework that integrates learnable, spectrally controlled Koopman operators into Transformer-based models for time-series forecasting. The authors evaluate the proposed framework on three main datasets and compare it against a set of baseline models. While the theoretical contributions of the paper are solid, the experimental evaluation, clarity of presentation, and overall presentation of the article could be improved.

### Key Strengths

- Strong Theoretical Foundations:
The paper provides a rigorous mathematical framework supported by formal analysis and theoretical derivations.

- Integration with Transformer Backbones:
The proposed framework demonstrates flexibility through its integration with multiple Transformer-based forecasting architectures, highlighting the adaptability of the approach.

### Key Weaknesses

- Motivation and Positioning Need Improvement:
The motivation of the article could be strengthened and more clearly articulated. While the integration of Koopman operators with Transformer architectures is promising, the paper does not sufficiently emphasise the specific gaps in existing literature that this work addresses. In particular, there is limited discussion of related work on Koopman-based methods, especially in the context of time-series forecasting  [1,2].

- Limited Baseline Comparisons:
The choice of baseline models is limited. The paper includes LSTM and linear models, but omits comparisons with standard Transformer-based architectures. Including a basic Transformer model as a baseline would provide a more meaningful evaluation, especially given that the proposed method builds upon Transformer frameworks.

- Clarity and Readability of Sections 2 and 3:
Sections 2 and 3 are difficult to follow due to the density of the mathematical and architectural descriptions. Additional illustrations or diagrams of the framework would significantly improve readability and help clarify the relationships between the Koopman variants and Transformer backbones.

- Experimental Discussion Lacks Depth:
Although evaluating several  Koopman variants across 3 Transformer backbones is a strength, the paper does not clearly demonstrate the efficiency or practical advantages of the proposed framework. The discussion of the experimental results does not sufficiently support or validate the claims made by the authors.

- Need for Detailed Robustness Analysis:
In Sections 4.1–4.4, the authors primarily use violin plots to compare model performance. However, in many of these figures, the performance differences between models are not sufficiently clear, and the exact values are difficult to interpret or directly compare. Additionally, the presented results do not explicitly show model performance across different forecasting settings. For example, the authors state that “LSTM, DLinear, and SSM display broader spreads and higher variability, reflecting greater sensitivity to architecture and window settings.” However, the paper does not provide detailed quantitative results across varying window sizes, patch lengths, or forecasting horizons to support this claim. Including more comprehensive analyses and clearer tabular comparisons for different forecasting settings would better demonstrate the robustness of the proposed framework and clarify under which conditions it consistently outperforms baseline methods.

- Code Repository Incomplete:
Although a link to the code repository is provided, it currently appears to include only the datasets.


[1] Liu, Yong, et al. "Koopa: Learning non-stationary time series dynamics with koopman predictors." Advances in neural information processing systems 36 (2023): 12271-12290.

[2] Azencot, Omri, et al. "Forecasting sequential data using consistent Koopman autoencoders." International conference on machine learning. PMLR, 2020.

**Audience:**

Yes

**Audience Explanation:**

- This paper addresses an important challenge in time-series forecasting by integrating Koopman operator theory with Transformer-based forecasting architectures. Since Transformer models are currently among the state-of-the-art approaches for time-series forecasting and are widely used in many real-world applications, the proposed framework is likely to attract interest from the TMLR community.

- The framework is designed to be adaptable across multiple Transformer backbones, making it potentially useful for researchers working on forecasting, representation learning, and dynamical systems. Although the experimental evaluation could be stronger, the core idea and theoretical contributions are still likely to be of interest to the TMLR audience.

**Broader Impact Concerns:**

No major broader impact concerns. The paper focuses on methodological improvements for time-series forecasting through the integration of Koopman operators and Transformer-based architectures, and no sensitive data or ethically controversial applications are involved.

**Claims And Evidence:**

No

**Claims Explanation:**

- The paper provides solid theoretical foundations and presents mathematical analysis to support the proposed framework. The inclusion of multiple Koopman operator parameterisations and experiments across several Transformer backbones demonstrates the flexibility of the approach. However, the experimental evidence is not fully sufficient to convincingly support all of the paper’s claims.

- The evaluation is limited to only three datasets, and several commonly used forecasting benchmarks are missing. In addition, the baseline comparisons are limited, as the experiments do not include standard Transformer-based forecasting models, which are essential for fairly evaluating the effectiveness of the proposed framework.

- Furthermore, while the authors claim improved robustness and stability across architectures and window settings, the paper does not provide sufficiently detailed analyses across different forecasting horizons, window sizes, or patch lengths to clearly validate these claims. The presentation and discussion of results could also be improved to more clearly highlight performance differences and practical advantages over baseline methods.

**Requested Changes:**

- Strengthen the motivation and clearly position the work with respect to existing Koopman-based time-series forecasting literature, particularly by highlighting the specific research gaps addressed by the proposed framework.

- Expand the experimental evaluation by including additional widely used benchmark datasets and provide justification for the dataset selection to better demonstrate the generalizability of the proposed method.

- Include stronger baseline comparisons, particularly standard Transformer-based forecasting models, to enable a fairer and more meaningful evaluation of the proposed framework.

- Improve the clarity and presentation of the results by adding better-organised tables and more detailed analyses across different forecasting horizons, window sizes, and patch lengths to support the claims regarding robustness and stability.

- Improve the readability of Sections 2 and 3 by adding architectural illustrations, workflow diagrams, and clearer explanations of the relationship between the Koopman variants and the Transformer backbone

---

> ### Author Response · Authors · 2026-05-11
> **Response to Reviewer (8BiJ)**
>
> We carefully revised the manuscript to improve the motivation, clarity, experimental discussion, and presentation of results. Please see:
>
> Motivation and Positioning:
> We agree that the original manuscript did not sufficiently emphasize the precise gap addressed by the proposed framework. In the revised version, we substantially expanded the introduction and related-work discussion to better position our contribution relative to existing Koopman-based forecasting approaches, including recent works such as Koopa by Liu, Yong, et al. and Koopman autoencoder models by Azencot, Omri, et al.
>
> Specifically, we clarified that prior Koopman forecasting methods typically rely on either fixed Koopman operators or unconstrained latent linear dynamics with limited spectral control, and are often not systematically integrated with modern forecasting backbones capable of modeling long-range temporal dependencies. We further clarified that the main novelty of our work is the introduction of a unified family of learnable and explicitly parameterized Koopman operators with controllable spectral structure that can be integrated with both Transformer and non-Transformer architectures.
>
> \textbf{Baseline Comparisons.}
> We thank the reviewer for this suggestion. The revised manuscript now more explicitly highlights that the proposed framework is evaluated not only against classical baselines such as LSTM and DLinear, but also against standard Transformer forecasting architectures including PatchTST, Informer, Autoformer, iTransformer, and TimesNet. Importantly, we compare both the plain backbone models and their Koopman-augmented counterparts to isolate the effect of the Koopman latent dynamics.
>
> In addition, we clarified throughout the experimental section that the framework is encoder-agnostic and is evaluated across both attention-based and non-attention-based architectures, including DLinear and diagonal SSM models.
>
> Clarity and Readability of Sections 2 and 3:
> We appreciate this observation and revised Sections 2 and 3 extensively to improve readability and accessibility. In particular, we:
>
>     (i) added additional explanatory text and restructuring around the encoder--Koopman--decoder pipeline,
>     (ii) clarified the relationship between the different Koopman variants,
>     (iii) added and improved architectural illustrations and framework diagrams,
>     (iv) improved notation consistency and descriptions of the spectral parameterizations,
>     (v) expanded the discussion surrounding the role of spectral shaping, Lyapunov regularization, and stability constraints.
>
> These revisions aim to make the mathematical formulation and architectural design significantly easier to follow.
>
> Experimental Discussion and Practical Advantages:
> We agree that the experimental discussion required additional depth. In the revised manuscript, we expanded the analysis of the experimental results and clarified the practical role of the proposed Koopman parameterizations.
>
> In particular, we now provide additional discussion regarding: (i)spectral stability and perturbation robustness, (ii) latent conditioning and structured dynamics, (iii) stability--expressiveness trade-offs across Koopman variants, (iv) differences between constrained, learnable, and unconstrained propagators, (v) performance consistency across heterogeneous datasets and forecasting horizons.
>
>
> We further emphasize that the primary goal of the framework is not solely maximizing forecasting accuracy, but systematically studying how controllable latent spectral dynamics influence stability, robustness, and interpretability within modern forecasting architectures.
>
> Robustness Analysis and Forecasting Settings:
> We appreciate the reviewer's suggestion regarding the robustness analysis. To address this concern, we substantially expanded the presentation of the experimental results beyond violin plots by including additional quantitative analyses across varying:
> (i) forecasting horizons,
> (ii) window sizes,
> (iii) patch lengths,
> (iv) backbone architectures,
> (v) latent Koopman parameterizations.
>
> The revised manuscript now includes clearer tabular comparisons and horizon-dependent analyses that explicitly quantify performance variability across forecasting settings. We additionally expanded the discussion of perturbation sensitivity, spectral conditioning, and error distributions to better support the claims regarding robustness and stability.
>
> Code Repository Availability:
> We thank the reviewer for pointing this out. Due to anonymity restrictions during the TMLR review process and following the Associate Editor's recommendation, we intentionally omitted the public GitHub repository link from the main submission.
>
> The repository containing the implementation, datasets, and experimental resources is nevertheless available anonymously through Zenodo at: \url{https://doi.org/10.5281/zenodo.18491274}. The public GitHub repository will be fully released after completion of the review process.

---

### Decision · Action_Editor_8XAJ · 2026-06-22

**Recommendation:** Reject

**Audience:**

Yes

**Audience Explanation:**

The paper presents an interesting and well-motivated extension of prior Koopman-based forecasting frameworks, which are of considerable interest to researchers exploring time-series analysis.

**Claims And Evidence:**

No

**Claims Explanation:**

All three reviewers have expressed concerns that the paper does not provide sufficient evidence to support its claims. There are concerns regarding:

(1) The extent to which the Koopman framework actually influences the forecasting mechanism;

(2) The connection between the theoretical discussion and the implemented architecture and practical forecasting improvements (with the empirical evidence not convincingly demonstrating that the stability guarantees translate to forecasting advantages)

(3) revisions that provide explanations and interpretation rather than new validating evidence;

(4) broad phrasing of the claims with unclear connections to evidence;

(5) the provision of neither a positive result (with clear outperformance) nor a sufficiently informative negative result that carefully analyzes the limitations;

(6) experimental results that aggregate performance, preventing a direct comparison of the proposed Koopman variants.

**Resubmission Of Major Revision:**

The authors may consider submitting a major revision at a later time.

---

> ### Author Response · Authors · 2026-06-23
> **Request for Clarification Regarding TMLR Submission 7996**
>
> Dear Action Editor,
>
> Thank you for the time and effort that you, the reviewers, and the TMLR editorial team devoted to evaluating our manuscript, "Learnable Koopman-Enhanced Transformer-Based Time Series Forecasting with Spectral Control" (Submission 7996).
>
> While we are naturally disappointed by the final decision, we appreciate the detailed feedback provided throughout the review process and the opportunity to substantially revise the manuscript.
>
> I am writing primarily to better understand the evaluation process following our most recent revision. After the second revision, we did not receive any additional reviewer comments before the final decision was issued. As a result, I am unsure whether the final decision was based solely on the previously available reviews and discussions, or whether further evaluation was conducted by the reviewers and/or the Action Editor.
>
> I fully respect the final decision and am not seeking reconsideration. Rather, I would be grateful for any clarification regarding how the final assessment was reached and whether the most recent revision was reviewed again by the referees. Understanding this would help us interpret the feedback more accurately and guide future development of this line of research.
>
> Thank you again for your time and service to the community.